# OPERATOR THEORY-DRIVEN AUTOFORMULATION OF MDPS FOR CONTROL OF QUEUEING SYSTEMS

**Victor Baillet[1], Yuanzhang Xiao[2], Nicolás Astorga[1] & Mihaela van der Schaar[1]**
[1]University of Cambridge    [2]University of Hawai'i at Mānoa

## ABSTRACT

Autoformulation is an emerging field that uses large language models (LLMs) to translate natural-language descriptions of decision-making problems into formal mathematical formulations. Existing works have focused on autoformulating mathematical optimization problems for *one-shot* decision-making. However, many real-world decision-making problems are *sequential*, best modeled as *Markov decision processes* (MDPs). MDPs introduce unique challenges for autoformulation, including a significantly larger formulation search space, and for computing and interpreting the optimal policy. In this work, we address these challenges in the context of queueing problems—central to domains such as healthcare and logistics—which often require substantial technical expertise to formulate correctly. We propose a novel operator-theoretic autoformulation framework using LLMs. Our approach captures the underlying decision structure of queueing problems through constructing the Bellman equation as a graph of *operators*, where each operator is an *interpretable* transformation of the value function corresponding to certain *event* (e.g., arrival, departure, routing). Theoretically, we prove a universal three-level operator-graph topology covering a broad class of MDPs, significantly shrinking the formulation search space. Algorithmically, we propose customized Monte Carlo tree search to build operator graphs while incorporating self-evaluation, solver feedback, and intermediate syntax checking for early assessment, and present a provably low-complexity algorithm that automatically identifies structures of the optimal policy (e.g., threshold-based), accelerating downstream solving. Numerical results demonstrate the effectiveness of our approach in formulating queueing problems and identifying structural results.

## 1 INTRODUCTION

Autoformulation with large language models (LLMs) aims to translate natural-language descriptions of decision-making problems into formal optimization models with minimal human intervention (Zhang et al., 2025b). It democratizes the access to advanced operations research (OR) modeling tools for non-OR domain experts and facilitates rapid prototyping and adaptation for OR practitioners (Gurobi Optimization, 2023; Wasserkrug et al., 2025).

Existing works on autoformulation have been focusing on *mathematical optimization*, which models *one-shot* decision-making (Ramamonjison et al., 2023; Xiao et al., 2023; AhmadiTeshnizi et al., 2024; Astorga et al., 2025; Bertsimas & Margaritis, 2024; Liang et al., 2025; Yang et al., 2025; Lu et al., 2025; Zhang et al., 2025a; Huang et al., 2025). However, many real-world scenarios evolve dynamically and stochastically, thus requiring *sequential* decision-making over time. These problems are naturally modeled as *Markov decision processes* (MDPs) (Puterman, 2014). Autoformulating MDPs presents unique challenges that cannot be addressed by current works on autoformulating optimization problems (see examples and Table 4 in Appendix A for a more detailed breakdown).

**Formulation Challenges.** Similar to autoformulating optimization, autoformulating MDPs requires searching the vast space of possible formulations. Moreover, MDPs have *additional components* (e.g., states, transition probabilities) and *implicit constraints* (e.g., nonnegative states, state-dependent action sets) that are not present in optimization and often omitted in the problem description. To ensure the accuracy, autoformulation must identify and infer these hidden structures (e.g., figuring out state transition probabilities of a queue from arrival and service rates).

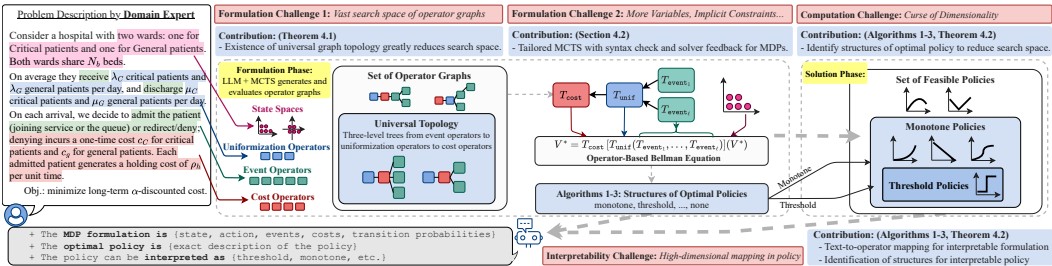

Figure 1: **Challenges in formulation, computation, and interpretation, and our contributions in addressing them.** ▶ Text-to-operator mappings improve *accuracy* and *interpretability of problem formulation*. ▶ Operator-based Bellman equations reveal *structures* of optimal policies (e.g., monotone or threshold) and value functions (e.g., convex), enhancing both *computational tractability* (by reducing the search space) and *interpretability* (by revealing structures of policies).

**Computational Challenges.** Many optimization problems (e.g., convex optimization) are considered solved once formulated (Boyd & Vandenberghe, 2004). In contrast, MDPs are notorious for the *curse of dimensionality* (Puterman, 1994). Although formulating and solving are two distinct phases, we advocate for autoformulation that is amenable to discovering *structural properties* of the optimal policy (e.g., the optimal action is monotone in the state value) *prior to* the solving phase, which can mitigate computational challenges (e.g., search among monotone policies instead of all policies).

**Interpretability Challenges.** Optimization models tend to be more readable, because variables and constraints in the problem formulation often have semantic meanings, and the optimal decision variables are easier to understand. In comparison, the optimal policy of an MDP is a mapping between multi-dimensional state-action pairs, which can be hard to interpret. Therefore, identifying structures of the optimal policy (e.g. monotonicity of action in state) makes the solution interpretable, which is important for decision-support systems (Hajek, 1984; Koole, 1995; Zhou et al., 2015).

**Our Solution.** Fig. 1 illustrates how our framework addresses the above challenges. Our work builds on the *operator theory* (Koole, 1998; 2007), which views the Bellman equation as a concatenation of *operators*. Each operator is an *interpretable* transformation of the value function that corresponds to certain event (e.g., arrival and departure in the context of controlling queue systems). The operator-based Bellman equation provides an interpretable problem formulation, as well as theoretical foundations for identifying structures of the optimal policy, which can reduce the complexity of solving MDPs and enhance the interpretability of the solution. We make significant contributions to fulfill the potential of operator theory-driven autoformulation of MDPs. ▶ *Operator graph and universal topology:* We are the *first* to represent Bellman equations as directed acyclic graphs (DAGs) of *operators*, namely the *operator graph*, and prove the existence of a *universal topology* for a large class of MDPs (Theorem 4.1). This greatly reduces the autoformualtion search space from all possible operator graphs to graphs with the fixed universal topology (**formulation challenge**). ▶ *Tailored Monte Carlo tree search (MCTS):* We propose a customized MCTS with LLMs to generate and evaluate operator graphs (Sec. 4.2), while incorporating LLM self-evaluation, solver feedback, and intermediate syntax checking to improve the accuracy and efficiency of autoformulation *without expensive fine-tuning of LLMs* (**formulation challenge**). ▶ *Automatic identification of structures:* We propose a low-complexity algorithm (Algorithms 1–3) that is guaranteed to identify theoretically known structures of the optimal policy (Theorem 4.2), thus mitigating the curse of dimensionality (**computation challenge**) and enhancing solution interpretability (**interpretability challenge**).

**Contributions.** ① *Conceptually*, we propose an operator theory-driven framework that *for the first time*, jointly automates the formulation of sequential decision-making problems from natural language and the discovery of the structures of the optimal policies (Sec. 4). Our novel view of Bellman equations as operator graphs addresses not only formulation challenges but also computation and interpretability challenges in the formulation phase. ② *Theoretically*, we rigorously prove the existence of a universal operator graph topology for a large class of MDPs, greatly reducing the search space of operator graphs (Theorem 4.1). ③ *Algorithmically*, we tailor MCTS for autoformulation of event-based MDPs by incorporating dense rewards and integrating the feedback from the solver for improved accuracy and efficiency (Sec. 4.2), and propose a provably low-complexity algorithm

Table 1: Comparison with representative works on autoformulation.

| Representative work | Problem formulated | Method | Challenges addressed | | |
|---|---|---|---|---|---|
| | | | Formulation | Computation | Interpretation |
| ORLM (Huang et al., 2025) | Mathematical optimization | Fine-tuning | Optimization-specific | – | – |
| Autoformulator (Astorga et al., 2025) | Mathematical optimization | Prompting | Optimization-specific | – | – |
| DPLM (Zhou et al., 2025) | Discrete-time dynamic programming | Fine-tuning | ✓ | ✗ | ✗ |
| **Our work** | Discrete-time and continuous-time MDPs | Prompting with operator-graph search | ✓ | ✓ | ✓ |

to automatically uncover structural results from the operator graph (Theorem 4.2). ④ *Empirically*, we create the first dataset on autoformulation of queueing problems, containing natural-language problem descriptions labeled with the optimal policies and their structures, and demonstrate the accuracy and efficiency of our framework.

## 2 RELATED WORKS

**Autoformulation of mathematical optimization.** There have been considerable efforts in creating datasets containing natural-language description of optimization problems (Ramamonjison et al., 2023; Yang et al., 2025) and developing LLMs and agents fine-tuned for optimization autoformulation (Xiao et al., 2023; AhmadiTeshnizi et al., 2024; Liang et al., 2025; Lu et al., 2025; Zhang et al., 2025a; Huang et al., 2025). Recent works have shown that through prompting and efficient MCTS, open-source LLMs can achieve comparable or better performance without the cost of fine-tuning (Bertsimas & Margaritis, 2024; Astorga et al., 2025). Our framework also uses MCTS without fine-tuning. However, we make significant contributions in addressing the formulation challenges *specific to MDPs*, and computation and interpretability challenges that *these works do not face*.

**Autoformulation of dynamic programming.** The most related is the recent work on autoformulating dynamic programming problems (Zhou et al., 2025). It focuses on synthetic dataset generation and LLM fine-tuning, but did not consider computation challenges and interpretability challenges.

**Addressing computation challenges in MDPs.** A large body of research addresses efficient solving of MDPs, notably through approximate dynamic programming (Bertsekas, 2012), reinforcement learning (Sutton, 2018), and exploiting structural properties of the solution (Yang, 2020; Koutas et al., 2025). These approaches are complementary to ours, and can be used in conjunction with our work after structural properties are identified in our formulation phase.

**Operator-based Bellman equations for control of queueing systems.** Significant OR research is devoted to uncovering structural properties of the optimal solution (Zhuang & Li, 2010; Hsu et al., 2015; Çil et al., 2011). Despite the unifying operator theory framework Koole (1998; 2007), such practices are still on a manual, case-by-case basis. In our attempt to autoformulate MDPs, we are *the first* to view the operator-based Bellman equation as an *operator graph*, and prove the novel result on the existence of a universal graph topology. In addition, we propose a low-complexity algorithm to automate the process of identifying structural results.

We compare with closely related works in Table 1 and discuss extended related works in Appendix B.

## 3 PROBLEM FORMULATION

Our framework autoformulates discrete-time MDPs, as well as continuous-time MDPs through their equivalent discrete-time *embedded* MDPs. Throughout the paper, we illustrate our framework using examples from healthcare (Chan et al., 2025; Bekker et al., 2017). But our framework can be applied to a variety of applications such as inventory management (Schwarz & Daduna, 2006), logistics (Adelman, 2007), transportation (Stidham, 1985; Ebben et al., 2004), and telecommunication (Koole & Mandelbaum, 2002; Bhulai & Koole, 2003); see Appendix C for a comprehensive list.

### 3.1 PRELIMINARIES

We summarize existing results in MDPs here, with detailed derivations in Appendix D.1.

**Control of Queuing Systems as Continuous-Time MDPs.** We consider continuous-time MDPs specified by six elements (Lippman, 1975; Serfozo, 1979): (1) a countable state space $\mathcal{S}$, (2) a finite set of eligible actions $\mathcal{A}_s$ in each state $s \in \mathcal{S}$, (3) a cost $\hat{c}(s, a)$ incurred by action $a$ in state $s$, (4) the state transition probability $\hat{P}(s' \mid s, a)$, (5) the *random* transition time $\tau$ from state $s$ to a different state after taking action $a$, which follows an exponential distribution with rate $\lambda(s, a)$, and (6) a discount rate $\alpha \geq 0$ that discounts the cost at time $t$ by $e^{-\alpha t}$.

**Optimization Criteria.** For a stationary policy $\pi : \mathcal{S} \to \mathcal{A}_s$, the $\alpha$-*discounted cost* is (Serfozo, 1979)

$$\hat{V}_{\alpha,\pi}(s) \triangleq \mathbb{E}_\pi \left[ \sum_{i=0}^{\infty} e^{-\alpha t_i} \hat{c}(s_i, a_i) \mid s_0 = s \right], \tag{1}$$

where $t_i$ is the time of the $i$-th state transition. The *average cost* is (Sennott, 2009; Serfozo, 1979)

$$\hat{J}_\pi(s) \triangleq \limsup_{t \to \infty} \mathbb{E}_\pi \left[ \frac{\sum_{i=0}^{I_t} \hat{c}(s_i, a_i)}{t} \mid s_0 = s \right], \tag{2}$$

where $I_t = \max\{i : t_i \leq t\}$ is the number of state transitions that have occurred up to time $t$.

**Discrete-Time Embedded MDPs and Standard Bellman Equations.** For an arbitrary upper bound of state transition rates $\Lambda > \sup_{s,a} \lambda(s, a)$, we define a discrete-time MDP with discount factor $\gamma = \frac{\Lambda}{\Lambda + \alpha}$ and the following state transition probabilities and the cost function

$$P(s' \mid s, a) = \begin{cases} \lambda(s, a) \cdot \hat{P}(s' \mid s, a)/\Lambda, & \text{if } s' \neq s \\ 1 - \lambda(s, a)/\Lambda, & \text{if } s' = s \end{cases} \text{ and } c(s, a) = \frac{\lambda(s, a) + \alpha}{\Lambda + \alpha} \cdot \hat{c}(s, a). \tag{3}$$

The discrete-time embedded MDP is obtained by setting a Poisson clock with rate $\Lambda$ and sampling the continuous-time process when the clock ticks. So the state may remain the same ($s' = s$).

Given a stationary policy $\pi$, the $\gamma$-discounted cost is $V_{\gamma,\pi}(s) = \mathbb{E}_\pi \left[ \sum_{i=0}^{\infty} \gamma^n c(s_i, a_i) \mid s_0 = s \right]$, and the average cost is $J_\pi(s) = \limsup_{I \to \infty} \mathbb{E}_\pi \left[ \sum_{i=0}^{I-1} c(s_i, a_i)/I \mid s_0 = s \right]$.

The discrete-time MDP $(\mathcal{S}, \mathcal{A}_s, c, P, \gamma)$ is *equivalent* to the continuous-time MDP $(\mathcal{S}, \mathcal{A}_s, \hat{c}, \tau, \hat{P}, \alpha)$, in the sense that $\hat{V}_{\alpha,\pi}(s) = V_{\gamma,\pi}(s)$ and $J_\pi(s) = \hat{J}_\pi(s)/\Lambda$ for any stationary policy $\pi$ (Serfozo, 1979). We solve the discrete-time MDP by solving the *standard* Bellman equation:

$$V_{n+1,\gamma}(s) = \min_{a \in \mathcal{A}_s} \left\{ c(s, a) + \gamma \cdot \sum_{s' \in \mathcal{S}} P(s'|s, a) V_{n,\gamma}(s') \right\}, \tag{4}$$

where $V_{n,\gamma}(s)$ is the *minimum* discounted cost during the last $n$ state transitions when starting from $s$.

Under mild conditions (Puterman, 2014), the *minimum* discounted cost $V_\gamma(s) = \inf_\pi V_{\gamma,\pi}(s)$ is the limit of $V_{n,\gamma}(s)$ when $n \to \infty$ (Sennott, 2009, Proposition 4.3.1), and the *minimum* average cost $J(s) = \inf_\pi J_\pi(s)$ is the limit of $(1 - \gamma)V_\gamma(s)$ when $\gamma \to 1$ (Sennott, 2009, Proposition 6.2.3). Therefore, it suffices to focus on the $n$-transition discounted cost $V_{n,\gamma}$ in the Bellman equation (4). For the remainder of the paper, we omit the discount factor in the subscript of $V_{n,\gamma}$ and use $V_n$.

### 3.2 EVENT-BASED MDP AND OPERATOR-BASED BELLMAN EQUATION

**Definition 3.1.** *Event-based MDPs* are MDPs whose state $s = (x, e)$ has two components: a controllable component $x$ (e.g., number of patients in the system) and an exogenous, uncontrollable component $e$ (e.g., arrivals), with transition probabilities decomposed as:

$$P\left[(x', e') \mid (x, e), a\right] = P_{\mathbf{x}}\left[x' \mid (x, e), a\right] \cdot P_{\mathbf{e}}(e' \mid x'). \tag{5}$$

Event-based MDPs are general enough to model decision-making problems in various applications (see Appendix C for a comprehensive list). Many problems in control of queuing systems are special cases of event-based MDPs, where transitions of the queuing state $x$ are deterministic.

Figure 2: **State transition dynamics of event-based MDPs**. Solid lines indicate temporal order, and dashed lines indicate dependency.

**Definition 3.2.** (Koole, 2007, Definition 3.1) Let $\mathcal{X}$ be the set of controllable state components $x$ and $\mathcal{V}$ be the set of all functions from $\mathcal{X}$ to $\mathbb{R}$. An *operator* is a mapping

$$T : \mathcal{V}^\ell \to \mathcal{V}, \quad \ell \geq 1. \tag{6}$$

The definition of an operator is general, giving us flexibility to express the Bellman equation as concatenation of operators. Given the context, the operators also have physical meaning.

For example, the prompt in Fig. 1 describes two parallel queues, one for critical ($C$) patients and one for general ($G$) patients, with the controlled arrivals (CA) and uncontrolled departures (D). The controllable state component is the queue lengths $x = (x_C, x_G)$. We can define *event operators*

$$T_{\text{CA}_C}[U(x)] = \min\{U[(x_C+1, x_G)], c_C + U(x)\}, \; T_{\text{D}_C}[U(x)] = U[((x_C-1)^+, x_G)], \tag{7}$$

$$T_{\text{CA}_G}[U(x)] = \min\{U[(x_C, x_G+1)], c_G + U(x)\}, \; T_{\text{D}_G}[U(x)] = U[(x_C, (x_G-1)^+)], \tag{8}$$

where $c_C$ and $c_G$ are costs of rejecting critical and general patients, $\text{CA}_C, \text{CA}_G$ and $\text{D}_C, \text{D}_G$ are arrival and departure events for critical and general patients, and $(\cdot)^+ \triangleq \max\{\cdot, 0\}$. We can also define a *uniformization operator*

$$T_{\text{unif}}[U_1(x), U_2(x), U_3(x), U_4(x)] = \tfrac{\lambda_C}{\lambda} \cdot U_1(x) + \tfrac{\lambda_G}{\lambda} \cdot U_2(x) + \tfrac{\mu_C}{\lambda} \cdot U_3(x) + \tfrac{\mu_G}{\lambda} \cdot U_4(x), \tag{9}$$

where $\lambda = \lambda_C + \lambda_G + \mu_C + \mu_G$, and a *cost operator*

$$T_{\text{cost}}[U(x)] = \rho_h \cdot (x_C + x_G)/(\lambda + \alpha) + \gamma \cdot U(x). \tag{10}$$

Then the Bellman equation for this system can be written as a composition of operators (i.e., an *operator graph*):

$$V_{n+1}^*(x) = T_{\text{cost}}\{T_{\text{unif}}(T_{\text{CA}_C}[V_n^*(x)], T_{\text{CA}_G}[V_n^*(x)], T_{\text{D}_C}[V_n^*(x)], T_{\text{D}_G}[V_n^*(x)])\}. \tag{11}$$

Note that for event-based MDPs, it is often more convenient to study the value function $V_n^*(x)$ defined on the controllable state component $x$, than the standard value function $V_n(s)$ defined on the full state $s$. The detailed derivations of the above results can be found in Appendix D.2.

Appendix E contains a more detailed coverage of operator theory beyond the example in Fig. 1.

## 4 METHOD

Our framework has two steps: autoformulation of operator-based Bellman equations and identification of structural results (see Fig. 1 for overview).

### 4.1 THEORETICAL FOUNDATION: UNIVERSAL TOPOLOGY OF OPERATOR GRAPHS

We can view the Bellman equation (11) as an operator graph with input $V_n^*$ and output $V_{n+1}^*$. If we view the process of problem formulation as searching in the space of all operator graphs, the search space is vast due to the variety of operators (Koole, 2007) and the many ways they can be connected (i.e., graph topology). Specifically, the number of possible DAGs with $N$ nodes (operators) and one out-point (the operator that outputs $V_{n+1}^*$) grows in the order of $2^{N^2}$ (Robinson, 1973). To have a sense of how fast this number grows with $N$, the numbers of possible DAGs for $N = 2, \ldots, 9$ are 1, 2, 15, 316, 16885, 2174586, 654313415, 450179768312 (Sloane, 2026).

We prove the existence of a *universal* graph topology for *all* event-based MDPs. This allows us to fix the graph topology, thus significantly reducing the search space.

**Theorem 4.1.** *For any event-based MDP with event set $\{e_1, \ldots, e_\ell\}$, its Bellman equation can be constructed by the following operator graph (the universal topology in Fig. 1):*

$$V_{n+1}^*(x) = T_{\text{cost}} \left\{ T_{\text{unif}} \left( T_{\text{e}_1} \left[ V_n^*(x) \right], \ldots, T_{\text{e}_\ell} \left[ V_n^*(x) \right] \right) \right\}, \tag{12}$$

*where* $T_{\text{cost}} \left[ U(x) \right] = c(x) + \gamma \cdot U(x)$, $T_{\text{unif}} \left[ U_1(x), \ldots, U_\ell(x) \right] = \sum_{j=1}^{\ell} P(e_j \mid x) \cdot U_j(x)$, *and* $T_{\text{e}_j} \left[ V_n^*(x) \right] = V_{n+1}(x, e_j)$.

*Proof.* See Appendix F. □

Theorem 4.1 reduces the search space to *three-level trees with $T_{\text{cost}}$ as the root, $T_{\text{unif}}$ as the single child of the root, and event operators as leaves*. Hence, in addition to specifying the universal topology, it also reduces the search space by specifying the *types of operators at each level of the tree*.

> **Takeaways:** ① Theorem 4.1 is crucial for reducing the search space. The original search space includes all the operator graphs with any topology and any operators as nodes, which is huge. Theorem 4.1 proves that there exists a universal topology: a three-level tree with certain types of operators at each level. This significantly reduces the search space. ② Theorem 4.1 is non-trivial, because it is possible to have operator graphs with alternative topologies for event-based MDPs, and because it may be impossible to construct an operator graph using the universal topology for non-event-based MDPs. See examples in Appendix G for details.

## 4.2 MCTS FOR AUTOFORMULATING EVENT-BASED MDPS.

Now that we know the universal graph topology, we aim to identify the correct nodes of the operator graph. Due to dependencies among components, MCTS is well-suited for this hierarchical search (see Appendix M for prompts). We decompose the search into four layers: problem parameters such as queue sizes ($m_1$), state variables and constraints ($m_2$), events, actions, costs, and their probabilities ($m_3$), and operators ($m_4$). This structure reflects the dependency hierarchy and guides exploration. Our MCTS follows the standard loop of *selection, expansion, evaluation,* and *backpropagation*, with two key modifications during backpropagation: (1) terminal nodes receive rewards from a combination of LLM preference and solver feedback, and (2) intermediate nodes are evaluated for syntax validity to provide dense supervision and penalize early errors.

**Terminal Rewards.** Every full rollout is scored relative to a baseline defined as the initial rollout. The LLM provides a preference score $\text{score}_{\text{LLM}} \in [0, 1]$. To reduce bias from LLM self-evaluation, we incorporate solver convergence $\text{score}_{\text{converged}} \in \{0, 1\}$, and compute the final reward as $\text{score}_{\text{final}} = \text{score}_{\text{LLM}} \times \text{score}_{\text{converged}}$.

**Intermediate Rewards.** Inspired by AlphaZero (Silver et al., 2018), we assign rewards to intermediate nodes based on syntactic correctness. If a partial formulation violates syntax constraints, the rollout is terminated early with a zero reward. This enables faster pruning of invalid branches and accelerates convergence.

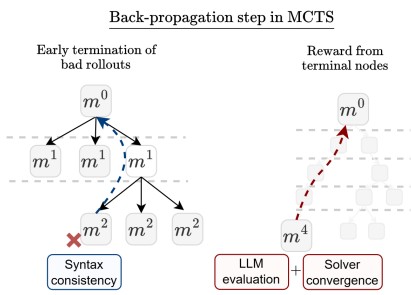

Figure 3: **MCTS for constructing operator graphs.** (1) Syntax check at intermediate nodes detects errors early, preventing failed full rollouts. (2) Solver feedback complements LLM self-evaluation for more objective rewards.

**Iterative Prompting.** Syntax errors are often local and should not always be penalized with a zero reward. We allow the LLM up to five attempts to fix syntax issues, using the error message as context. If correction fails, the error is attributed to earlier steps, and a zero reward is backpropagated. The backpropagation mechanism is illustrated in Fig. 3, with further details about our MCTS provided in Appendix H.

## 4.3 IDENTIFYING STRUCTURES OF OPTIMAL POLICIES USING DYNAMIC PROGRAMMING

Given the operator-based Bellman equations, we aim to identify the structures of optimal policies. This is usually done by identifying the properties of the value function $V^*(x)$. For example, if $V^*(x)$

is convex in $x$, the optimal policy $\pi^*$ is decreasing in $x$. For any operator $T$, we say that it *propagates* a property if, whenever $V_n$ satisfies the property, the transformed function $T[V_n]$ also satisfies it. For example, the linear cost operator $T_{\texttt{cost}}[U(x)] = \beta x + \gamma U(x)$ propagates monotonicity and convexity, because $T_{\texttt{cost}}[U(x)]$ is monotone (convex) in $x$ if $U(x)$ is monotone (convex) in $x$. In the following and in the Appendix, we also say that an operator propagates $A$, where $A$ is the *space* of functions having a certain property (for instance, all convex functions), and we may refer to $A$ as a "property" by abuse of language. There is some common wisdom regarding which properties certain typical operators propagate. However, since $V_n^*(x)$ needs to go through the operator graph, the challenge is to find the properties that are propagated by *all* operators in the graph. A bruteforce approach would require checking an exponentially growing number of possibilities.

To illustrate, consider two operators $T_1$ and $T_2$ and six spaces of functions $A$–$F$ with certain properties. For example, $A$ can be the space of convex functions, $B$ the space of increasing functions, and $A \cap B$ the space of convex increasing functions. Operator $T_1$ propagates $A \cap B$, $E \cap C$, and $D$, while operator $T_2$ propagates $C \cap A$, $D \cap F$, and $B$. In addition, we have $B \cap C \subset E$.

By computing the closure under intersection, $T_1$ is found to propagate, for example, $A \cap B \cap D$ and $E \cap C \cap D$. Identifying the smallest common space propagated by both $T_1$ and $T_2$ requires leveraging the inclusion relationship; in this case, the minimal shared space is $A \cap B \cap C$, which is not immediately evident from the original lists of properties that each operator propagates. A more detailed explanation of this example can be found in Appendix I.1.

We introduce a general dynamic programming algorithm to address this problem, and provide detailed description of the algorithm in Appendix I.

**Theorem 4.2** (Identification of Structural Results). *Given an operator graph $\mathcal{G}$, execution of Algorithms 1–3 gives us the set of properties propagated by all operators, with memory and time complexity of $\mathcal{O}(N \cdot |\mathcal{G}|)$ and $\mathcal{O}(N \cdot |\mathcal{G}|^2)$, where $|\mathcal{G}|$ is the number of operators in the graph, $N = \max_{T \in \mathcal{G}} n_T$, and $n_T$ is the number of properties propagated by operator $T$.*

*Proof.* See Appendix I.3. □

A direct brutal force approach would construct, for each operator, the full family of spaces obtained by closing its propagated properties under intersection, but this closure grows exponentially. A key observation in our algorithm is that any common propagated space can be written as the intersection of a subset of the properties that each operator initially propagates. Thus, instead of generating the full closure, we focus on identifying the properties that may appear in the intersection defining the smallest common propagated space. This is achieved by iteratively removing any property that cannot belong to this intersection. In the example above, $F$ never appears in any space propagated by $T_1$ and is not implied by any inclusion relationship, so no valid common space can involve $F$ in its intersection representation. Consequently, properties such as $D \cap F$ propagated by $T_2$ can be discarded. Repeating this pruning step yields a stable family of properties whose intersection is guaranteed to be both a common propagated space and the *smallest* of such spaces. Proofs and details on the treatment of inclusion relationships are provided in Appendix I.3.

> **Takeaway:** Theorem 4.2 guarantees that, given an operator graph, Algorithm 1 can identify *all* structural properties detectable within our framework. Thus, our ability to recover structure from a problem description depends entirely on the operator graph generated by the LLM.

## 5    EXPERIMENTS

**Dataset.** We constructed a dataset of 36 natural language descriptions of queueing control problems, varying in difficulty by size and shape of state spaces and number of event types. To assess performance in structure identification and support future research, the dataset includes three categories: (1) problems with provable structural results (e.g., Example 1); (2) problems with empirically observed, but unprovable, structures (e.g., Example 2); and (3) problems with no structural results. All problems are adapted from papers addressing realistic issues from domains such as hospital management (Bekker et al., 2017), telecommunications (Koole & Mandelbaum, 2002; Bhulai & Koole, 2003;

Table 2: Comparison with baselines and ablation study. Values in parentheses denote the number of completion tokens. Targeted prompts split the task into successive prompts, mirroring the steps of MCTS that can be found in Appendix M. SF: solver feedback, SC: syntax check.

| Method | 1 Rollout | 5 Rollouts | 12 Rollouts |
|---|---|---|---|
| GPT-4o (single prompt w. SF) | 0% (2k) | 0% (14k) | 0% (38k) |
| CoT (single prompt w. SF) | 0% (4k) | 2.7% (20k) | 2.7% (50k) |
| GPT-4o (targeted prompts w. SF) | 5.5% (2k) | 8.3% (16k) | 8.3% (42k) |
| CoT (targeted prompts w. SF) | 8.3% (6k) | 11.0% (36k) | 11.0% (85k) |
| MCTS (w/o. SF & SC) | 11.0% (10k) | 13.0% (43k) | 16.0% (85k) |
| MCTS (w. SF, w/o. SC) | 8.3% (10k) | 36.1% (44k) | 41.6% (86k) |
| GPT-4o (targeted prompts w. SF & SC) | 44.4% (6k) | 63.8% (30k) | 72.0% (80k) |
| CoT (targeted prompts w. SF & SC) | 52.7% (14k) | 72.2% (74k) | 75.2% (180k) |
| MCTS (w. SF & SC) | **63.8%** (11k) | **77.7%** (47k) | **83.3%** (96k) |

Bekker et al., 2011; Zhang et al., 2025c), freight dispatching (Schwarz & Daduna, 2006; Amjath et al., 2023), assembly lines (Adeyinka & Kareem, 2018), and traffic control (Boon et al., 2023).

**Experimental Setup.** For each problem, we perform multiple MCTS roll-outs and select the best candidate by greedily following the highest-scoring path. Each formulation is evaluated by running a dynamic programming solver; if it fails to converge, it is deemed incorrect. The resulting value function is compared to the ground truth and accepted if within a predefined tolerance. We apply our structure analysis algorithm to each roll-out. Results are summarized in the following tables and interpreted with respect to the challenges in Appendix K, focusing on correctness, tractability, and interpretability. The code and dataset are available here.

## 5.1 ACCURACY OF AUTOFORMALIZATION AND ERROR ANALYSIS

Table 2 shows that our autoformulation framework outperforms baseline methods. Single-prompt methods fail entirely to solve the task, even with CoT prompting. CoT is both more computationally demanding and less effective than MCTS as a test-time scaling strategy. In contrast, MCTS achieves better performance with the same level of feedback (LLM, solver feedback, or syntax check). Although the first rollout is relatively costly, MCTS becomes increasingly efficient by reusing prior computations. Notably, it achieves comparable or better performance using fewer computational resources than baseline methods.

Ablation study—comparing with MCTS without SF and/or SC—shows that the incorporation of syntax checks significantly enhances performance, as all formulations proposed by MCTS are executable by the solver—effectively addressing the challenge of *Syntactic Validity*.

The majority of remaining errors arise from *Semantic Misunderstanding*, including issues with *variable definitions*, *missing constraints*, *incorrect uniformizations*, and *misused operators*.

Figure 4: Accuracy against number of rollouts. Our method improves formulations continuously during search.

Table 3: Error types in failed roll-outs.

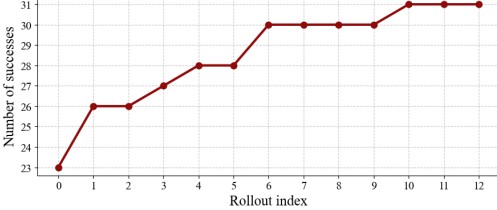

| Type of error | Occurrence |
|---|---|
| Parameter identification ($m^1$) | 5% |
| Variable definitions ($m^2$) | 26% |
| Missing constraints ($m^2$) | 24% |
| Incorrect events dynamics ($m^3$) | 12% |
| Incorrect uniformization ($m^3$) | 33% |

▶ *Parameter Identification errors* occur when the problem description includes irrelevant information that misleads the LLM. These cases are not dominant and can be caught via LLM self-evaluation.

► *Variable definition errors* often occur when the LLM introduces unnecessary queues. In hospital scenarios, for example, only one queue is needed for one ward, but the LLM may incorrectly use two queues, one for patients in beds and one for those waiting, to model one ward. ► *Missing constraints* arise from implicit assumptions in the problem description (e.g., number of patients should be non-negative). Solver feedback helps detect such issues by identifying unbounded state spaces. ► *Incorrect uniformizations* stem from deeper semantic misunderstandings. For instance, treatment probabilities differ when teams work in parallel vs. sequentially. These subtleties can be caught by LLM self-evaluation. ► *Incorrect event dynamics* is another issue. While most events are identified, the LLM may omit actions in large action spaces or invent spurious events (e.g., to model per-time-unit costs). These are typically detected via a combination of solver and LLM feedback.

## 5.2 COMPUTATIONAL TRACTABILITY AND INTERPRETABILITY

**Structure Identification.** When the operators identified by MCTS share a known propagated space, the second phase of our algorithm (Sec. 4.3) consistently recovers it, demonstrating that the *Structural Inference* challenge can be effectively addressed within our framework. Structural properties were identified in 74% of cases, also indicating strong performance on the second challenge: *Expressiveness of the Formulation*. Examples of successful structure extraction, along with a discussion of their interpretability, are provided in Appendix J. All failure cases fall into four categories: (i) *Incorrect problem translation by MCTS*, due to formalization errors discussed in Section 5.1, and not revisited here. (ii) *Operator mislabeling*, where the LLM correctly models state dynamics but misnames operators. This is the only remaining bottleneck for *Structural Inference* in our framework. (iii) *Limited structural expressiveness*, where the formulation is valid but does not expose the structure. This reveals that some correct formulations are less amenable to structural analysis. We illustrate this in Example 1. (iv) *Structural results beyond Koole's framework*, where certain properties cannot be captured regardless of the operator graph. These cases expose fundamental limitations of the current framework and suggest directions for future extensions. This is illustrated by Example 2.

> **Takeaway:** Quantitative evaluation across the dataset shows that our method correctly identifies 74% of the structural properties, *prior to* solving the problem. Therefore, we can reduce the computational complexity by calling specialized solvers for a large portion of the problems.

**Example 1** (Equivalent problem formulations with different structural expressiveness). *Our hospital has 1 ward that manages 2 types of patients with **shared** healthcare teams. There are $N_b$ beds in total. The average arrival rates of the patients are $\lambda_1$/hour and $\lambda_2$/hour respectively. The teams take care of patients **in parallel** with an average rate that depends on their type : $\mu_1$/hour and $\mu_2$/hour respectively. When a patient arrive we can refuse it, it occurs a cost of $c_1$ for the first type of patients and $c_2$ of the others.*

**Key challenges:** We cannot obtain structural results from the straightforward problem formulation. How to find an equivalent combination of operators that allow us to obtain structural results?

**Straightforward problem formulation.** The natural events of this problem are controlled arrivals and departures of the two types of patients, leading to the operator graph (found by MCTS):

$$V_{n+1}^* = T_{\text{cost}} \left\{ T_{\text{unif}} \left[ T_{\text{CA}(1)}(V_n^*), T_{\text{CA}(2)}(V_n^*), T_{\text{D}(1)}(V_n^*), T_{\text{D}(2)}(V_n^*), V_n \right] \right\}$$

In this formulation, the probabilities in $T_{\text{unif}}$ depend on the state. For instance,

$$p_{\text{D}(1)} = (\mu_1 n_1)/(\lambda_1 + \lambda_2 + \mu N_b) \quad \text{with} \quad \mu = \max(\mu_1, \mu_2).$$

Due to state dependent probabilities in $T_{\text{unif}}$, we cannot obtain any structural result.

**Equivalent problem formulation.** We define a new departure operator $T_{\text{D}_{\text{modified}}}$: ($\Gamma = \lambda_1 + \lambda_2 + \mu N_b$)

$$T_{\text{D}_{\text{modified}}(i)} f(x) = \frac{2\mu_1 n_1}{\Gamma - (\lambda_1 + \lambda_2)} f((x - e_i)^+) + \left(1 - \frac{2\mu_i n_i}{\Gamma - (\lambda_1 + \lambda_2)}\right) f(x)$$

With the new departure operator, the probabilities in $T_{\text{unif}}$ are independent of the state:

$$p_{\text{CA}(i)} = \lambda_i / \Gamma, \quad p_{\text{D}_{\text{modified}}(i)} = \frac{1}{2} \left[1 - (\lambda_1 + \lambda_2)/\Gamma\right].$$

**Structural results.** From the equivalent problem formulation, we can identify the *monotonicity* property of the optimal policy.

> **Takeaway:** Finding equivalent problem formulation with higher structural expressiveness is critical to identify structural results.

**Example 2** (Problem with intractable structural results). *Our hospital has 3 wards arranged sequentially, with capacities of 5, 15, and 15 beds, respectively. Each ward has its own healthcare team and manages its own patients. On average, new patients arrive at rates of 3, 20, and 5 patients/day in the respective wards. The wards serve patients one at a time at rates of 10, 5, and 3 patients/day, respectively. After being served in the first or second ward, we can transfer to the next ward at a cost of 2 per transfer or keep them in the current ward. Patients served in the third ward leave the hospital. Additionally, **patients can be moved back** from ward 2 to ward 1 at a rate of 3 patients/day or from ward 3 to ward 2 at a rate of 1 patient/day, each transfer incurring a cost of 2. Incoming patients can also be refused, incurring costs of 5, 10, and 15 for wards 1, 2, and 3, respectively.*

**Key challenges:** The solution to the problem exhibits structural properties, but these cannot be anticipated regardless of the choice of operator graph.

**Problem formulation.** Our autoformulator correctly output the operator graph:

$$V_{n+1} = T_{\text{cost}} \left\{ T_{\text{unif}} \left[ T_{\text{CA},1}(V_n), T_{\text{CA},2}(V_n), T_{\text{CA},3}(V_n), \right. \right. \tag{13}$$

$$\left. \left. T_{\text{CTD},(1,2)}(V_n), T_{\text{CTD},(2,3)}(V_n), T_{\text{CTD},(2,1)}(V_n), T_{\text{CTD},(3,2)}(V_n), T_{\text{D1},3}(V_n) \right] \right\}. \tag{14}$$

**Structural results.** We can observe the structures (e.g., monotone switching curve) of the optimal policy empirically (see Fig. 8 in Appendix J). However, we have yet to find an equivalent problem formulation that would express these structural results.

> **Takeaway:** The current method for identifying structural results fails on certain problems, as it depends on limited theoretical results.

## 6 DISCUSSION

We propose the first-ever autoformulator of event-based MDPs, a class of sequential decision-making problems encountered in various domains. Key to our framework is representation of the Bellman equation as an operator graph, which ensures interpretability and improves accuracy. By proving a universal operator graph topology for event-based MDPs, we significantly reduce the search space of autoformulation. We also propose a low-complexity algorithm to identify structural results based on the operator graph. To construct the operator graph, we make significant modifications to MCTS by evaluating intermediate nodes for denser rewards and utilizing solver feedback for more objective evaluations. Experimental results demonstrate the effectiveness of our approach in accurately formulating problems and uncovering key structural insights. We also create the first dataset on autoformulating queueuing control problems, with a wide variety of labeled problems.

**Autoformulating broader classes of operator graphs.** A natural direction for future work is to extend autoformulation beyond event-based MDPs, such as models with continuous actions or deterministic events occurring at fixed frequencies. While the universal topology of Theorem 4.1 may no longer hold in these settings, the operator graph viewpoint remains applicable, and the structural identification component of our framework applies to any operator graph without modification. The main challenge lies in searching for the correct operator graph topologies.

**Beyond analytic properties of the value function.** This paper focuses on analytic structural properties, such as convexity of the value function, that can be propagated through the operator graph. Many MDPs, however, do not admit such low-level analysis due to complex dynamics or irregular state spaces. We argue that the broader philosophy of designing autoformulators that are both computation-aware and interpretability-aware naturally extends to higher-level structural patterns beyond the analytical properties considered here. For example, hierarchical RL exploits the hierarchical structure inherent in certain tasks, illustrating how higher-level structure can guide algorithmic design even when low-level analytic results are unavailable. Incorporating such perspectives would enable autoformulation in domains with substantially more complex dynamics.

## ACKNOWLEDGEMENTS

We thank the anonymous ICLR reviewers, members of the van der Schaar lab, and Andrew Rashbass for many insightful comments and suggestions. Nicolás Astorga thanks W.D. Armstrong Trust for their support. Victor Baillet thanks Merck KGaA for their financial support. Yuanzhang Xiao was supported by the National Science Foundation under Grant NRT-AI 2244574.

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

CONTENTS

# A    AUTOFORMULATING OPTIMIZATION PROBLEMS VERSUS MDPs

Here is an example natural-language description of two parallel M/M/1 queues with controlled arrival and uncontrolled departure. It is adapted from problems in hospital admission control (Bekker et al., 2017; Naor, 1969). Some representative **formulation challenges** are annotated.

*We consider a hospital with two wards: one for critical patients and one for general patients, each staffed by a dedicated team. Both wards share $N_B$ beds. On average, $\lambda_C$ critical and $\lambda_G$ general patients arrive per day [State-dependent action sets: admission decision only at an arrival, readmission decision only at a departure.]. Treatment rates are $\mu_C$ and $\mu_G$ patients per day for Critical and General wards, respectively, with each team serving one patient at a time [Transition probabilities need to be inferred from the arrival and departure rates.]. Treated patients leave the system, releasing their beds. Upon arrival, a patient may be admitted if a bed is available; otherwise, the patient is rejected, incurring a penalty cost $c_C$ (critical) or $c_G$ (general). Each admitted patient generates a holding cost $\rho_h$ per unit time. The objective is to minimize the long-run average operating cost, with discount factor $\alpha$. [Throughout the description, the implicit constraint of queue length being nonnegative was not mentioned.]*

**Computation challenges.** The optimal policy maps queue lengths to controlled admission decisions for critical patients and general patients. Standard dynamic programming enumerates all queue lengths to determine the optimal decisions, repeating this process until convergence. However, if the optimal admission policy can be shown to follow a threshold structure, the search reduces to identifying the threshold values. Likewise, if the value function is convex, specialized dynamic programming solvers converge faster. Thus, uncovering structural properties of optimal policies and value functions before solving significantly reduces computational complexity.

**Interpretability challenges.** It is also desirable if the policy is interpretable, such as "admit only when the queue length is smaller than this threshold".

Next, we inspect a hospital logistics problem in the NL4Opt dataset (Ramamonjison et al., 2023).

*A hospital can transport their patients either using a type II ambulance or hospital van. The hospital needs to transport 320 patients every day. A type II ambulance is mounted on a truck-style chassis and can move 20 patients every shift and costs the hospital (including gas and salary) $820. A hospital van can move 15 patients and costs the hospital $550 every shift. The hospital can have at most 60% of shifts be hospital vans due to union limitations of the type II ambulance drivers. How many of shift using each type of vehicle should be scheduled to minimize the total cost to the hospital?*

In terms of formulation challenges, there is no notion of states or transition probabilities, and the decision variables are two scalars. In terms of computation challenges, this is a linear program that can scale with tens of thousands of decision variables. In terms of interpretability challenges, the optimal decisions are the numbers of vehicles of each type, which is easy to understand.

Table 4 provides a side-by-side comparison of challenges in autoformulating optimization and MDPs.

Table 4: Challenges in autoformulating mathematical optimization problems and MDPs.

| Challenge | Aspect | Optimization | MDP |
|---|---|---|---|
| **Formulation** | Variables | Often explicitly defined. | Distinction and dependence between states and actions need to be inferred. |
| | Constraints | Often explicitly specified. | May be implicit, incomplete, or omitted. |
| | Stochastic dynamics | Usually absent in static optimization | Central to the model and must be accurately represented. |
| **Computation** | Scalability | Convex instances often scale well with mature solvers. | Dynamic programming suffers from curse of dimensionality. |
| **Interpretability** | Problem formulation | Variables, objectives, and constraints usually inspectable | High-dimensional components, such as transition probabilities, are hard to inspect. |
| | Solution | Values of decisions often directly understandable. | Policies, as state-to-action mappings, can be difficult to interpret. |

# B  EXTENDED RELATED WORK

In Section 2, we mostly discussed work on autoformulation. We now provide additional background on two technical dimensions central to our approach: operator-based formulations of Bellman equations and DAG-based structures for the study of MDPs. Our goal is to position our operator-graph perspective within these broader lines of research.

## B.1  OPERATOR-BASED FORMULATIONS OF BELLMAN EQUATIONS

A substantial body of work has examined Bellman equations through the lens of operators acting on value functions. Classical dynamic programming texts already present the Bellman update as a *contraction operator* (Freedman, 1974; Blackwell, 1965). However, the formal meaning of "operator" varies significantly across the literature and is used in conceptually different ways.

The operators employed in our work are representative of one such interpretation. Koole (1998; 2007) introduced a collection of *elementary* or *event-based* operators from which the Bellman update can be constructed *by composition*. These elementary operators are not Bellman updates themselves; rather, they model primitive system events (arrivals, departures, etc.). The key insight is that one can establish structural properties (e.g., monotonicity) at the level of these elementary operators, and then deduce analogous results for broad classes of Bellman equations without repeating the full analysis. This philosophy has been adopted in subsequent work, which introduces additional operators to establish monotonicity or dominance properties in diverse stochastic control problems (Helm et al., 2011; Xiong et al., 2014; van Wijk et al., 2019; Benjaafar et al., 2010a;b). Once such properties are proved at the operator level, they can be reused whenever those operators appear as components of a Bellman equation.

Other definitions of operators focus on variants or generalizations of the canonical Bellman update. One example is the *generalized Bellman equation* (Yu et al., 2018), which defines an operator not only on value functions but on the parameter space of the learning algorithm itself (here, temporal-difference learning). Another example is the *distributional Bellman equation* (Bellemare et al., 2017), where the operator acts on the *distribution* of returns rather than the value function (i.e., the expected return). Finally, in some literature, the term *operator* is used more informally for individual pieces of the Bellman update, without the systematic decomposition seen in Koole's framework. For instance, in one of Bellman's original formulations (Bellman, 1952), the operator is essentially the full update except for the max or min optimization step, whereas in Yin et al. (2024), it is essentially the opposite, with the operator reduced to that optimization step alone.

## B.2  DAG-BASED STRUCTURES FOR THE STUDY OF MDPS

One classical use of DAGs in MDPs that is closely related to our operator DAGs is the AND/OR search graph, where nodes are time-indexed states and edges represent possible transitions, often weighted by their probabilities (Bonet & Geffner, 2012). In dynamic programming or reinforcement learning (RL), this structure is typically implicit, as it algebraically collapsed into the Bellman equation. In contrast, several algorithms—especially for finite-horizon problems—operate directly on this graph, such as THTS (Keller & Helmert, 2013) and the MCTS algorithm used in this paper. Our operator DAG can be viewed as an intermediate representation between the full AND/OR search graph and its complete collapse into the Bellman equation. It factorizes the large AND/OR graph using the analytical structure of the Bellman update, retaining enough of the original graph topology to decompose the Bellman update into more atomic analytical transformations, namely the operators.

Another use of DAGs in MDPs, which is less related to but can be confused with ours, arises in hierarchical reinforcement learning (HRL), where a decision problem is decomposed into higher-level tasks and lower-level subtasks. Each node in this hierarchy is itself a sequential decision problem, and the structure reshapes both the control problem and the learning dynamics of the optimal policy (Dietterich, 1999; Gopalan et al., 2017). The DAGs we consider in this paper are fundamentally different. Our operator DAG is an analytic hierarchy that specifies the order in which operations are applied to the value function within a single Bellman update. Although some nodes may involve a choice, they correspond to a one-shot evaluation of the policy, not to sub-MDPs with their own temporal evolution or repeated policy execution. HRL thus defines a hierarchy of behaviors unfolding

over many transitions, whereas our operator DAG defines a hierarchy of operations applied within one transition.

Beyond these settings, DAGs appear in many other parts of the MDP and RL literature, reflecting the general versatility of graph-based abstractions. These uses, however, are conceptually quite different from our operator DAGs. For example, factored MDPs represent the transition model as a dynamic Bayesian network (Guestrin et al., 2011), exploiting the conditional-independence structure to obtain compact transition models and more efficient computations. Other works employ graphs as high-level planning abstractions (Eysenbach et al., 2019; Zhang et al., 2021), for instance by performing Dijkstra-style shortest-path computations on an abstract task graph while delegating low-level control to an RL policy (Jothimurugan et al., 2021).

## C  EVENT-BASED MDPs IN VARIOUS APPLICATIONS

Table 5 provides a non-exhaustive list of event-based MDPs in a variety of applications domains. We focus on problems that can be modeled as control of queueing systems.

Table 5: Event-based MDPs in diverse application domains.

| Domain | Problem | Representative works |
|---|---|---|
| **Healthcare** | Diabetes management | (Bertsimas et al., 2017) |
| | Organ transplantation | (Berrevoets et al., 2021) |
| | Hospital admission control | (Bekker et al., 2017) |
| **Business** | Inventory management and logistics | (Schwarz & Daduna, 2006; Adelman, 2007) |
| | Assembly lines | (Adeyinka & Kareem, 2018) |
| | Freight dispatching | (Schwarz & Daduna, 2006; Amjath et al., 2023) |
| **Telecommunications** | Call center management | (Koole & Mandelbaum, 2002; Bhulai & Koole, 2003; Bekker et al., 2011; Zhang et al., 2025c) |
| **Transportation** | Intersection management | (Stidham, 1985; Ebben et al., 2004) |
| | Traffic control | (Boon et al., 2023) |

## D  Detailed Derivations of Results in Section 3

In this section, we provide detailed proofs and derivations of the results in Sec. 3. In the first part, we provide some key results on continuous-time and discrete-time MDPs to make the paper self-contained. In the second part, we go through the entire process of solving the example of parallel M/M/1 queues with controlled arrivals and shared beds (the prompt in Fig. 1).

### D.1  Key Results on Continuous-Time and Discrete-Time MDPs

#### D.1.1  Bellman Equations for Continuous-Time MDPs

To solve for the minimum cost, we define $\hat{V}_{n,\alpha}(s)$ as the *minimum* $\alpha$-discounted cost during the last $n$ state transitions when starting from state $s$. Defining $\hat{V}_{0,\alpha}(s) \equiv 0$, we have the following Bellman equation (Lippman, 1975; Serfozo, 1979)

$$\hat{V}_{n+1,\alpha}(s) = \min_{a \in \mathcal{A}_s} \left\{ \hat{c}(s,a) + \frac{\lambda(s,a)}{\lambda(s,a)+\alpha} \cdot \sum_{s' \in \mathcal{S}} \hat{P}(s' \mid s,a) \cdot \hat{V}_{n,\alpha}(s') \right\}, \tag{15}$$

where $\frac{\lambda(s,a)}{\lambda(s,a)+\alpha} = \mathbb{E}_{\tau \sim \mathrm{Exp}[\lambda(s,a)]}\left[e^{-\alpha\tau}\right]$ is the expected discounting of the next state value $\hat{V}_{n,\alpha}(s')$.

The Bellman equation (15) can be derived as follows.

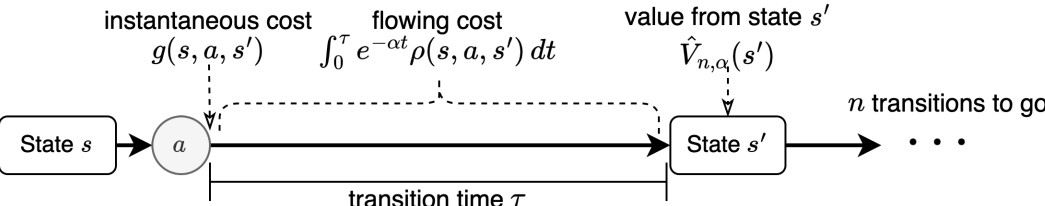

Figure 5: State transition dynamics and costs during a sample path of a continuous-time MDP.

The figure above shows the state dynamics and the costs incurred during the transition from state $s$ to state $s'$. The expected total cost starting from state $s$ consists of two parts: the expected cost accumulated until the transition and the expected total cost starting from the next state. The cost during the transition can be calculated as

$$\hat{c}(s,a) = \mathbb{E}_{\tau \sim \mathrm{Exponential}[\lambda(s,a)],\, s' \sim \hat{P}(\cdot|s,a)} \left[ g(s,a,s') + \int_0^\tau e^{-\alpha t} \rho(s,a,s')\, dt \right] \tag{16}$$

$$= \mathbb{E}_{\tau \sim \mathrm{Exponential}[\lambda(s,a)],\, s' \sim \hat{P}(\cdot|s,a)} \left[ g(s,a,s') + \rho(s,a,s') \cdot \int_0^\tau e^{-\alpha t}\, dt \right] \tag{17}$$

$$= \mathbb{E}_{\tau \sim \mathrm{Exponential}[\lambda(s,a)],\, s' \sim \hat{P}(\cdot|s,a)} \left[ g(s,a,s') + \rho(s,a,s') \cdot \frac{1 - e^{-\alpha\tau}}{\alpha} \right] \tag{18}$$

$$= \mathbb{E}_{s' \sim \hat{P}(\cdot|s,a)} \left\{ g(s,a,s') + \rho(s,a,s') \cdot \mathbb{E}_{\tau \sim \mathrm{Exponential}[\lambda(s,a)]}\left[ \frac{1 - e^{-\alpha\tau}}{\alpha} \right] \right\} \tag{19}$$

$$= \mathbb{E}_{s' \sim \hat{P}(\cdot|s,a)} \left[ g(s,a,s') + \rho(s,a,s') \cdot \frac{1}{\lambda(s,a)+\alpha} \right] \tag{20}$$

$$= \sum_{s' \in \mathcal{S}} \left[ g(s,a,s') + \frac{\rho(s,a,s')}{\lambda(s,a)+\alpha} \right] \hat{P}(s' \mid s,a) \tag{21}$$

Note that without discounting ($\alpha = 0$), the term $\frac{\rho(s,a,s')}{\lambda(s,a)}$ is the rate of cost $\rho(s,a,s')$ times the expected transition time $\frac{1}{\lambda(s,a)}$. With discounting, $\frac{1}{\lambda(s,a)+\alpha} = \mathbb{E}_{\tau \sim \mathrm{Exponential}[\lambda(s,a)]} \int_0^\tau e^{-\alpha t}\, dt$ can be interpreted as the expectation of "discounted" transition time.

The second part is the cost starting from the next state $s'$, discounted based on the transition time $\tau$:

$$\mathbb{E}_{\tau \sim \text{Exponential}[\lambda(s,a)]} \left[ e^{-\alpha\tau} \cdot \hat{V}_{n,\alpha}(s') \right] = \mathbb{E}_{\tau \sim \text{Exponential}[\lambda(s,a)]} \left[ e^{-\alpha\tau} \right] \cdot \hat{V}_{n,\alpha}(s') \quad (22)$$

$$= \frac{\lambda(s,a)}{\lambda(s,a) + \alpha} \cdot \hat{V}_{n,\alpha}(s'). \quad (23)$$

Note that this is the cost *if* the next state is $s'$. So we need to take the expectation over $s'$ to get the expected cost starting from state $s$ and action $a$:

$$\hat{Q}_{n,\alpha}(s,a) = \hat{c}(s,a) + \frac{\lambda(s,a)}{\lambda(s,a) + \alpha} \cdot \sum_{s' \in \mathcal{S}} \hat{P}(s' \mid s,a) \, \hat{V}_{n,\alpha}(s'). \quad (24)$$

Therefore, starting from state $s$, the *minimum* $\alpha$-discounted cost during the last $n+1$ transitions is

$$\hat{V}_{n+1,\alpha}(s) = \min_{a \in \mathcal{A}_s} \hat{Q}_{n,\alpha}(s,a), \quad (25)$$

which gives us the Bellman equation in (15).

We can see that for a continuous-time MDP, the Bellman equation (15) is not standard, because the discount factor $\frac{\lambda(s,a)}{\lambda(s,a)+\alpha}$ depends on the state-action pair $(s,a)$. We can remedy this issue by studying the discrete-time embedded MDP (Serfozo, 1979; Sennott, 2009; Lippman, 1975).

### D.1.2 EQUIVALENT DISCRETE-TIME MDPS AND VARIOUS PERFORMANCE CRITERIA

In Sec. 3, we introduced several notions of *value functions* for continuous-time MDPs:

- $\hat{V}_{\alpha,\pi}$: discounted cost of a continuous-time MDP with discount rate $\alpha$ and under policy $\pi$;
- $\hat{V}_\alpha$: minimum discounted cost of a continuous-time MDP with discount rate $\alpha$;
- $\hat{V}_{n,\alpha}$: minimum $\alpha$-discounted cost of a continuous-time MDP during the last $n$ transitions;
- $\hat{J}_\pi$: average cost of a continuous-time MDP under policy $\pi$;
- $\hat{J}$: minimum average cost of a continuous-time MDP.

The counterparts in discrete-time MDPs are $V_{\gamma,\pi}, V_\gamma, V_{n,\gamma}, J_\pi, J$, where we use regular letters and use the discount factor $\gamma$ in the subscript.

The relationship between these value functions are (Serfozo, 1979; Sennott, 2009)

$$\begin{array}{ccccccc}
& & \hat{V}_{n,\alpha} & & & & \\
& & {\scriptstyle n\to\infty}\Big\updownarrow & & & & \\
\text{continuous-time:} & \hat{V}_{\alpha,\pi} & \xrightarrow{\inf_\pi} \hat{V}_\alpha & & \hat{J} \xleftarrow{\inf_\pi} \hat{J}_\pi & & \\
& \Updownarrow & \Updownarrow & & \Updownarrow \quad \Updownarrow & & . \\
\text{discrete-time:} & V_{\gamma,\pi} & \xrightarrow{\inf_\pi} V_\gamma & \xrightarrow{\lim_{\gamma\to 1}(1-\gamma)V_\gamma} J \xleftarrow{\inf_\pi} J_\pi & & & \\
& & {\scriptstyle n\to\infty}\Big\updownarrow & & & & \\
& & V_{n,\gamma} & & & &
\end{array} \quad (26)$$

In (26), the equivalence "$\Leftrightarrow$" between the continuous-time MDP and its discrete-time embedded MDP defined in (27) is due to the result in Serfozo (1979), which we restated here.

**Theorem D.1** (Theorem in Serfozo (1979)). *Let $(\mathcal{S}, \mathcal{A}_s, \hat{c}, \tau, \hat{P}, \alpha)$ be a continuous-time MDP with bounded state transition rates ($\sup_{s,a} \lambda(s,a) < \Lambda$). Let $(\mathcal{S}, \mathcal{A}_s, c, P, \gamma)$ be the discrete-time MDP with discount factor $\gamma = \frac{\Lambda}{\Lambda + \alpha}$ and the following state transition probabilities and the cost function*

$$P(s' \mid s,a) = \begin{cases} \lambda(s,a) \cdot \hat{P}(s' \mid s,a)/\Lambda, & \text{if } s' \neq s \\ 1 - \lambda(s,a)/\Lambda, & \text{if } s' = s \end{cases} \quad \text{and} \quad c(s,a) = \frac{\lambda(s,a) + \alpha}{\Lambda + \alpha} \cdot \hat{c}(s,a). \quad (27)$$

*For any stationary policy $\pi$, we have $V_\gamma = \hat{V}_\alpha$ and $J_\pi = \hat{J}_\pi/\Lambda$.*

This transformation, often referred to as the *uniformization* technique, allows us to reduce a continuous-time MDP with exponential transition times into an equivalent discrete-time MDP with adjusted transition probabilities and costs.

The relationship "→" between other value functions are well-established results. The minimum discounted cost $V_\gamma(s)$ as the limit of $V_{n,\gamma}(s)$ when $n \to \infty$ is due to (Sennott, 2009, Proposition 4.3.1), and the minimum average cost $J(s) = \inf_\pi J_\pi(s)$ as the limit of $(1 - \gamma)V_\gamma(s)$ when $\gamma \to 1$ is due to (Sennott, 2009, Proposition 6.2.3). Since these are not the focus of our paper, we omit the formal statements of these results and the conditions under which they hold. We note that these conditions are usually satisfied in practice.

As we can see from (26), solving any value function can be done by solving the value function $V_{n,\gamma}$ for the discrete-time $n$-period discounted cost case. Suppose, for example, that we want to solve the value function $\hat{J}$ for the continuous-time average cost case. Due to the equivalence established by uniformization, we can instead solve for the discrete-time average cost $J$, which is obtained by solving $V_{n,\gamma}$ under a sufficiently large discount factor $\gamma$ and then taking the limit as $n \to \infty$.

### D.2 Detailed Walk-Through of the Example in Fig. 1: Parallel M/M/1 Queues with Controlled Arrivals and Shared Beds

For ease of reference, we rewrite the prompt here:

*We consider a hospital with two wards: one for critical patients and one for general patients, each staffed by a dedicated team. Both wards share $N_B$ beds. On average, $\lambda_C$ critical and $\lambda_G$ general patients arrive per day. Treatment rates are $\mu_C$ and $\mu_G$ patients per day for critical and general wards, respectively, with each team serving one patient at a time. Treated patients leave the system, releasing their beds. Upon arrival, a patient may be admitted if a bed is available; otherwise, the patient is rejected, incurring a penalty cost $c_C$ for a critical patient or $c_G$ for a general patient. Each admitted patient generates a holding cost $\rho_h$ per unit time. The objective is to minimize the long-run average operating cost, with discount rate $\alpha$.*

The ground truth model is a parallel M/M/1 queueing system with controlled arrivals (CA) and uncontrolled departures (D).

#### D.2.1 Uniformization of the Continuous-Time MDP

Let $x = (x_C, x_G)$ denote the current queue lengths and let $e \in \{A_C, A_G, D_C, D_G\}$ denote the most recent event: arrival of a critical patient, arrival of a general patient, departure of a critical patient, or departure of a general patient. We define the post-event state as $s = (x, e)$. The action set is empty for departure events. Upon arrivals, the decision is to accept (1) or reject (0). Hence

$$\mathcal{A}_{(x,e)} = \{0,1\}, \quad e \in \{A_C, A_G\}, \quad \text{if } x_C + x_G < N_B, \tag{28}$$

and

$$\mathcal{A}_{(x,e)} = \{0\}, \quad \text{if } x_C + x_G \geq N_B. \tag{29}$$

Interarrival and service times are naturally modeled as exponential (i.e., Poisson processes). Thus, the state transition time is exponentially distributed with rate

$$\lambda = \lambda_C + \lambda_G + \mu_C + \mu_G. \tag{30}$$

The state transition probability decomposes as

$$\hat{P}\left[(x', e') \mid (x, e), a\right] = \hat{P}_x[x' \mid (x, e), a] \cdot \hat{P}_e(e' \mid x'), \tag{31}$$

with transition probabilities for the queue lengths calculated as

$$\hat{P}_x\left[x' \mid (x, A_C), a\right] = \mathbf{1}_{x'_C = x_C + a}, \tag{32}$$

$$\hat{P}_x\left[x' \mid (x, A_G), a\right] = \mathbf{1}_{x'_G = x_G + a}, \tag{33}$$

$$\hat{P}_x\left[x' \mid (x, D_C), a\right] = \mathbf{1}_{x'_C = \max\{x_C - 1, 0\}}, \tag{34}$$

$$\hat{P}_x\left[x' \mid (x, D_G), a\right] = \mathbf{1}_{x'_G = \max\{x_G - 1, 0\}}. \tag{35}$$

Since the arrivals and departures are independent Poisson processes, the event probabilities are

$$\hat{P}_{\mathsf{e}}(\mathsf{A}_C \mid x') = \frac{\lambda_C}{\lambda}, \quad \hat{P}_{\mathsf{e}}(\mathsf{A}_G \mid x') = \frac{\lambda_G}{\lambda}, \quad \hat{P}_{\mathsf{e}}(\mathsf{D}_C \mid x') = \frac{\mu_C}{\lambda}, \quad \hat{P}_{\mathsf{e}}(\mathsf{D}_G \mid x') = \frac{\mu_G}{\lambda}. \tag{36}$$

The cost consists of one-time rejection penalties $c_C, c_G$ and a holding cost $\rho_h$ per patient per unit time, which can be calculated as

$$\hat{c}\left[(x, \mathsf{A}_C), 1\right] = \frac{\rho_h \cdot [(x_C + 1) + x_G]}{\lambda + \alpha}, \tag{37}$$

$$\hat{c}\left[(x, \mathsf{A}_C), 0\right] = c_C + \frac{\rho_h \cdot (x_C + x_G)}{\lambda + \alpha}, \tag{38}$$

$$\hat{c}\left[(x, \mathsf{A}_G), 1\right] = \frac{\rho_h \cdot [x_C + (x_G + 1)]}{\lambda + \alpha}, \tag{39}$$

$$\hat{c}\left[(x, \mathsf{A}_G), 0\right] = c_G + \frac{\rho_h \cdot (x_C + x_G)}{\lambda + \alpha}, \tag{40}$$

$$\hat{c}\left[(x, \mathsf{D}_C), a\right] = \frac{\rho_h \cdot (\max\{x_C - 1, 0\} + x_G)}{\lambda + \alpha}, \tag{41}$$

$$\hat{c}\left[(x, \mathsf{D}_G), a\right] = \frac{\rho_h \cdot (x_C + \max\{x_G - 1, 0\})}{\lambda + \alpha}. \tag{42}$$

### D.2.2 DERIVATION OF THE BELLMAN EQUATION

Using the discrete-time embedded MDP of the original continuous-time problem, we can write the following Bellman equations

$$V_{n+1}(x, \mathsf{A}_C) = \min\left\{V_n^*\left[(x_C + 1, x_G)\right], \; c_C + V_n^*(x)\right\}, \tag{43}$$
$$V_{n+1}(x, \mathsf{A}_G) = \min\left\{V_n^*\left[(x_C, x_G + 1)\right], \; c_G + V_n^*(x)\right\}, \tag{44}$$
$$V_{n+1}(x, \mathsf{D}_C) = V_n^*\left[(\max\{x_C - 1, 0\}, x_G)\right], \tag{45}$$
$$V_{n+1}(x, \mathsf{D}_G) = V_n^*\left[(x_C, \max\{x_G - 1, 0\})\right], \tag{46}$$

where $V_n^*(x)$ is the value function defined on the controllable state $x$ (i.e., queue lengths)

$$V_n^*(x) \triangleq \frac{\rho_h \cdot (x_C + x_G)}{\lambda + \alpha}$$
$$+ \gamma \left[\frac{\lambda_C}{\lambda} V_n(x, \mathsf{A}_C) + \frac{\lambda_G}{\lambda} V_n(x, \mathsf{A}_G) + \frac{\mu_C}{\lambda} V_n(x, \mathsf{D}_C) + \frac{\mu_G}{\lambda} V_n(x, \mathsf{D}_G)\right]. \tag{47}$$

Substituting the expressions in (43)–(46) into (47), we have

$$V_{n+1}^*(x) \triangleq \frac{\rho_h \cdot (x_C + x_G)}{\lambda + \alpha}$$
$$+ \gamma \left\{\frac{\lambda_C}{\lambda} \min\left\{V_n^*\left[(x_C + 1, x_G)\right], \; c_C + V_n^*(x)\right\}\right.$$
$$+ \frac{\lambda_G}{\lambda} \min\left\{V_n^*\left[(x_C, x_G + 1)\right], \; c_G + V_n^*(x)\right\}$$
$$+ \frac{\mu_C}{\lambda} V_n^*\left[(\max\{x_C - 1, 0\}, x_G)\right]$$
$$+ \left.\frac{\mu_G}{\lambda} V_n^*\left[(x_C, \max\{x_G - 1, 0\})\right]\right\}. \tag{48}$$

With the event operators, the uniformization operator, and the cost operator defined in Sec. 3.2, we have the operator-based Bellman equation in (11).

# E DETAILS ON OPERATOR THEORY

We provide an overview of the operator theory in constructing the operator-based Bellman equations. We focus on the results used in this paper. We refer the readers to Koole (2007) for a comprehensive discussion of this topic.

The central idea is to decompose the Bellman equation of an MDP into a sequence of operators (Koole, 2007). Each operator intuitively captures a distinct type of dynamic that arises as time progresses, such as randomness, decision-making, state transitions, or incurred costs.

## E.1 AN ILLUSTRATIVE EXAMPLE

We recall the example presented in Appendix D.2:

*We consider a hospital with two wards: one for Critical patients and one for General patients, each staffed by a dedicated team. Both wards share $N_B$ beds. On average, $\lambda_C$ Critical and $\lambda_G$ General patients arrive per day. Treatment rates are $\mu_C$ and $\mu_G$ patients per day for Critical and General wards, respectively, with each team serving one patient at a time. Treated patients leave the system, releasing their beds. Upon arrival, a patient may be admitted if a bed is available; otherwise, the patient is rejected, incurring a penalty cost $c_C$ (Critical) or $c_G$ (General). Each admitted patient generates a holding cost $\rho_h$ per unit time. The objective is to minimize the long-run average operating cost, with discount factor $\alpha$.*

This model involves several distinct dynamics:

- **Randomness**: Patient arrivals and departures occur at rates $\lambda_{C/D}$ and $\mu_{C/D}$, respectively.
- **Decision-making**: The system must decide whether to accept or reject a patient upon arrival.
- **State transitions**: The queue length may increase, stay the same, or decrease depending on the decisions made and event type.
- **Costs**: Rejection incurs a penalty, and holding patients generates a time-dependent cost.

These elements are all incorporated into the full Bellman equation. In operator theoery, we view the Bellman equation as a combination of operators, each capturing a specific aspect of the dynamics.

For the example above, the Bellman equation reads:

$$V_n^*(x) \triangleq \frac{\rho_h \cdot (x_C + x_G)}{\lambda + \alpha}$$
$$+ \gamma \cdot \left[ \frac{\lambda_C}{\lambda} V_n^*(x, \mathtt{A}_C) + \frac{\lambda_G}{\lambda} V_n^*(x, \mathtt{A}_G) + \frac{\mu_C}{\lambda} V_n^*(x, \mathtt{D}_C) + \frac{\mu_G}{\lambda} V_n^*(x, \mathtt{D}_G) \right]. \quad (49)$$

with

$$V_{n+1}^*(x, \mathtt{A}_C) = \min \left\{ V_n^*((x_C + 1, x_G), \ c_C + V_n^*(x) \right\}, \quad (50)$$
$$V_{n+1}^*(x, \mathtt{A}_G) = \min \left\{ V_n^*((x_C, x_G + 1), \ c_G + V_n^*(x) \right\}, \quad (51)$$
$$V_{n+1}^*(x, \mathtt{D}_C) = V_n^*(((x_C - 1)^+, x_G)), \quad (52)$$
$$V_{n+1}^*(x, \mathtt{D}_G) = V_n^*((x_C, (x_G - 1)^+)). \quad (53)$$

We can identify and separate the stochastic dynamics into the uniformization operator $T_{\mathtt{unif}}$, the decision-making upon arrival with its corresponding impact and cost into $T_{\mathtt{CA}}$, and the patient departure mechanism into $T_{\mathtt{D}}$. The holding cost and discount factor are captured by the cost operator $T_{\mathtt{cost}}$. These are formally defined as follows:

$$T_{\mathtt{CA,C}}[V_n^*(x)] = V_{n+1}(x, \mathtt{A}_C), \quad T_{\mathtt{CA,G}}[V_n^*(x)] = V_{n+1}(x, \mathtt{A}_G), \quad (54)$$
$$T_{\mathtt{D,C}}[V_n^*(x)] = V_{n+1}(x, \mathtt{D}_C), \quad T_{\mathtt{D,G}}[V_n^*(x)] = V_{n+1}(x, \mathtt{D}_G), \quad (55)$$

$$T_{\mathtt{unif}}[U_1(x), U_2(x), U_3(x), U_4(x)] = \frac{\lambda_C}{\lambda} \cdot U_1(x) + \frac{\lambda_G}{\lambda} \cdot U_2(x)$$
$$+ \frac{\mu_C}{\lambda} \cdot U_3(x) + \frac{\mu_G}{\lambda} \cdot U_4(x). \quad (56)$$

$$T_{\text{cost}}[U(x)] = \rho_h \cdot (x_C + x_G)/(\lambda + \alpha) + \gamma \cdot U(x). \tag{57}$$

Then the Bellman equation equation 49 can be rewritten as $V_n^*$ going through an *operator graph* to get $V_{n+1}^*$:

$$V_{n+1}^*(x) = T_{\text{cost}} \{T_{\text{unif}} (T_{\text{CA,C}}[V_n^*(x)], T_{\text{CA,G}}[V_n^*(x)], T_{\text{D,C}}[V_n^*(x)], T_{\text{D,G}}[V_n^*(x)])\}. \tag{58}$$

The motivation for introducing operators as a tool for autoformalism is twofold. First, it offers a structured framework wherein identifying the Bellman equation reduces to two subtasks: (i) identifying the relevant operators, and (ii) specifying the operator graph. In this work, we focus on a class of problems for which the second task—the structure of the graph—is solved in advance (Theorem 4.1). Thus, the only remaining task is to identify the appropriate operators in this graph. Second, the operator-based framework enables the automatic derivation of structural properties of the value function, by analyzing the propagation behavior of the operators within the graph.

For instance, consider the space of convex value functions $Conv$, defined by the property that $2V_n^*(x+1) \leq V_n^*(x) + V_n^*(x+2)$ for all $x \geq 0$. We can show that under any parameter values—provided the refusal cost is positive and the holding cost $\rho_h(x)$ is convex—the previously defined operators preserve convexity:

$$V^* \in Conv \implies T_{\text{CA}}(V^*), T_{\text{D}}(V^*), T_{\text{unif}}(V^*), T_{\text{cost}}(V^*) \in Conv.$$

Consequently, we have the propagation result:

$$V_n^* \in Conv \implies V_{n+1}^* = T_{\text{cost}} \Big\{ T_{\text{unif}} \Big( T_{\text{CA,G}}[V_n^*], \ T_{\text{CA,C}}[V_n^*],$$

$$T_{\text{CD,G}}[V_n^*], \ T_{\text{CD,C}}[V_n^*] \Big) \Big\} \in Conv. \tag{59}$$

From this, it follows that $V^* \in Conv$, which in turn implies that the optimal policy $\pi^*$ is *threshold*. Specifically, there exist two thresholds $n_{\text{T},C}$ and $n_{\text{T},G}$ such that an arriving Critical (resp. General) patient is accepted if and only if $x_C + x_G \leq n_{\text{T},C}$ (resp. $n_{\text{T},G}$)

This approach is generalizable: whenever the Bellman equation can be decomposed into known operators for which we have propagation results, structural properties of the value function—and hence of the optimal policy—can be deduced, provided these operators share a common invariant function space. In the following sections, we list the operators considered and detail their respective propagation properties.

### E.2 Operators Used in This Paper

We introduce the operators used in this work; additional operators can be found in Koole (2007). In the following, $\varepsilon_i$ denotes the canonical basis of $\mathbb{R}^k$, where $k$ represents the number of queues.

$$- T_{\text{cost}}[V^*(x)] = C(x) + \gamma V^*(x)$$

$$- T_{\text{unif}}[V_1^*(x), \ldots, V_J^*(x)] = \sum_{j=1}^{J} p(j) V_j(x) \quad \text{with } \sum_{j=1}^{J} p(j) = 1$$

$$- T_{\text{CA(i)}}[V^*(x)] = \min \{(V^*(x) + c_1, V^*(x + \varepsilon_i) + c_2)\}$$

$$- T_{\text{D1(i)}}[V^*(x)] = V^*((x - \varepsilon_i)^+)$$

$$- T_{\text{D(i)}}[V^*(x)] = \mu(x_i) V^*((x - \varepsilon_i)^+) + (1 - \mu(x_i)) V^*(x))$$

$$- T_{\text{CD}}[V^*(x)] = \begin{cases} \min\{(c_1 + V^*(x), c_2 + V^*[(x - \varepsilon_i)^+]\} & \text{if } x_i > 0 \\ c_1 + V(x) & \text{otherwise.} \end{cases}$$

$$- T_{\text{TD1(i,j)}}[V^*(x)] = \begin{cases} V^*(x - \varepsilon_i + \varepsilon_j) & \text{if } x_i > 0 \\ V^*(x) & \text{otherwise.} \end{cases}$$

$$- T_{\text{CTD(i,j)}}[V^*(x)] = \begin{cases} \min\{c_1 + V^*(x), c_2 + V^*(x - \varepsilon_i + \varepsilon_j)\} & \text{if } x_i > 0 \\ V^*(x) & \text{otherwise.} \end{cases}$$

The operator $T_{\text{cost}}$ represents the cost operator, while $T_{\text{unif}}$ corresponds to the uniformization operator.

The operator $T_{\text{CA},i}$ represents controlled arrivals to queue $i$, whereas $T_{\text{D1},i}$ models a standard departure. The operator $T_{\text{D},i}$ represents departures in a multi-server queue and is specifically used in Example 1. Controlled departures from queue $i$ are denoted by $T_{\text{CD},i}$.

To model tandem queues, we use the operator $T_{\text{TD1}}$, which describes the transition of a customer from queue $i$ to queue $j$. Similarly, $T_{\text{CTD}}$ represents controlled tandem departures.

### E.3 SET OF SPECIAL FUNCTIONS

We introduce the set of functions used in the propagation results.

$-\ V \in I(i)$ if
$$V(x) \le V(x + e_i)$$
for all $x \in \mathbb{N}^k, 1 \le i \le k$,

$-\ I = I(1) \cap \cdots \cap I(m)$,

$-\ V \in \text{UI}(i)$ if
$$V(x + e_{i+1}) \le V(x + e_i)$$
for all $x$ and $1 \le i < k$,

$-\ \text{UI} = \text{UI}(1) \cap \cdots \cap \text{UI}(k-1)$,

$-\ V \in Cx(i)$ if
$$2V(x + e_i) \le V(x) + V(x + 2e_i)$$
for all $x$ and $1 \le i \le k$,

$-\ Cx = Cx(1) \cap \cdots \cap Cx(k)$,

$-\ V \in \text{Super}(i,j)$ if
$$V(x + e_i) + V(x + e_j) \le V(x) + V(x + e_i + e_j)$$
for all $x$ and $1 \le i < j \le k$,

$-\ \text{Super} = \cap_{1 \le i < j \le k} \text{Super}(i,j)$,

$-\ V \in \text{Sub}(i,j)$ if
$$V(x) + V(x + e_i + e_j) \le V(x + e_i) + V(x + e_j)$$
for all $x$ and $1 \le i < j \le k$,

$-\ \text{Sub} = \cap_{1 \le i < j \le k} \text{Sub}(i,j)$,

$-\ V \in \text{SuperC}(i,j)$ if
$$V(x + e_i) + V(x + e_j + e_i) \le V(x + e_j) + V(x + 2e_i)$$
for all $x$ and $1 \le i, j \le k, i \ne j$

$-\ \text{SuperC} = \cap_{1 \le i,j \le k: i \ne j} \text{SuperC}(i,j)$,

$-\ V \in \text{SubC}(i,j)$ if
$$V(x + e_i) + V(x + e_j + e_i) \le V(x) + V(x + 2e_i + e_j)$$
for all $x$ and $1 \le i, j \le k, i \ne j$

$-\ \text{SubC} = \cap_{1 \le i,j \le k: i \ne j} \text{SubC}(i,j)$,

$-\ V \in \text{MM}(i,j)$ if
$$V(x) + V(x + d_i + d_j) \le V(x + d_i) + V(x + d_j)$$
for all $x$ and $1 \le i < j \le k$ such that $x + d_i, x + d_j \in \mathbb{N}^k$,
with $d_1 = e_1, d_k = -e_k + e_{k+11}, k = 2, \ldots, k - 1$, and $d_k = -e_k$,

$-\ \text{MM} = \cap_{1 \le i < j \le k} \text{MM}(i,j)$.

Further explanations for each can be found in Koole (2007). The following inequalities hold for these sets:

$$\text{Super}(i,j) \cap \text{SuperC}(i,j) \subset Cx(i)$$
$$\text{Sub}(i,j) \cap \text{SubC}(i,j) \subset Cx(i)$$
$$\text{MM} \subset \text{Super} \cap \text{SuperC} \subset \text{Super} \cap Cx$$

We construct a non-trivial inclusion basis that satisfies the rules introduced in Appendix I.3 and corresponds to the given inequalities. We define $\mathcal{B}$ as the union of all the spaces introduced above, excluding superspaces such as *SuperC*. Instead, we retain only the sets of the form $\text{SuperC}(i,j)$ to prevent non-trivial equalities that cannot be derived solely through *intersection* or *augmentation*.

Next, we decompose inequalities to ensure they follow the standard form $A \subset \{i\}$. For example, an inclusion such as

$$\text{MM} \subset \text{Super} \cap \text{SuperC} \tag{60}$$

is rewritten as multiple separate inequalities, such as $\text{MM} \subset \text{Super}(i,j)$ for specific indices $i, j$. The resulting set of inequalities constitutes the non-trivial inclusion basis, which satisfies the necessary constraints.

### E.4 PROPAGATION RESULTS

We present some of the propagation results for the operators used in this work:

$- T_{\text{CA(i)}} : I \to I, \ \ \text{UI} \to \text{UI}, \ \ Cx(i) \to Cx(i), \text{Super}(i,j) \to \text{Super}(i,j), \ \ \text{Sub} \to \text{Sub},$
$\text{Super}(i,j) \cap \text{SuperC}(i,j) \to \text{SuperC}(i,j), \text{Super}(i,j) \cap \text{SuperC}(j,i) \to \text{SuperC}(j,i),$
$\text{Sub}(i,j) \cap \text{SubC}(i,j) \to \text{SubC}(i,j), \text{Sub}(i,j) \cap \text{SubC}(j,i) \to \text{SubC}(j,i), \ \ \text{MM} \to \text{MM} \text{ for } i = 1;$
$- T_{D1(i)} : I \to I, \ \ I \cap \text{UI} \to \text{UI} \text{ for } i = m, \ \ I(i) \cap Cx(i) \to Cx(i), \ \ Cx(j) \to Cx(j)$
$\text{for } j \neq i, \ \ \text{Super} \to \text{Super}, \text{Sub} \to \text{Sub}, \ \ \text{SuperC}(j,k) \to \text{SuperC}(j,k) \ \ (j,k \neq i),$
$I(i) \cap \text{SuperC}(i,j) \to \text{SuperC}(i,j) \ \ (i \neq j), \ \ Cx(j) \cap \text{SuperC}(j,i) \to \text{SuperC}(j,i) \ \ (j \neq i),$
$\text{SubC}(j,k) \to \text{SubC}(j,k) \ \ (j,k \neq i), \ \ I(i) \cap \text{SubC}(i,j) \to \text{SubC}(i,j)(j \neq i),$
$Cx(j) \cap \text{SubC}(j,i) \to \text{SubC}(j,i), \text{UI} \cap \text{MM} \to \text{MM} \text{ for } i = m;$
$- T_{\text{TD1(i)}} : I \to I, \ \ \text{UI} \to \text{UI} \text{ for } i < m, \text{UI} \cap \text{MM} \to \text{ for } i < m, \text{UI} \cap Cx \cap \text{Super} \to$
$Cx \text{ for } i < m, \text{UI} \cap Cx \cap \text{Super} \to \text{Super} \text{ for } i < m;$

For the rest of the operators, again refer to Koole (2007).

# F  PROOF OF THEOREM 4.1

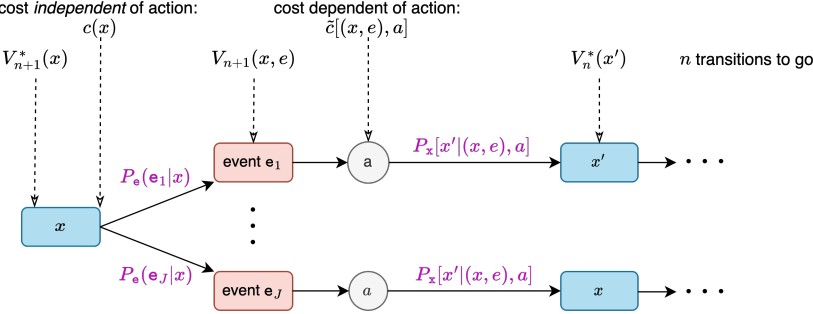

State transition dynamics and costs during a sample path of the discrete-time embedded MDP.

For any MDP $(\mathcal{S}, \mathcal{A}_s, c, P, \gamma)$, the standard Bellman equation for the value function $V_{n+1}(s)$ on the full state $s$ can be written as:

$$V_{n+1}(s) = \min_{a \in \mathcal{A}_s} \left\{ c(s, a) + \gamma \sum_{s' \in \mathcal{S}} P(s' \mid s, a) V_n(s') \right\}. \tag{61}$$

Separating the controllable state and the event $s = (x, e)$ and using the definition of event-based MDPs equation 5 we have

$$V_{n+1}(x, e) = \min_{a \in \mathcal{A}_{(x,e)}} \left\{ c[(x, e), a] + \gamma \sum_{(x', e')} P\left[(x', e') \mid (x, e),\, a\right] \cdot V_n(x', e') \right\} \tag{62}$$

$$= \min_{a \in \mathcal{A}_{(x,e)}} \left\{ c[(x, e), a] + \gamma \sum_{(x', e')} P_{\mathsf{x}}\left[x' \mid (x, e),\, a\right] \cdot P_{\mathsf{e}}(e' \mid x') \cdot V_n(x', e') \right\} \tag{63}$$

$$= \min_{a \in \mathcal{A}_{(x,e)}} \left\{ c[(x, e), a] + \sum_{x'} P_{\mathsf{x}}\left[x' \mid (x, e),\, a\right] \cdot \underbrace{\gamma \sum_{e'} P_{\mathsf{e}}(e' \mid x') \cdot V_n(x', e')}_{\triangleq V_n^*(x')} \right\} \tag{64}$$

$$= \min_{a \in \mathcal{A}_{(x,e)}} \left\{ c[(x, e), a] + \sum_{x'} P_{\mathsf{x}}\left[x' \mid (x, e),\, a\right] \cdot V_n^*(x') \right\}, \tag{65}$$

where we define the value function $V_n^*(x)$ on the controllable state :

$$V_n^*(x) = \gamma \sum_{e} P_{\mathsf{e}}(e \mid x) V_n(x, e). \tag{66}$$

$V_n^*(x)$ is the value of the state immediately after an action is taken but before the waiting time until the next event. Because there is a temporal gap between $V_n^*$ and $V_n$, the discount factor must be applied at this stage to account for that delay.

We can always decompose the cost $c[(x, e), a]$ into two parts: (1) a cost $c(x)$ that depends only on the controllable state (e.g., the holding cost in the M/M/1 example), and (2) a cost $\tilde{c}[(x, e), a]$ that depends on the full state-action pair, namely

$$c[(x, e), a] = c(x) + \tilde{c}[(x, e), a].$$

Note that if the component $c(x)$ does not exist, we can always set $c(x) = 0$ and $\tilde{c}[(x, e), a] = c[(x, e), a]$.

Then the Bellman equation in equation 65 can be rewritten as

$$V_{n+1}(x, e) = c(x) + \min_{a \in \mathcal{A}_{(x,e)}} \left\{ \tilde{c}[(x, e), a] + \sum_{x'} P_{\mathtt{x}}\left[x' \mid (x, e),\, a\right] \cdot V_n^*(x') \right\}, \tag{67}$$

and the value function $V_{n+1}^*(x)$ can be rewritten as

$$
\begin{aligned}
& V_{n+1}^*(x) \\
=\ & \gamma \sum_e P_{\mathtt{e}}(e \mid x) V_{n+1}(x, e) \\
=\ & \gamma \sum_e P_{\mathtt{e}}(e \mid x) \left[ c(x) + \min_{a \in \mathcal{A}_{(x,e)}} \left\{ \tilde{c}[(x, e), a] + \sum_{x'} P_{\mathtt{x}}\left[x' \mid (x, e),\, a\right] \cdot V_n^*(x') \right\} \right] \\
=\ & \underbrace{\gamma c(x)}_{\triangleq c'(x)} + \gamma \sum_e P_{\mathtt{e}}(e \mid x) \left[ \min_{a \in \mathcal{A}_{(x,e)}} \left\{ \tilde{c}[(x, e), a] + \sum_{x'} P_{\mathtt{x}}\left[x' \mid (x, e),\, a\right] \cdot V_n^*(x') \right\} \right].
\end{aligned}
\tag{68}
$$

Therefore, we can go from $V_n^*(x)$ to $V_{n+1}^*(x)$ through the following three operators:

$$
\begin{cases}
T_{\mathtt{e}_j}[V_n^*(x)] = V_{n+1}(x, \mathtt{e}_j) = \min_{a \in \mathcal{A}_{(x,e)}} \left\{ \tilde{c}[(x, e), a] + \sum_{x'} P_{\mathtt{x}}\left[x' \mid (x, e),\, a\right] \cdot V_n^*(x') \right\} \\
T_{\mathtt{unif}}[U_1(x), \ldots, U_\ell(x)] = \sum_{j=1}^{\ell} P(\mathtt{e}_j \mid x) \cdot U_j(x) \\
T_{\mathtt{cost}}[U(x)] = c'(x) + \gamma \cdot U(x)
\end{cases}
$$

The operator-based Bellman equation on the value function $V_n^*(x)$ can be written as

$$V_{n+1}^*(x) = T_{\mathtt{cost}}\left\{ T_{\mathtt{unif}}\left( T_{\mathtt{e}_1}[V_n^*(x)], \ldots, T_{\mathtt{e}_\ell}[V_n^*(x)] \right) \right\}. \tag{69}$$

## G WHY THEOREM 4.1 IS NOT TRIVIAL? – ILLUSTRATIVE EXAMPLES

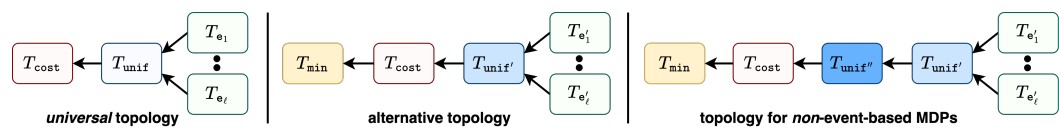

Figure 6: Universal topology for event-based MDPs, an alternative topology for an event-based MDP, and the topology for a non-event-based MDP.

In this section, we explain why the existence of a universal operator graph topology for event-based MDPs is not trivial. First, for an event-based MDP, it is possible to have operator graphs with alternative topologies. Second, for a non-event-based MDP, it may be impossible to construct an operator graph using the universal topology. We illustrate these two points through two examples. The summary of the results is illustrated in the figure above.

### G.1 EXAMPLE OF EVENT-BASED MDPS WITH MULTIPLE OPERATOR GRAPH TOPOLOGIES

---

**M/M/1 with Controlled Arrival and Controlled Departure**

We illustrate the model components using a simplified example of M/M/1 queues with controlled arrival (CA) and controlled departure (CD).

Patients arrive according to a Poisson process with rate $\lambda$ and wait to be served at rate $\mu$. The state $s = (x, e)$ has two components: the number of patients in the queue $x \in \mathbb{N}_+$ and the event $e \in \{A, D\}$, where $A$ denotes arrival and $D$ denotes departure. Upon arrival, we take action $a_{CA} \in \{0, 1\}$ to decide whether we accept (1) or deny (0) the patient. Upon departure, we take action $a_{CD} \in [0, 1]$, which is the probability of moving the patient back to the queue after the service is complete. This may happen when the care team is not yet confident that the patient can be discharged and wants to "re-service" the patient to reduce risk. So the action set is $\mathcal{A}_{(x,e)} = \{0, 1\}$ when $e = A$ and $\mathcal{A}_{(x,e)} = [0, 1]$ when $e = D$.

The state transition time is exponentially distributed with rate $\lambda(s, a) = \lambda + \mu$ when the system is not empty ($x > 0$) and with rate $\lambda(s, a) = \lambda$ when the system is empty. The state transition probability can be decomposed into

$$\hat{P}\left[(x', e') \,|\, (x, e), a\right] = \hat{P}_x\left[x' \,|\, (x, e), a\right] \cdot \hat{P}_e(e' \,|\, x'), \tag{70}$$

where the transition probabilities of the number of patients are

$$\hat{P}_x\left[x' \,|\, (x, A), a_{CA}\right] = \mathbf{1}_{x'=x+a_{CA}}, \tag{71}$$

$$\hat{P}_x\left[x' \,|\, (x, D), a_{CD}\right] = a_{CD} \cdot \mathbf{1}_{x'=x} + (1 - a_{CD}) \cdot \mathbf{1}_{x'=x-1}, \tag{72}$$

and the event probabilities are

$$\hat{P}_e(A \,|\, x') = \frac{\lambda}{\lambda + \mu}, \ \hat{P}_e(D \,|\, x') = \frac{\mu}{\lambda + \mu}, \ \text{for } x' > 0 \quad \text{and} \quad \hat{P}_e(A|0) = 1. \tag{73}$$

The cost includes a one-time reward $r$ for accepting a patient or a one-time penalty $p$ for rejecting a patient, a one-time cost $g_r(a_{CD})$ for re-servicing the patient that is increasing in $a_{CD}$, and a holding cost per unit time $\rho_h(x)$ that depends on the number of patients in the system. So the cost can be calculated as

$$\hat{c}\left[(x, A), 1\right] = -r + \frac{\rho_h(x+1)}{\lambda + \mu + \alpha}, \qquad \hat{c}\left[(x, A), 0\right] = c + \frac{\rho_h(x)}{\lambda + \mu + \alpha}, \tag{74}$$

$$\hat{c}\left[(x, D), a_{CD}\right] = g_r(a_{CD}) + \frac{a_{CD} \cdot \rho_h(x) + [1 - a_{CD}] \cdot \rho_h(x-1)}{\lambda + \mu + \alpha}. \tag{75}$$

---

The Bellman equations for this example are

$$V_{n+1}(x, \mathtt{A}) = \min \left\{ -r \cdot \frac{\lambda + \mu + \alpha}{\Lambda + \alpha} + V_n^*(x+1), \ c \cdot \frac{\lambda + \mu + \alpha}{\Lambda + \alpha} + V_n^*(x) \right\}, \quad (76)$$

$$V_{n+1}(x, \mathtt{D}) = \min_{a_{\mathtt{CD}} \in [0,1]} \left[ g_r(a_{\mathtt{CD}}) \cdot \frac{\lambda + \mu + \alpha}{\Lambda + \alpha} + a_{\mathtt{CD}} \cdot V_n^*(x) + (1 - a_{\mathtt{CD}}) \cdot V_n^*(x-1) \right], \quad (77)$$

where $V_n^*(x)$ is the value function defined on the queuing state $x$:

$$V_n^*(x) \triangleq \frac{\rho_h(x)}{\Lambda + \alpha} + \gamma \left[ \frac{\lambda}{\Lambda} \cdot V_n(x, \mathtt{A}) + \frac{\mu}{\Lambda} \cdot V_n(x, \mathtt{D}) + \left( 1 - \frac{\lambda + \mu}{\Lambda} \right) V_n(x, \varnothing) \right]. \quad (78)$$

Instead of the value function $V_n^*(x)$ on the queue length, we can decompose the Bellman equation on the standard value function $V_n(x, e)$ on the full state $(x, e)$.

For example, the value function $V_{n+1}(x, \mathtt{A})$ can be rewritten as

$$V_{n+1}(x, \mathtt{A}) = T_{\mathtt{min}} \Big\{ T_{\mathtt{cost},1} \left( T_{\mathtt{unif}'} \{ T_{\mathtt{A}'} [V_n^*(x)], T_{\mathtt{D}'} [V_n^*(x)], T_{\varnothing'} [V_n^*(x)] \} \right), \quad (79)$$

$$T_{\mathtt{cost},0} \left( T_{\mathtt{unif}'} \{ T_{\mathtt{A}'} [V_n^*(x)], T_{\mathtt{D}'} [V_n^*(x)], T_{\varnothing'} [V_n^*(x)] \} \right) \Big\}, \quad (80)$$

with the modified event operators

$$T_{\mathtt{A}'} [V_n^*(x)] = V_n(x, \mathtt{A}), \quad T_{\mathtt{D}'} [V_n^*(x)] = V_n(x, \mathtt{D}), \quad T_{\varnothing'} [V_n^*(x)] = V_n(x, \varnothing), \quad (81)$$

a modified uniformization operator

$$T_{\mathtt{unif}'} [U(x, \mathtt{A}), U(x, \mathtt{D}), U(x, \varnothing)] = \quad (82)$$

$$\frac{\rho_h(x)}{\Lambda + \alpha} + \gamma \left[ \frac{\lambda}{\Lambda} \cdot U(x, \mathtt{A}) + \frac{\mu}{\Lambda} \cdot U(x, \mathtt{D}) + \left( 1 - \frac{\lambda + \mu}{\Lambda} \right) U(x, \varnothing) \right], \quad (83)$$

an action-dependent cost operator

$$T_{\mathtt{cost},a} \{ U[(x, e), a] \} = c[(x, e), a] + U[(x, e), a], \quad (84)$$

and a minimization operator

$$T_{\mathtt{min}} \{ U[(x, e), a] \} = \min_{a \in \mathcal{A}_{(x,e)}} U[(x, e), a]. \quad (85)$$

## G.2 Example of Non-Event-Based MDP With a Different Operator Graph Topology

An example of a non-event-based MDP is the M/M/1 queue with controlled arrival and service rate optimization.

Patients arrive according to a Poisson process with rate $\lambda$ and wait to be served. The state $s = (x, e)$ has two components: the number of patients in the queue $x \in \mathbb{N}_+$ and the event $e \in \{\mathtt{A}, \mathtt{D}\}$, where $\mathtt{A}$ denotes arrival and $\mathtt{D}$ denotes departure. The action $a = (a_{\mathtt{CA}}, a_{\mathtt{RO}})$ consists of admission control ($\mathtt{CA}$) $a_{\mathtt{CA}} \in \{0, 1\}$ when a new patient arrives, where $0$ denotes rejection and $1$ denotes acceptance, and service rate optimization ($\mathtt{RO}$) $a_{\mathtt{RO}} \in \{\mu_1, \ldots, \mu_K\}$. The actions space is

$$\mathcal{A}_s = \begin{cases} \{0, 1\} \times \{\mu_1, \ldots, \mu_K\} & s \in \{(x, e) \mid e = \mathtt{A}\} \\ \varnothing \times \{\mu_1, \ldots, \mu_K\} & s \in \{(x, e) \mid e = \mathtt{D}\} \end{cases}. \quad (86)$$

When the system is not empty ($x > 0$), the state transition time is exponentially distributed with rate $\lambda(s, a) = \lambda + a_{\mathtt{RO}}$, and the state transition probability is

$$\hat{P}(s' \mid s, a) = \begin{cases} \mathbf{1}_{\{x' = x + a_{\mathtt{CA}}\}} \cdot \frac{\lambda}{\lambda + a_{\mathtt{RO}}}, & e = \mathtt{A}, e' = \mathtt{A} \\ \mathbf{1}_{\{x' = x + a_{\mathtt{CA}} - 1\}} \cdot \frac{a_{\mathtt{RO}}}{\lambda + a_{\mathtt{RO}}}, & e = \mathtt{A}, e' = \mathtt{D} \\ \mathbf{1}_{\{x' = x\}} \cdot \frac{\lambda}{\lambda + a_{\mathtt{RO}}}, & e = \mathtt{D}, e' = \mathtt{A} \\ \mathbf{1}_{\{x' = x - 1\}} \cdot \frac{a_{\mathtt{RO}}}{\lambda + a_{\mathtt{RO}}}, & e = \mathtt{D}, e' = \mathtt{D} \end{cases}. \quad (87)$$

The cost includes a one-time reward $r$ for accepting a patient or a one-time penalty $p$ of rejecting a patient, a holding cost per unit time $\rho_\mathrm{h}(x)$ that depends on the number of patients in the system, and a service cost per unit time $\rho_\mathrm{s}(a_\mathrm{RO})$ that depends on the service rate. So the cost can be calculated as

$$
\hat{c}(s, a) = \begin{cases} -r + \frac{\rho_\mathrm{h}(x+1)+\rho_\mathrm{s}(a_\mathrm{RO})}{\lambda+a_\mathrm{RO}+\alpha}, & e = \mathtt{A}, a_\mathtt{CA} = 1 \\ c + \frac{\rho_\mathrm{h}(x)+\rho_\mathrm{s}(a_\mathrm{RO})}{\lambda+a_\mathrm{RO}+\alpha}, & e = \mathtt{A}, a_\mathtt{CA} = 0 \\ \frac{\rho_\mathrm{h}(x)+\rho_\mathrm{s}(a_\mathrm{RO})}{\lambda+a_\mathrm{RO}+\alpha}, & e = \mathtt{D} \end{cases} . \tag{88}
$$

We can see from (87) that while the state transition probability can be decomposed, the probability of the event $P_\mathrm{e}(e \mid x, a)$ actually depends on the action. This is because the service rate affects the probability of the next event being an arrival or a departure.

In this case, we cannot use the universal topology for non-event-based MDPs.

# H    DETAILS ON THE PROPOSED MCTS TAILORED FOR AUTOFORMULATION

The Monte Carlo Tree Search (MCTS) formulates the problem in four steps, which correspond to four levels of the tree. A node in the first layer, denoted $m_1$, represents the parameters of the problem, such as the number and size of queues. A second-layer node $m_2$ represents the state variables and the constraints defining the state space. A third-layer node $m_3$ represents the possible events, their probabilities, the corresponding actions and operational costs. Finally, a fourth-layer node $m_4$ represents the operators associated with each event.

For a given problem description, each $m_i$ is identified through the standard MCTS steps: *selection*, *expansion*, *evaluation*, and *backpropagation*, omitting the *simulation* step in this context. Details are given bellow. For a give node $m_i$ we denote $\text{Child}(m_i)$ the children of the node in the tree.

## H.1    SELECTION

The selection step guides the search towards promising regions of the tree. Starting from the root, the algorithm recursively selects child nodes using the Upper Confidence Bound for Trees (UCT) (Kocsis & Szepesvári, 2006):

$$m_{i+1}^* = \arg \max_{m_{i+1} \in \text{Child}(m_i)} \left( V(m_{i+1}) + \omega \sqrt{\frac{\ln N(m_i)}{N(m_{i+1})}} \right). \tag{89}$$

This process continues until reaching an unexpanded node. Here, $m_{i+1}^*$ is the selected child node, $V(m_{i+1})$ is its estimated value, $N(m_i)$ and $N(m_{i+1})$ are visit counts for the parent and child nodes respectively, and $\omega$ is an exploration constant. This formula balances exploitation (first term, favoring high-value nodes) with exploration (second term, favoring less-visited nodes).

## H.2    EXPANSION

Upon reaching an unexpanded node $m_i$ of depth $i$, we generate its child nodes through an expansion process. Unlike traditional MCTS, which operates within a predefined search space, our approach explores an open-ended hypothesis space of component formulations. To facilitate this expansion, we employ LLMs as adaptive hypothesis generators. These models, conditioned on the partial formulation constructed up to node $m_i$, propose potential formulations for the next component in the search process.

At each node $m_i$, the LLM generates potential child nodes $m_i$ corresponding to next-step component formulations. This process follows the probability distribution:

$$p_\phi(m_{i+1}|m_{\leq i}, d), \tag{90}$$

where $d$ represents the problem description and $m_{\leq i}$ represents the partial formulation constructed up to that depth.

The LLM is queried using a structured prompt consisting of three components: (1) the original problem description $d$, provided in natural language; (2) the partial formulation $m_{\leq i}$, represented in JSON format; and (3) level-specific instructions that define the expected output format and relevant constraints. Additionally, we instruct the LLM to return candidate formulations using the same structured dictionary format to ensure consistency across iterations.

For each node expansion, we sample $H$ candidate formulations from the LLM's output distribution:

$$\text{Child}(m_i) = \{m_{i+1}^h \mid m_{i+1}^h \sim p_\phi(\cdot|m_{\leq i}, d), \forall h \in [H]\} \tag{91}$$

where $m_{i+1}^h$ represents the $h$-th candidate formulation.

When generating new candidate formulations, we systematically verify their syntax consistency with the existing partial formulation by evaluating the mathematical expressions as the constraints or the probabilities of the events. This step allows us to immediately discard invalid options, ensuring coherence throughout the expansion process. If a candidate fails to meet syntax consistency requirements, we re-query the LLM for a revised formulation.

If the maximum number of retries is reached, we assume that the inconsistency is not solely due to the stochastic nature of the LLM but rather stems from an issue in the existing partial formulation—such as a missing variable definition in earlier steps. In such cases, we terminate the rollout and immediately backpropagate a score of $0$ along the current branch.

## H.3 EVALUATION

After expanding a node, each newly created child node undergoes an initial evaluation to estimate its value, guiding subsequent selection in the search process. Assessing the correctness of a partial formulation relative to the original problem description is non-trivial, to address this challenge, we employ an LLM-based ranking evaluation for each set of child nodes, providing a more informed initial assessment.

Specifically we give the LLM the partial formulation till the current node $m_{\leq i}$ and let it rank the child nodes $\text{Child}(m_i)$. The resulting ranks are then center-normalized to the interval $[0, 1]$, with the middle rank positioned at 0.5. We define the normalized score as $s(m_{i+1}^h)$, which is used to initialize the value of each child node:

$$V_{\text{prior}}(m_{i+1}^h) \leftarrow s(m_{i+1}^h). \tag{92}$$

This approach deviates from traditional Monte Carlo Tree Search (MCTS), which typically assigns uniform priors to newly expanded nodes. Instead, the LLM evaluates the formulations by incorporating optimization principles and problem-specific context, potentially capturing aspects such as formulation correctness, constraint feasibility, and alignment with the overall problem structure.

## H.4 BACKPROPAGATION

Unlike conventional Monte Carlo Tree Search (MCTS), which typically simulates the problem to a terminal state after expanding a child node, our approach continues expansion until a terminal node $m_t$ is reached. The resulting formalization $m_{\leq t}$ is evaluated against a baseline, typically the formalization obtained after one rollout. The LLM assigns a score between $0$ and $1$ based on its preference for the new formalization over the baseline. To mitigate bias in the LLM signal $s_{\text{LLM}}$, we also check whether the solver successfully converges on the formalization, setting $s_{\text{converged}} = 1$ if it converges and $0$ otherwise. The final backpropagated score is then given by $s_{\text{LLM}} \times s_{\text{converged}}$.

The backpropagation process consists of updating the value of each node $m_i$ along the current branch using the following update rule:

$$V_{\text{back}}(m_i) \leftarrow \frac{V_{\text{back}}(m_i) \cdot N(m_i) + s_{\text{LLM}} \times s_{\text{converged}}}{N(m_i) + 1} \tag{93}$$

where $N(m_i)$ denotes the number of times the value of $m_i$ has been updated. After applying this update, we increment the count:

$$N(m_i) \leftarrow N(m_i) + 1. \tag{94}$$

# I   ALGORITHMS FOR IDENTIFYING STRUCTURAL RESULTS (SECTION 4.3)

In this section, we provide a detailed discussion of the algorithm used to identify structural properties of the solution from the operator graph (Section 4.3).

## I.1   WHY IT IS CHALLENGING TO IDENTIFY STRUCTURAL RESULTS?

Given the operator graph, our goal is to deduce structural properties of the optimal policy and the value function $V^*(x)$. Consider the example from Section 3, further detailed in Appendix E.1. In that case, one can show that $V^*(x)$ is convex, which implies that the optimal acceptance policy is threshold. This follows from the fact that convexity is *propagated* through each operator in the graph. Consequently, convexity is preserved by the entire Bellman equation. Since convexity is a fixed property under this composition of operators, it must also be a property of the fixed point of the Bellman equation—namely, the optimal value function.

More generally, consider a problem where the operator graph consists of operators $(T_1, \ldots, T_k)$. For each operator, we are given a list of functional spaces (or properties) that it propagates. Our objective is to identify a common functional space propagated by all operators; the optimal value function must then belong to this space.

In practice, we do not have explicit lists of all propagated spaces, but only a set of primitive spaces from which additional spaces can be generated. New propagated spaces may be obtained by applying the two fundamental set operations, intersection and union. Indeed, if $T$ propagates $A$ and $B$, then it propagates both $A \cap B$ and $A \cup B$.

In principle, one could therefore consider the closure of the initial families under both intersection and union. However, because the initial spaces are only intersections of basis spaces, and because we are ultimately interested in the smallest common propagated space across all operators, it suffices to consider closure under intersection only.

A brief intuitive justification is as follows. Suppose a space of the form $A \cup B$ (with $A \neq B$) appears in the propagated closure for the operators. Writing this set in its Disjunctive Normal Form (a union of intersections), we obtain a union of terms, each of which must be propagated individually by all operators, since unions do not appear in the primitive families. If we now replace every union in this representation by an intersection, we obtain a new space that is propagated by all operators and is contained in the original set. Consequently, any common propagated space built using unions admits a smaller counterpart formed purely through intersections. As a result, the smallest common propagated space is built entirely from intersections, and closure under intersections is therefore sufficient.

Our problem is therefore to find the smallest element (under $\subseteq$) among all intersection-closures of the propagated spaces associated with each operator. This task is made more difficult by the presence of non-trivial inclusion relationships between these spaces, which may cause distinct expressions to represent the same underlying space.

To illustrate these challenges, consider the example discussed in Section 4.3, involving two operators $T_1$ and $T_2$ and six properties $A$ through $F$:

$$\begin{cases} T_1 \text{ propagates } A \cap B, \ E \cap C, \ D, \\ T_2 \text{ propagates } C \cap A, \ D \cap F, \ B, \\ B \cap C \subset E. \end{cases}$$

Taking the closure under intersection, we obtain for $T_1$ the family of propagated spaces

$$\mathcal{P}_1 = \{A \cap B, \ E \cap C, \ D, \ A \cap B \cap D, \ E \cap C \cap E, \ A \cap B \cap E \cap C, \ A \cap B \cap E \cap C \cap D\}.$$

Similarly, for $T_2$ we obtain

$$\mathcal{P}_2 = \{C \cap A, \ D \cap F, \ B, \ C \cap A \cap B, \ D \cap F \cap B, \ C \cap A \cap D \cap F, \ C \cap A \cap D \cap F \cap B\}.$$

At first glance, the intersection $\mathcal{P}_1 \cap \mathcal{P}_2$ appears empty. However, using the inclusion $B \cap C \subset E$, we see that

$$A \cap B \cap C = A \cap B \cap C \cap E,$$

which shows that $A \cap B \cap C$ belongs to both families. Hence,

$$\mathcal{P}_1 \cap \mathcal{P}_2 = \{A \cap B \cap C\}.$$

In general, for a typical operator, the number of primitive propagated spaces may grow quadratically with the size of the state space, while the intersection-closure of these spaces can grow exponentially in the number of primitives. Consequently, a naive approach that explicitly computes every propagated space for each operator and then intersects them is computationally infeasible.

To address this, we introduce a dynamic programming algorithm that reduces both the time and memory complexity of the procedure. The following sections present a detailed discussion of this algorithm, including a proof of convergence, an analysis of its computational complexity and a running example.

## I.2 ALGORITHM DESCRIPTION IN PSEUDO-CODE

---
**Algorithm 1** Find smallest common propagated space

---
**Require:** $\mathcal{O}$ : A set of operators for which we know propagation results.
**Require:** $\mathcal{B}$ : A set of basis function spaces.
**Require:** $P$ : The propagation results for all the operators in $\mathcal{O}$.
**Require:** $(T_1, \ldots, T_J)$ list of operators in the graph.
**Require:** $R$ the non-trivial inclusion basis.
**Ensure:** $\mathcal{F}$ the smallest common propagated space.
 1: Create a mapping $m$ between $\{1, \ldots, K\}$ and $\mathcal{B}$, with $K = \#\mathcal{B}$.
 2: Create the sets $\mathcal{P}_j \in \mathcal{P}(\{1, \ldots, K\})$ for each $j$ based on $P$ and $m$.
 3: Create a dictionary $\mathcal{R}$ based on $R$ and $m$ such that if $\{i\} \subset^* U$ then $U \in \mathcal{R}[i]$.
 4: $\mathcal{P}_j^0 \leftarrow \mathcal{P}_j$ for all $j$
 5: $n \leftarrow 0$
 6: **while** $\{P\}_{n=0}^{\infty}$ does not converge **do**
 7:    $(p_k^n)_{k \leq J} \leftarrow \left( \bigcup_{p \in \mathcal{P}_j^n} p \right)_{k \leq J}$
 8:    **for** $j$ in $(1, \ldots, J)$ **do**
 9:       $\mathcal{P}_j^{n+1} \leftarrow$ **Refine_propagated_space**$(\mathcal{P}_j^n, (p_k^n)_{k \leq J}, \mathcal{R})$
10:    **end for**
11:    $n \leftarrow n + 1$
12: **end while**
13: $p^{\infty} \leftarrow \bigcup_{p \in \mathcal{P}_1^{n-1}} p$
14: Create $\mathcal{F}$ by mapping back $p^{\infty}$ to $\mathcal{B}$ using $m$
15: **Return** $\mathcal{F}$

---

---

**Algorithm 2** Refine Propagated Spaces

---

**Require:** $\mathcal{P}_j^n$: Set of elements of $\mathcal{P}(\{1, \ldots, K\})$
**Require:** $(p_k^n)_{k \leq J}$: List of sets of integers in $\{1, \ldots, N\}$, $\mathcal{B}_k^n$ is the set of elements that appear in $\mathcal{P}_k^n$.
**Require:** $\mathcal{R}$: Dictionary corresponding to the non-trivial inclusion basis. If $U \subset^* \{i\}$ then $U \in \mathcal{R}[i]$
**Ensure:** $\mathcal{P}_j^{n+1}$: Refined list of spaces consistent across all operators. ($\mathcal{P}_j^{n\prime}$ with the notation of the subsection I.3.3)
1: $\mathcal{P}_j^{t+1} \leftarrow \mathcal{P}_j^t$
2: **for** each $a$ in $\mathcal{P}_j^t$ **do**
3:      **for** each $i$ in $a$ **do**
4:          **if Do_$i$_covers_$\mathcal{P}_j^n$**$(i, \mathcal{R}, p_j^n)$ **then**
5:              **break**
6:          **end if**
7:          **for** each $p_k^n$ in $(p_k^n)_{k \neq j}$ **do**
8:              **if** $i \notin p_k^n$ **then**
9:                  Remove $a$ from $\mathcal{S}_j^{n+1}$
10:               **break**
11:              **end if**
12:          **end for**
13:      **end for**
14: **end for**
15: **return** $\mathcal{S}_j^{n+1}$

---

**Algorithm 3** Do $i$ covers $\mathcal{P}_j^n$

---

**Require:** $i$: An integer
**Require:** $\mathcal{R}$: Non-trivial inclusion basis.
**Require:** $p_j^n$: Set of elements of $\mathcal{P}(\{1, \ldots, K\})$, it is the set of elements that appear in $\mathcal{P}_j^n$.
**Ensure:** $covers$: Boolean indicating whether $i$ covers $\mathcal{P}_j^n$
1: $covers \leftarrow$ False
2: **if** $i$ is a key of $\mathcal{R}$ **then**
3:      **for** $C$ in $\mathcal{R}[i]$ **do**
4:          **if** $C \subset p_j^n$ **then**
5:              $covers \leftarrow$ True
6:              **Break**
7:          **end if**
8:      **end for**
9: **end if**
10: **return** $covers$

---

## I.3 PROOF OF THEOREM 4.2

### I.3.1 MATHEMATICAL DEFINITION OF THE PROBLEM

In the framework introduced by Koole (2007), structural properties of the optimal value function and policy can be derived from the propagation behavior of the operators forming the operator graph representation of the Bellman equation.

Formally, consider a graph of operators $T_1, \ldots, T_k$, where $n$ is the dimension of the state space, and value functions belong to $\mathbb{R}^{\mathbb{N}^n}$. Define a base family of $K$ subspaces of $\mathbb{R}^{\mathbb{N}^n}$, denoted as $\mathcal{B} = \{B_1, \ldots, B_K\}$. In this framework, these base subspaces are generated systematically, with details provided in ??.

From these base subspaces, we define the set of function spaces for which propagation results can be derived as:
$$\mathcal{S} = \text{span}\,\mathcal{B} = \Big\{ \bigcap_{i \in Q} B_i \mid Q \in \mathcal{P}(\{1, \ldots, K\}) \Big\},$$

where $\mathcal{P}(\{1, \ldots, K\})$ is the power set of $\{1, \ldots, K\}$. We equip $\mathcal{S}$ with a non-trivial $\subset$ relationship. We discuss more precisely what it means bellow.

For each operator $T_j$, we can identify a set of propagated spaces $\mathcal{S}_j$, which is a subset of $\mathcal{S}$. The objective is to determine the smallest common space under $\subset$ across all $\mathcal{S}_j$:

$$\mathcal{F} = \min \bigcap_{j=1}^{N} \operatorname{span} \mathcal{S}_j.$$

$\mathcal{F}$ is indeed the smallest element of $\mathcal{S}$ that is propagated through all the operators and therefore through the overall graph. From Theorem 4.1, we can conclude that $V^* \in \mathcal{F}$ and derive structural results for the optimal policies.

**Definition of the non-trivial $\subset$ relationship.**

The ordering relationship $\subset$ must be defined according to the following rules:

- **Base (trivial relationships):** The relationship starts from a set of trivial inclusion relationships, such as $A \cap B \subset B$.

- **Generation rule for non-trivial relationships:** We add a finite family of *non-trivial* inclusions of the form $A \subset \{i\}$. From these, all other needed inclusions must be generated by the following rule alone (without invoking the general transitivity rule to create new ones):
  $$\text{If } A \subset B \text{ and } C \subset D, \text{ then } A \cap C \subset B \cap D.$$

  This rule should be sufficient to produce a consistent ordering. We refer to this finite set of non-trivial inclusions as the *non-trivial inclusion basis*. A similar constraint hold for equalities, in particular we should not have to use the rule $(A \subset B) \wedge (B \subset A) \implies A = B$ and only the following ones :

  1. **Intersection** If $A = B$ and $C = D$, then $(A \cap C) = (B \cap D)$.
  2. **Augmentation:** If $A = B$ and $A \subset C$, then $(A \cap C) = B$.

$\subset$ is fully defined by the non-trivial inclusion basis.

The idea behind this constraint is that we can check if any inequality is verified in a constant time if we have the non-trivial inclusion basis and non-trivial equalities have a common form. It is used in lemma I.2 and in lemma I.3. We show that these rules are verified in our framework in section E.3.

### I.3.2 REFORMULATION OF THE PROBLEM

We reformulate the problem to better align it with an algorithmic approach:

- Replace $\mathcal{B}$ with $\mathcal{I} = \{1, \ldots, K\}$. In other words, each space is identified by an index.

- Replace $\mathcal{S}$ with $\mathcal{P}(\mathcal{I}) = \mathcal{P}(\{1, \ldots, K\})$.

- Replace $\subset$ with an ordering relationship $\subset^*$ over $\mathcal{P}(\mathcal{I})$. This extends the canonical inclusion relationship on $\mathcal{P}(\mathcal{I})$. For example, $\bigcap_{u \in U} B_u \subset \bigcap_{v \in V} B_v$ becomes $V \subset^* U$.

- The non canonical $\subset^*$ implies a non canonical equality relationship $=^*$ defined as : if $U \subset^* V$ and $V \subset^* U$ then $U =^* V$. Notably we can now have equalities that are non trivial, such as $(1, 2, 3) =^* (2, 3)$.

- For each relationship in the *non-trivial inclusion basis* $\{i\} \subset^* U$, we introduce a tuple representation $r = (i, U)$. Denote $\mathcal{R}$ as the set of all such inclusion relationships.

- Replace $\mathcal{S}_j$ with the corresponding $\mathcal{P}_j \subset \mathcal{P}(\mathcal{I})$, such that $\mathcal{S}_j = \{\bigcap_{i \in Q} B_i \mid Q \in \mathcal{P}_j\}$.

The closure under intersection, span $\mathcal{S}_j$, is equivalent to span $\mathcal{P}_j$, which is defined as the closure under two operations: *union* and the generation of new sets using the extended ordering relationship $\subset^*$. In other words if $U \cup W \in \operatorname{span} \mathcal{P}_j$ and $V \subset^* U$, then the set $V \cup U \cup W$ is also included in span $\mathcal{P}_j$.

The problem now becomes finding the biggest common element for $\subset^*$ across all span $\mathcal{P}_j$ :

$$\mathcal{I}_{\mathcal{F}} =^* \max\left(\bigcap_{j=1}^{N} \text{span } \mathcal{P}_j.\right)$$

Here, the intersection is taken with respect to the $=^*$ relationship.

This reformulation is advantageous for algorithmic purposes because the implicit relationships among functional spaces, captured by $\subset^*$ and $=^*$ in $\mathcal{P}(\mathcal{I})$, are made explicit through the set $\mathcal{R}$.

### I.3.3 SOME NOTATIONS

Below we introduce several pieces of notation that will be used in our construction. Throughout, let $\mathcal{A}_1, \ldots, \mathcal{A}_N \subset \mathcal{P}(\mathcal{I})$, and let $i$ be any index in $\mathcal{I}$.

- We say that $i$ *appears in each family* $(\mathcal{A}_1, \ldots, \mathcal{A}_N)$ if, for every $j \in \{1, \ldots, N\}$, there exists at least one set $a \in \mathcal{A}_j$ such that $i \in a$. This notion naturally extends to a subset $I \subseteq \mathcal{I}$: we say that $I$ *appears in each family* if all $i \in I$ *appear in each family* according to the above definition. If we only consider one family $\mathcal{A}$, we say that $i$ *appears* in $\mathcal{A}$.

- For a particular $\mathcal{A}_j$, we say $i$ *covers* $\mathcal{A}_j$ if there exists $(i, U) \in \mathcal{R}$ such that $U$ appears in $\mathcal{A}_j$.

- We write $i \ \lhd \ \big(j, (\mathcal{A}_1, \ldots, \mathcal{A}_N)\big)$ if either $i$ appears in each family $(\mathcal{A}_1, \ldots, \mathcal{A}_N)$, *or* $i$ covers $\mathcal{A}_j$.

Finally, given $\mathcal{A}_1, \ldots, \mathcal{A}_J \subset \mathcal{P}(\mathcal{I})$, for each $j \in \{1, \ldots, J\}$ define

$$\mathcal{A}'_j \ = \ \Big\{ a \in \mathcal{A}_j \ \Big| \ \forall i \in a : i \ \lhd \ \big(j, (\mathcal{A}_1, \ldots, \mathcal{A}_J)\big) \Big\}.$$

We then define the function

$$F\big(\mathcal{A}_1, \ldots, \mathcal{A}_J\big) \ = \ \big(\mathcal{A}'_1, \ldots, \mathcal{A}'_J\big).$$

### I.3.4 THE ALGORITHM

A straightforward yet naive method would be to construct the set span $\mathcal{P}_j$ for each $j$ by exhaustively applying every closure rule, then intersect these sets at the end. However, due to the non-trivial inclusion relationships, this expansion can be both difficult to implement and prohibitively large in memory usage—scaling exponentially with the size of each $\mathcal{P}_j$.

Instead, we propose a more efficient procedure that avoids this blowup. We form a sequence of tuples

$$\{P^n\}_{n=0}^{\infty} \quad \text{with} \quad P^0 = (\mathcal{P}_1, \ldots, \mathcal{P}_J) \quad \text{and} \quad P^{n+1} = F(P^n).$$

It will be shown below that this sequence converges to a stationary limit

$$P^{\infty} = (\mathcal{P}_1^{\infty}, \ldots, \mathcal{P}_J^{\infty}).$$

We then define

$$p^{\infty} \ = \ \bigcup_{p \in \mathcal{P}_1^{\infty}} p$$

and prove that for each $j$,

$$p^{\infty} \ =^* \bigcup_{p \in \mathcal{P}_j^{\infty}} p, \quad \text{and} \quad p^{\infty} \ =^* \mathcal{I}_{\mathcal{F}} \ =^* \max\Big(\bigcap_{j=1}^{N} \text{span } \mathcal{P}_j\Big).$$

In other words, the family of subspaces $\{B_k \mid k \in p^{\infty}\}$ constitutes the smallest space that is propagated through the entire operator graph.

### I.3.5 PROOF OF CONVERGENCE

**Fixed points correspond to common propagated spaces.**

**Lemma I.1.** *Let $P = (P_1, \ldots, P_J)$ be a tuple of subsets in $\mathcal{P}(\mathcal{I})$ such that the corresponding function spaces of each $P_j$ are propagated by operators $T_j$. Define $p_j = \bigcup_{p \in P_j} p$ for each $j$. If $P$ is a fixed point of $F$, then $p_i =^* p_j$ for all $i, j$, and the set $\mathcal{F} = \{B_i \mid i \in p_1\}$ is a common propagated space across the operators.*

*Proof.* Since $P$ is a fixed point of $F$, every index $i \in p_1$ must satisfy

$$i \vartriangleleft (1, (P_1, \ldots, P_J)).$$

There are two ways this can happen:

1. $i$ **appears in each family** $(P_1, \ldots, P_J)$**.** In this case, $i$ belongs to $p_j$ for every $j$.

2. $i$ **covers** $P_1$**.** Here, there exists $(i, U) \in \mathcal{R}$ such that $U \subset p_1$. By definition, $\{i\} \subset^* U$. Consequently, $p_1 \setminus \{i\} =^* p_1$.

Combining these observations, define

$$p_1' = \{ i \in p_1 \mid i \text{ appears in each family } (P_1, \ldots, P_J)\}.$$

From the cases above, we see $p_1' =^* p_1$, and in fact $p_1' =^* p_j'$ for all $j$. Hence,

$$p_1 =^* p_2 =^* \cdots =^* p_J.$$

Finally, let $\mathcal{F}_j = \{B_i \mid i \in p_j\}$. Since each $\mathcal{F}_j$ is propagated by $T_j$ and $\mathcal{F}_j =^* \mathcal{F}_i$ for all $i, j$, we conclude that $\mathcal{F} = \{B_i \mid i \in p_1\}$ is indeed a single common propagated space for all the operators. $\square$

**Equalities under $=^*$ share a useful structure.**

**Lemma I.2.** *If $U =^* V$, then $U$ and $V$ can be decomposed as $U = U_1 \cup U_2$ and $V = V_1 \cup V_2$, where $U_1 = V_1$, $U =^* U_1$, and $V =^* V_1$.*

*Proof.* In our setting, and due to the specialized nature of the $\subset^*$ relation, *every* non-trivial equality under $=^*$ can be derived from a collection of trivial equalities by repeatedly applying two fundamental rules:

1. **Union** If $A =^* B$ and $C =^* D$, then $(A \cup C) =^* (B \cup D)$.

2. **Augmentation:** If $A =^* B$ and $A \subset^* C$, then $(A \cup C) =^* B$.

If the equalities used in these steps already satisfy the decomposition property of the lemma, then the newly derived equality also satisfies it. Since all *trivial* equalities fulfill this property at the outset, an induction argument ensures that *every* equality produced in this manner will do so as well. $\square$

**Common propagated spaces have a corresponding fixed point.**

**Lemma I.3.** *Let's consider a common propagated space $\mathcal{F}$ across the operators $T_j$. There exists a fixed point $P = (P_1, \ldots, P_J)$ of $F$ which corresponding propagated space (lemma I.1) is $\mathcal{F}$ and such that $P_j \subset \mathcal{P}_j$.*

*Proof.* Let's consider a common propagated space $\mathcal{F}$ and a corresponding representation $p \in \bigcap_{j=1}^{N} \text{span} \, \mathcal{P}_j$. Lets $p_j$ the representation of $p$ in each span $\mathcal{P}_j$. Thanks to the previous lemma I.2 we can write $p_j = p_j^1 \cup p_j^2$ for all $j$ such that $p_1^1 = p_2^1 = \cdots = p_J^1$. Now let $j$, and take $i \in p_j$. There are 2 possibilities :

- $i \in p_j^1$ and therefore $i$ appears in each family $(\mathcal{P}_1, \ldots, \mathcal{P}_J)$.

- $i \in p_j^2$ and therefore $\{i\} \subset^* p_j^1$. And with arguments similar as in the lemma I.2 (relying on the specialized nature of $\subset^*$) we can say that $i$ covers $\mathcal{P}_j$.

In other words $i \lhd \{j, (\mathcal{P}_1, \ldots, \mathcal{P}_J)\}$. We write each $p_j$ as an union of elements of $\mathcal{P}_j$ and define $\mathcal{P}'_j$ the set of these elements. From the previous result we can conclude that $P' = (\mathcal{P}'_1, \ldots, \mathcal{P}'_J)$ is a fixed point of $F$. This conclude the proof. $\qquad\square$

**Theorem I.4.** *Let $\{P^n\}_{n=0}^{\infty}$ be the sequence defined in the algorithm, where each $P^n = (\mathcal{P}_1^n, \ldots, \mathcal{P}_J^n)$. This sequence converges to a fixed point, and the corresponding function space is the smallest common space propagated by all operators.*

*Proof.* We note $P^n = (\mathcal{P}_1^n, \ldots, \mathcal{P}_J^n)$. For each $j$ $\{\mathcal{P}_j^n\}_{n=0}^{\infty}$ is a decreasing sequence of subsets of the finite set $\mathcal{P}(\mathcal{I})$. Each one of them is then stationary after a certain point, therefore $\{P^n\}_{n=0}^{\infty}$ is itself stationary after a certain point.

We can now let $P^{\infty}$ be the limit of this sequence. By the previous lemma I.1 we can define $p_{\infty}$ and $\mathcal{F}_{\infty} = \{B_i \mid i \in p_{\infty}\}$ to be the corresponding common propagated space.

Suppose there is another common propagated space $\mathcal{F}$. By Lemma I.3, there exists a fixed point $P = (P_1, \ldots, P_J)$ corresponding to $\mathcal{F}$. A routine induction shows that any set $a$ removed from $\mathcal{P}_j^n$ at some step of the algorithm can not appear in $P_j$. Therefore $\bigcup_{p \in P_j} \subset p_j^{\infty}$ and $\mathcal{F}_{\infty} \subset \mathcal{F}$. Hence, $\mathcal{F}_{\infty}$ is the smallest among all common propagated spaces. $\qquad\square$

### I.3.6 COMPLEXITY

Applying $F$ to each $\mathcal{P}_j$ requires $\mathcal{O}(NJ^2)$ time per iteration, where $N = \max_j \sum_{p \in \mathcal{P}_j} \#p$ denotes the maximum total number of spaces across all propagated sets for any operator. In practice, the sequence $\{P^n\}$ typically converges in only a few iterations, ensuring that the overall runtime remains tractable. Additionally, the memory complexity is $\mathcal{O}(NJ)$, representing a significant improvement over the naive approach, which is exponential in both time and space.

This gain is already meaningful for small-scale problems. For instance, in a state space of dimension $n$, the size of the base set $\mathcal{B}$, and hence the quantity $N$, typically scales as $n^2$. When $n = 4$, the base set already contains $\#\mathcal{B} = 53$ elements.

### I.3.7 A RUNNING EXAMPLE

In this section, we illustrate the algorithm using a running example. Consider the following setting:

$$\begin{cases} T_1 \text{ propagates } F \cap A, \ E \cap A, \ C, \ D \cap F, \ H, \\ T_2 \text{ propagates } H \cap C, \ D \cap F, \ A \cap B, \\ T_3 \text{ propagates } F \cap G, \ C \cap B \cap I, \ A \cap D, \ H, \\ D \cap C \subset E, \\ F \subset B, \\ A \subset G. \end{cases}$$

Using the notation from the formal proof, we start with the initiation ($n = 0$):

$$\begin{cases} \mathcal{P}_1^0 = \{F \cap A, \ E \cap A, \ C, \ D \cap F, \ H\}, \\ \mathcal{P}_2^0 = \{H \cap C, \ D \cap F, \ A \cap B\}, \\ \mathcal{P}_3^0 = \{F \cap G, \ C \cap B \cap I, \ A \cap D, \ H\}, \end{cases}$$

$$\begin{cases} p_1^0 = \{A, C, D, E, F, H\}, \\ p_2^0 = \{A, B, C, D, F, H\}, \\ p_3^0 = \{A, B, C, D, F, G, H, I\}. \end{cases}$$

**First Iteration ($n = 1$).** We now evaluate which propagated sets are preserved across all operators.

**For $T_1$:**

- $F \cap A$: $F, A \in p_2^0 \cap p_3^0 \Rightarrow$ keep.

- $C$: $C \in p_2^0 \cap p_3^0 \Rightarrow$ keep.
- $E \cap A$: $C \in p_2^0 \cap p_3^0$. $D, C \in p_1^0$ and $D \cap C \subset E \Rightarrow$ keep.
- $D \cap F$: $D, F \in p_2^0 \cap p_3^0 \Rightarrow$ keep.

Thus,

$$\mathcal{P}_1^1 = \{F \cap A,\ E \cap A,\ C,\ D \cap F,\ H\}.$$

**For $T_2$:**

- $H \cap C$: $H, C \in p_1^0 \cap p_3^0 \Rightarrow$ keep.
- $D \cap F$: $D, F \in p_1^0 \cap p_3^0 \Rightarrow$ keep.
- $A \cap B$: $A \in p_1^0 \cap p_3^0$, $F \subset B$ and $F \in p_2^0 \Rightarrow$ keep.

Hence,

$$\mathcal{P}_2^1 = \{H \cap C,\ D \cap F,\ A \cap B\}.$$

**For $T_3$:**

- $F \cap G$: $F \in p_1^0 \cap p_2^0$, $A \subset G$ and $A \in p_3^0 \Rightarrow$ keep.
- $C \cap B \cap I$: $I \notin p_1^0 \cap p_2^0$ and not involved in any relation $\Rightarrow$ discard.
- $A \cap D$: $A, D \in p_1^0 \cap p_2^0 \Rightarrow$ keep.
- $H$: $H \in p_1^0 \cap p_2^0 \Rightarrow$ keep.

Therefore,

$$\mathcal{P}_3^1 = \{F \cap G,\ A \cap D,\ H\}.$$

We now have :

$$\begin{cases} p_1^1 = \{A, C, D, E, F, H\}, \\ p_2^1 = \{A, B, C, D, F, H\}, \\ p_3^1 = \{A, D, F, G, H\}. \end{cases}$$

**Second Iteration ($n = 2$).** We now report only the spaces that are discarded in this iteration.

**For $T_1$:**

- $C$: $C \notin p_3^1 \Rightarrow$ discard.

$$\mathcal{P}_1^2 = \{F \cap A,\ E \cap A,\ D \cap F,\ H\}.$$

**For $T_2$:**

- $H \cap C$: $C \notin p_3^1 \Rightarrow$ discard.

$$\mathcal{P}_2^2 = \{D \cap F,\ A \cap B\}.$$

**For $T_3$:** We keep all the spaces, therefore :

$$\mathcal{P}_3^2 = \{F \cap G,\ A \cap D,\ H\}.$$

Update:

$$\begin{cases} p_1^2 = \{A, D, E, F, H\}, \\ p_2^2 = \{A, B, D, F\}, \\ p_3^2 = \{A, D, F, G, H\}. \end{cases}$$

**Third Iteration** ($n = 3$). **For $T_1$:**

- $E \cap A$: Previously kept due to $D \cap C \subset E$, but now $C \notin p_1^2 \Rightarrow$ discard.
- $H$: $H \notin p_2^2 \Rightarrow$ discard.

$$\mathcal{P}_1^3 = \{F \cap A, \ D \cap F\}.$$

**For $T_2$:** We keep all the spaces, hence :

$$\mathcal{P}_2^3 = \{D \cap F, \ A \cap B\}.$$

**For $T_3$:**

- $H$: $H \notin p_2^2 \Rightarrow$ discard.

$$\mathcal{P}_3^3 = \{F \cap G, \ A \cap D\}.$$

Update:

$$\begin{cases} p_1^3 = \{A, D, F\}, \\ p_2^3 = \{A, B, D, F\}, \\ p_3^3 = \{A, D, F, G\}. \end{cases}$$

**Fourth Iteration** ($n = 4$). At this stage, the propagated sets remain unchanged, indicating convergence. Thus, the largest common propagated space is:

$$A \cap B \cap D \cap F \cap G = A \cap D \cap F.$$

## J    DETAILED DESCRIPTIONS OF EXAMPLES IN SECTION 5

In this sections we give more details on the examples discussed in Section 5.

**Example 3** (The MCTS succeeds to generate formulation with high structural expressiveness in complex problems)**.** Let us consider the following problem :

*We aim to minimize the long-run average cost of operating our hospital. The hospital has 3 wards arranged sequentially, sharing a total capacity of 20 beds. Each ward has its own healthcare team and manages its own patients. On average, 7 patients arrive at the first ward per hour. The wards serve patients at average rates of 10, 5, and 2 patients/hour, respectively. Patients progress sequentially: from the first ward to the second, and then from the second to the third. After treatment in the third ward, patients leave the hospital. Incoming patients can be rejected, incurring a cost of 20. Additionally, patients can be directly transferred from one ward to the next before being served at an average rate of 3 patients/hour for each ward, with each transfer costing 5.*

**Key challenges:** There are various types of events (controlled arrivals, departures, transfers) and implicit constraints (nonnegativity) on the state space.

The first level of the tree consists of defining the parameters of the problem, while the second level identifies the state space. This gives:

$$\begin{cases} n_{\text{beds}} = 20, \\ r_{\text{arrivals}} = 7, \\ r_{\text{service}} = (10, 5, 2), \\ r_{\text{transfer opportunity}} = 3, \\ c_{\text{refusal}} = 20, \\ c_{\text{transfer}} = 5, \end{cases} \quad \begin{cases} x_1 + x_2 + x_3 \leq 20, \\ x_1, x_2, x_3 \geq 0. \end{cases}$$

Here, $x = (x_1, x_2, x_3)$ represents the number of patients in the respective wards. Notably the positive constraints were implicit.

The next step defines the events, their probabilities, and the available actions with their corresponding costs and effects on the state. Let $\Gamma = r_{\text{arrivals}} + r_{\text{service},1} + r_{\text{service},2} + r_{\text{service},3} + 3r_{\text{transfer opportunity}}$ and $(\varepsilon_i)_{1 \leq i \leq 3}$ the canonical base of $\mathbb{R}$. The events are as follows:

- **Patient arrival :**
$$\begin{cases} p_{\text{arrival}} = r_{\text{arrivals}}/\Gamma, \\ \mathcal{A}_{\text{arrival}} = \{a_{\text{accept}}, a_{\text{refuse}}\}, \\ P_{\text{arrival}}(x' \mid x, a_{\text{accept}}) = \mathbb{1}(x' = x + \varepsilon_1), \\ P_{\text{arrival}}(x' \mid x, a_{\text{refuse}}) = \mathbb{1}(x' = x), \\ c_{\text{arrival}}(x', x, a) = c_{\text{refusal}}\mathbb{1}(a = a_{\text{refuse}}). \end{cases}$$

- **Patient served ward 1 :**
$$\begin{cases} p_{\text{service},1} = r_{\text{service},1}/\Gamma, \\ \mathcal{A}_{\text{service},1} = \varnothing, \\ P_{\text{service},1}(x'|x) = \begin{cases} \mathbb{1}(x' = x - \varepsilon_1 + \varepsilon_2), & \text{if } x_1 > 0, \\ \mathbb{1}(x' = x), & \text{otherwise,} \end{cases} \\ c_{\text{service},1}(x', x) = 0. \end{cases}$$

- **Patient transferred ward 1 to ward 2:**
$$\begin{cases} p_{\text{transfer},1} = r_{\text{transfer opportunity}}/\Gamma, \\ \mathcal{A}_{\text{transfer},1} = \{a_{\text{transfer}}, a_{\text{keep}}\}, \\ P_{\text{transfer},1}(x' \mid x, a_{\text{transfer}}) = \mathbb{1}(x' = x - e_1 + e_2) \\ P_{\text{transfer},1}(x' \mid x, a_{\text{keep}}) = \mathbb{1}(x' = x) \\ c_{\text{transfer},1}(x', x, a) = c_{\text{transfer}}\mathbb{1}(a = a_{\text{transfer}}). \end{cases}$$

We do not detail the system dynamics for the remaining events: "Patient served in ward 2", "Patient served in ward 3", and "Patient transferred from ward 2 to ward 3," as they are similar to the examples illustrated above.

The next step is to identify the operational cost, which in this problem is equal to 0 (Maybe I should add one).

Using the dynamics of the system for each event, we can already derive the corresponding operators. In the last layer of the tree the LLM identifies the operators for which propagation results apply. For this problem, the identified operators are:

- **Patient arrival :**

$$T_{\text{CA},1}f(x) = \min\{c_{\text{refusal}} + f(x); f(x + e_1)\}$$

- **Patient served ward 1 :**

$$T_{\text{TD1},1}f(x) = \begin{cases} f(x - e_1 + e_2), & \text{if } x_1 > 0, \\ f(x), & \text{otherwise.} \end{cases}$$

- **Patient transferred ward 1 to ward 2 :**

$$T_{\text{CTD},(1,2)}f(x) = \min\{c_{\text{transfer}} + f(x - e_1 + e_2), f(x)\}$$

- **Patient served ward 3 :**

$$T_{\text{D1},3}f(x) = f((x - e_3)^+)$$

Similar operators can be derived for "Patient served in ward 2" and "Patient transferred from ward 2 to ward 3."

For each of these operators we can automatically list the functional spaces they propagate, for instance $T_{\text{CA},1}$ propagate all the following spaces (see Appendix for the details of each one of them) :

$$I, \text{ UI}, Cx(1), \text{ Super}(1,2), \text{ Super}(1,3), \text{ Sub},$$
$$\text{Super}(1,2) \cap \text{SuperC}(1,2), \text{ Super}(1,3) \cap \text{SuperC}(1,3),$$
$$\text{Super}(1,3) \cap \text{SuperC}(3,1), \text{ Super}(1,2) \cap \text{SuperC}(2,1),$$
$$\text{Sub}(1,2) \cap \text{SubC}(1,2) \text{ Sub}(1,3) \cap \text{SubC}(1,3),$$
$$\text{Sub}(1,2) \cap \text{SubC}(2,1), \text{ Sub}(1,3) \cap \text{SubC}(3,1), \text{ MM}$$

We also do this for $T_{\text{unif}}$ and $T_{\text{cost}}$. Also, depending on the shape of the state space certain spaces must be dropped. Finally, we can run our second algorithm introduced in the subsection 4.3. For this problem we end up with the following propagated space :

$$I \cap \text{UI} \cap \text{MM}$$

From which we can extract automatically the following structural results :

1. **Controlled arrival in ward 1 :** let $\pi^*_{\text{CA}(1)} : \mathcal{S} \to \{0, 1\}$ be the optimal acceptance policy in the first ward such that 0 is refusal and 1 acceptance.

    - $\pi^*_{\text{CA}(1)}$ is decreasing in the number of patients in the hospital.
    - $\pi^*_{\text{CA}(1)}$ is decreasing in the directions $(1, -1, 0)$ and $(1, 0, -1)$.

2. **Controlled departure from ward 1 to ward 2 :**, let $\pi^*_{\text{CTD}(1,2)} : \mathcal{S} \to \{0, 1\}$ be the optimal departure policy such that 0 correspond to keeping the patient in the ward 1 and 1 is moving them to ward 2.

    - $\pi^*_{\text{CTD}(1,2)}$ is decreasing in the number of patient in the ward 2, ie in the direction $(0, 1, 0)$.
    - $\pi^*_{\text{CTD}(1,2)}$ is increasing in the number of patient in the ward 1, ie in the direction $(1, 0, 0)$.

3. **Controlled departure from ward 2 to ward 3 :** let $\pi^*_{\text{CTD}(2,3)} : \mathcal{S} \to \{0, 1\}$ be the optimal departure policy such that 0 correspond to keeping the patient in the ward 2 and 1 is moving them to ward 3.

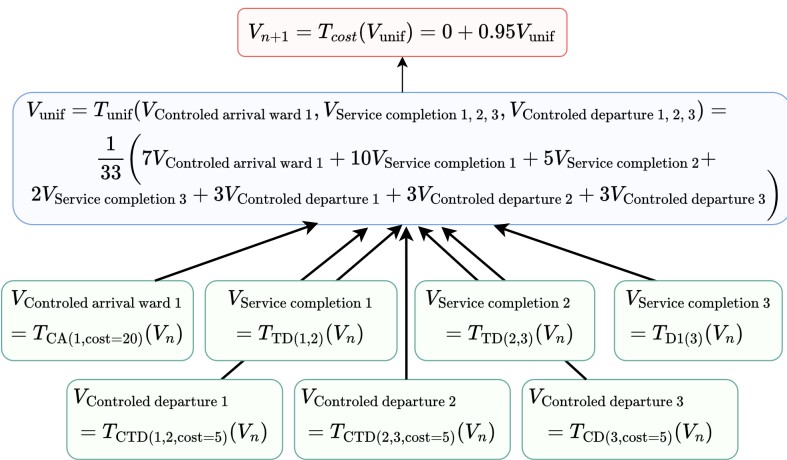

Figure 7: Operator graph of Example 3

- $\pi^*_{\text{CTD}(2,3)}$ is decreasing in the number of patient in the ward 3, ie in the direction $(0, 0, 1)$.
- $\pi^*_{\text{CTD}(2,3)}$ is increasing in the number of patient in the ward 2, ie in the direction $(0, 1, 0)$.

In other words, the optimal policy of the problem is threshold along many different directions. These structural results and the optimal policy obtained by running a solver on the formulation are then communicated back to the user.

**Key takeaways:** Autoformulating an event-based MDP involves multiple steps, and our proposed algorithm effectively navigates these challenges in complex problems. Most of the time the resulting formulation has high structural expressiveness

**Example 4.** ▶ **Two correct graphs of operators with different structural complexity**

Let us consider the following problem :

*We aim to minimize the long-run average cost of operating our hospital. The hospital has 1 ward that manages 2 types of patients with **shared** healthcare teams. There are $N_b$ beds in total. The average arrival rates of the patients are $\lambda_1$/hour and $\lambda_2$/hour respectively. A team take care of a patient with an average rate that depends on their type : $\mu_1$/hour and $\mu_2$/hour respectively. When a patient arrive we can refuse it, it occurs a cost of $c_1$ for the first type of patients and $c_2$ of the others.*

**Key challenges:** We cannot obtain structural results from the straightforward problem formulation. How to find an equivalent combination of operators that allow us to obtain structural results?

**Straightforward problem formulation.** The natural events of this problem are *Arrival of a patient of type 1*, *Arrival of a patient of type 2*, *Departure of a patient of type 1* and *Departure of a patient of type 2*.

This approach lead to the following operator graph :

$$V^*_{n+1} = T_{\text{cost}} \left\{ \left( T_{\text{unif}} \left[ T_{\text{CA}(1)}(V^*_n), T_{\text{CA}(2)}(V^*_n), T_{\text{D1}(1)}(V^*_n), T_{\text{D1}(2)}(V^*_n), V^*_n \right] \right) \right\}$$

With probabilities in $T_{\text{unif}}$ that depends on the state, you get for instance :

$$p_{\text{D}(1)} = \frac{\mu_1 n_1}{\lambda_1 + \lambda_2 + \mu N_b} \quad \text{with } \mu = \max(\mu_1, \mu_2).$$

Koole's results don't extend to probabilities that depend on the state in $T_{\text{unif}}$. We can't get any structural result from this formulation, even if it is a right one.

However, this formulation is actually equivalent to the following one:

$$V^*_{n+1} = T_{\text{cost}} \left\{ T_{\text{unif}} \left[ T_{\text{CA}(1)}(V^*_n), T_{\text{CA}(2)}(V^*_n), T_{\text{D}(1)}(V^*_n), T_{\text{D}(2)}(V^*_n) \right] \right\}.$$

This time the probabilities in $T_{\text{unif}}$ are (with $\Gamma = \lambda_1 + \lambda_2 + \mu N_b$) :

1. $p_{\text{CA}(i)} = \lambda_i / \Gamma$

2. $p_{\text{D}(1)} = p_{\text{D}(2)} = \frac{1}{2}\left(1 - \frac{\lambda_1 + \lambda_2}{\Gamma}\right)$

which don't depend on the state. The dependence in the state has been absorbed by the new $T_D$ operators for which we have structural results :

$$T_{\text{D}(1)}f(x) = \frac{2\mu_1 n_1}{\Gamma - (\lambda_1 + \lambda_2)}f((x - e_1)^+) + \left(1 - \frac{2\mu_1 n_1}{\Gamma - (\lambda_1 + \lambda_2)}\right)f(x)$$

Indeed, with this formulation we can show that the following space is propagated through the Bellman equation :

$$I \cap \text{Cx} \cap \text{Super}$$

And we can deduce structural results from there.

**Key Takeaways :** Problem formulation and structural analysis are inherently connected, as certain valid formulations may not permit structural analysis.

**Example 5. ▶ We don't have any formulation that reveals the structural results.**

Let us consider the following problem :

*We aim to minimize the long-run average cost of operating our hospital. The hospital has 3 wards arranged sequentially, with capacities of 5, 15, and 15 beds, respectively. Each ward has its own healthcare team and manages its own patients. On average, new patients arrive at rates of 3, 20, and 5 patients/day in the respective wards. The wards serve patients at rates of 10, 5, and 3 patients/day, respectively. After being served in the first or second ward, we can transfer to the next ward at a cost of 2 per transfer or keep them in the current ward. Patients served in the third ward leave the hospital. Additionally, **patients can be moved back** from ward 2 to ward 1 at a rate of 3 patients/day or from ward 3 to ward 2 at a rate of 1 patient/day, each transfer incurring a cost of 2. Incoming patients can also be refused, incurring costs of 5, 10, and 15 for wards 1, 2, and 3, respectively.*

**Key challenges:** The solution to the problem exhibits structural properties, but these cannot be anticipated regardless of the choice of operator graph.

The autoformulation part of the algorithm managed to find a correct operator graph :

$$\begin{aligned}
V_{n+1}^* = T_{\text{cost}}\big(&T_{\text{unif}}\big(T_{\text{CA},1}(V_n^*), T_{\text{CA},2}(V_n^*), T_{\text{CA},3}(V_n^*), \\
&T_{\text{CTD},(1,2)}(V_n^*), T_{\text{CTD},(2,3)}(V_n^*), T_{\text{CTD},(2,1)}(V_n^*), \\
&T_{\text{CTD},(3,2)}(V_n^*), T_{\text{DI},3}(V_n^*)\big)\big)
\end{aligned}$$

The structural results observed experimentally (see Figure 8) cannot be predicted from the operator graph. Unlike Example 2, this issue cannot be resolved with a better formulation, as the structural results have not yet been theoretically established.

**Key takeaways:** The current method for identifying structural results fails on certain problems, as it depends on limited theoretical results.

**Example 6** (Two wards with controlled jockeying). Natural language description :

*We aim to minimize the long-run average cost of operating our hospital. The hospital has two wards running in parallel, each managing its own patients with a dedicated healthcare team. The first ward can hold up to 5 patients, while the second can accommodate up to 10. Patients complete their treatment in the second ward before leaving the hospital. Each patient in the hospital cost 2/hour to the hospital. On average, 3 patients arrive at the first ward per hour, and 5 arrive at the second. New patients can be rejected, incurring a cost of 5 for the first ward and 10 for the second. The first ward operates at a frequency of 10 patients/hour, while the second operates at 5 patients/hour. Patients treated in the first ward can either remain there at no cost (but will require further care by the same team before leaving the ward) or be transferred to the second ward at a cost of 2. Additionally, patients can be transferred back from the second ward to the first at a frequency of 3 patients/hour, with each transfer costing 2. Refusing a transfer incurs no cost.*

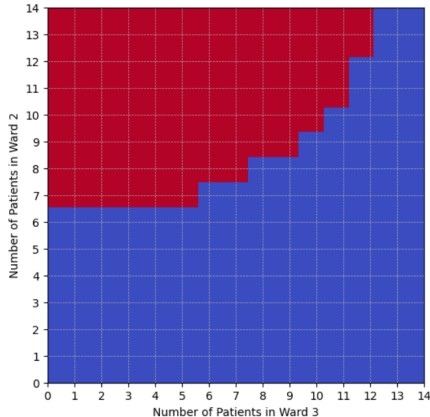

Figure 8: Optimal policy for a controlled jockeying problem across three wards with controlled arrival and unControlled departures in the last ward. One of the event is the opportunity to move a patient from ward 2 to ward 3, the possible actions are : *move* (in red) or *keep* (in blue). The graph shows the switching curve the optimal policy depending on the number of patients in the wards. The optimal policy is structured but we can not anticipate it with the results from Koole (2007) and therefore it's beyond the capacities of our algorithm.

The operator graph of the problem is illustrated 7. The Bellman equation propagate the following function space :

$$I \cap \text{Super} \cap \text{SuperC}$$

Let $x = (n_1, n_2)$ be the number of patients in the two wards. We have the following structural results for the optimal policy:

1. **Controlled arrival in ward 1 :** let $\pi^*_{\text{CA}(1)} : \mathcal{S} \to \{0, 1\}$ be the optimal acceptance policy in the first ward such that 0 is refusal and 1 acceptance.

   - $\pi^*_{\text{CA}(1)}$ is decreasing in the number of patients in the hospital.
   - $\pi^*_{\text{CA}(1)}$ is decreasing in the direction $(1, -1)$.

2. **Controlled arrival in ward 2 :** let $\pi^*_{\text{CA}(2)} : \mathcal{S} \to \{0, 1\}$ be the optimal acceptance policy in the first ward such that 0 is refusal and 1 acceptance.

   - $\pi^*_{\text{CA}(2)}$ is decreasing in the number of patients in the hospital.
   - $\pi^*_{\text{CA}(2)}$ is decreasing in the direction $(-1, 1)$.

3. **Controlled departure from ward 1 to ward 2 :**, let $\pi^*_{\text{CTD}(1,2)} : \mathcal{S} \to \{0, 1\}$ be the optimal departure policy such that 0 correspond to keeping the patient in the ward 1 and 1 is moving them to ward 2.

   - $\pi^*_{\text{CTD}(1,2)}$ is decreasing in the number of patient in the ward 2, ie in the direction $(0, 1)$.
   - $\pi^*_{\text{CTD}(1,2)}$ is increasing in the number of patient in the ward 1, ie in the direction $(1, 0)$.

4. **Controlled departure from ward 2 to ward 1 :**, let $\pi^*_{\text{CTD}(2,1)} : \mathcal{S} \to \{0, 1\}$ be the optimal departure policy such that 0 correspond to keeping the patient in the ward 2 and 1 is moving them to ward 1.

   - $\pi^*_{\text{CTD}(2,1)}$ is decreasing in the number of patient in the ward 1, ie in the direction $(1, 0)$.
   - $\pi^*_{\text{CTD}(2,1)}$ is increasing in the number of patient in the ward 2, ie in the direction $(0, 1)$.

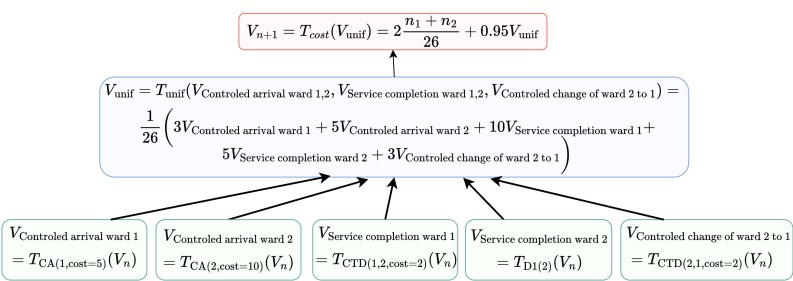

Figure 9: Operator graph of Example 5

# K DESIDERATA OF AUTOFORMULATION

## K.1 DESIDERATA AND CORRESPONDING CHALLENGES

The overarching objective of autoformulation is to autonomously solve problems expressed in natural language. This objective can be decomposed into three essential desiderata: (see Fig. 1 for how our framework fulfill them)

**Accuracy.** The ability to translate the natural language description into a suitable formal framework while preserving semantic accuracy. Autoformulation should correctly *formulate the problem*. (in the context of this paper output a MDP formulation that correctly reflects the problem description in natural language).

**Computational Tractability:** The resulting formalization must support efficient computation of a solution. For instance, autoformulation should *identify structures of the optimal policy* (e.g., the action is monotone in the state). The structures should be identified based on the formulation only, *before* the problem is solved. This facilitates in selecting low-complexity algorithms tailored for finding policies with certain structures (e.g., solvers for threshold policies).

**Interpretability:** Autoformulation should also be *interpretable* in two aspects. ▶ **Interpretability of formulation**: We should be able to trace each components of problem formulation back to the natural language problem description. ▶ **Interpretability of policies**: By identifying structural properties of the optimal policy, the autoformulator can explain the policy, making it easier for *non-technical* domain experts to understand and adopt the policy.

These desiderata come with corresponding challenges that must be addressed. We discuss them below, and use them in Section 5 to evaluate the performance of our proposed algorithm against these criteria.

**Challenges in Accuracy**
Achieving correct formalization is a non-trivial task, as it amounts to searching within a vast space of possible formulations. The main challenges include:

- *Semantic Understanding:* The system must correctly capture the underlying dynamics of the problem described in natural language. *For example, understanding that admitting a new patient to a hospital reduces the number of available beds*

- *Parameter Identification:* Relevant variables, constraints, and objective components must be identified and instantiated correctly. *For instance, the system should infer the arrival frequency of different types of patients to the hospital.*

- *Syntactic Validity:* The generated formulation must conform to the syntactic requirements of the chosen formal framework while preserving the original problem's intent. *For example, state updates should be expressed using syntactically valid expressions, such as correct Python formulas.*

**Challenges in Computational Tractability**
Fulfilling the second desideratum goes beyond achieving a correct formalization: the chosen repre-

sentation must also support efficient solving. In this work, we particularly emphasize the extraction of structural properties, which gives rise to two key challenges:

- ***Expressiveness of the Formulation:*** The formal representation must be sufficiently expressive to enable extraction of meaningful structural results.

- ***Structural Inference:*** Given a formalization, the system should be able to automatically identify structural properties that can guide or accelerate the solution process.

These two challenges are interconnected: the expressiveness of a formulation determines which structures can be extracted, while the usefulness of the formulation itself depends on the system's capacity to exploit these structures.

**Challenges in Interpretability**

For safety, usability, and insight, both the formulation and the solution should be interpretable. This is important for expert auditing and for practical deployment by non-expert users. The main challenges are:

- ***Formulation Traceability:*** Each element of the formalization should be traceable to a corresponding concept or statement in the original natural language problem description.

- ***Policy Understanding:*** The optimal policy for high-dimensional problems often behaves as a black box. Making its properties explicit enhances human understanding and trust.

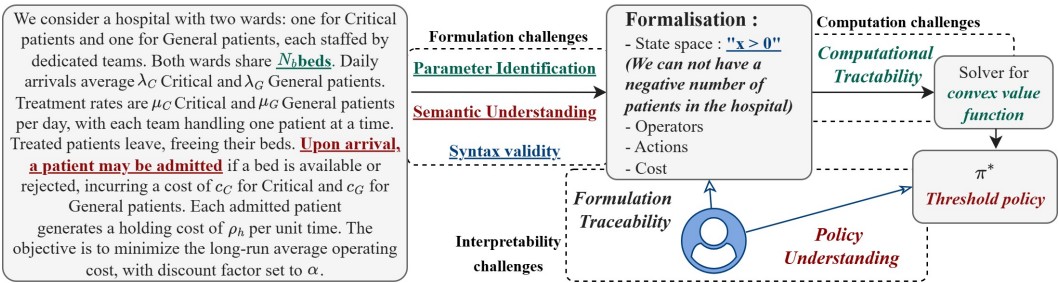

Figure 10: The challenges of autoformulation illustrated with an hospital example.

## K.2 TYPICAL ERRORS WITH RESPECT TO THESE CHALLENGES

In Section 5, we evaluate the extent to which our algorithm addresses the aforementioned challenges. Further details are provided below.

### K.2.1 ERRORS IN ACCURACY.

While our method largely resolves the *Syntactic Validity* challenge and exhibits strong performance in *Parameter Identification*, our primary focus is on *Semantic Understanding*, where most errors tend to arise. Semantic errors can occurs in several ways :

- *Missing Constraints:* The algorithm may overlook implicit constraints in the problem description, such as the non-negativity of the number of patients in a hospital.

- *Incorrect Event Modeling:* It may introduce artificial events that do not exist in the actual problem. A common example is inventing an event to account for a holding cost, modeled as an event with frequency 1 per time unit in which the state remains unchanged but a cost is incurred.

- *Failure in Uniformization:* The algorithm may miscompute event probabilities when uniformizing the process. For instance, it sometimes fails to distinguish between sequential and parallel service models. If $f$ denotes the service frequency per server:
  - In the single-server case (sequential), the probability of service is $f/\Gamma$.
  - In the multi-server case (parallel), the correct probability is $xf/\Gamma'$, where $x$ is the number of customers.

### K.2.2 ERRORS IN COMPUTATIONAL TRACTABILITY AND INTERPRETABILITY.

Our algorithm exhibits limitations in both *Structural Expressiveness* and *Structural Inference*.

- *Limited Structural Expressiveness:* In some cases, the generated formalization lacks the expressive power needed to enable structural inference. It is illustrated and discussed with Example 1.

- *Structural Inference:* Given the proven performance guarantees of our dynamic programming algorithm (see Appendix I.3), the primary remaining bottleneck in structural inference lies in the incorrect labeling of operators. For example, consider an assembly line where two elements from two queues are combined to produce an item in a third queue. The correct operator is:

$$TV(x) = V(x'_1 = [x_1 - 1, \ x_2 - 1, \ x_3 + 1])$$

  The algorithm may erroneously interpret this as a tandem departure operator:

$$T_{\text{TD}(1,3)}V(x) = V(x' = [x_1 - 1, \ x_2, \ x_3 + 1])$$

  which neglects the role of $x_2$ and leads to incorrect structural predictions.

**Limitations of the Current Framework.**
Example 2 highlights intrinsic limitations of the current framework for structural result extraction. Our approach relies on known theoretical propagation results for a fixed set of operators. In that example, we identify three possible reasons why structural results cannot be inferred:

1. The correct common propagated space has not yet been identified.

2. The appropriate operators to model the problem are missing from the current library and would need to be introduced along with corresponding propagation rules.

3. It is theoretically possible that the Bellman equation propagates a functional space even though none of the individual operators does—our current framework relies on a sufficient but not necessary condition, namely that *each* operator propagates the space.

To overcome these limitations, future work could involve extending the family of operators and enriching the library of propagation results. This can be done manually, following the methodology of Koole, or through automated discovery using machine learning techniques.

## L DATASET

We constructed a dataset of 36 natural language descriptions of queueing control problems, varying in difficulty by state space size, state constraints, and number of event types. To assess performance in structure identification and support future research, the dataset includes three categories: (1) problems with provable structural results (e.g., Example 1); (2) problems with empirically observed, but unprovable, structures (e.g., Example 2); and (3) problems with no structural results. All problems address realistic issues from domains such as hospital management Bekker et al. (2017), telecommunications Koole & Mandelbaum (2002); Bhulai & Koole (2003); Bekker et al. (2011); Zhang et al. (2025c), freight dispatching Schwarz & Daduna (2006); Amjath et al. (2023), assembly lines Adeyinka & Kareem (2018), and traffic control Boon et al. (2023).

The problems are inspired by the literature and have each been manually designed and solved by an expert in OR. The ground truth consists of five randomly chosen states together with their optimal values. These optimal values were computed from the OR formulation using a general-purpose value iteration solver, with convergence assumed once the value changed by less than 0.05 between two iterations. A formulation is therefore considered correct if its optimal value function matches the ground truth within a tolerance of 0.1.

Table 6 and Fig. 11 sum up the characteristics of the dataset.

Table 6: Overview of the dataset by domain.

| Domain | # Problems | With Structural Properties | Avg. # States | Avg. Dim. | Avg. # Events |
|---|---|---|---|---|---|
| Hospital management | 15 | 12 | 1017 | 3.3 | 6.7 |
| Freight dispatching | 9 | 6 | 335 | 2.2 | 5.9 |
| Assembly lines | 6 | 0 | 217 | 2.8 | 4.5 |
| Traffic control | 4 | 0 | 93 | 2.0 | 5.25 |
| Telecommunications | 2 | 2 | 726 | 2.5 | 5.0 |
| **Total** | **36** | **20 (56 %)** | **594** | **2.8** | **5.9** |

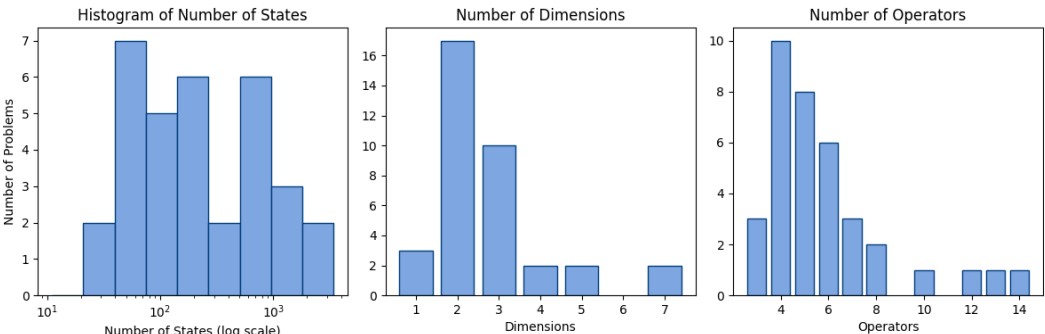

Figure 11: Distribution of some complexity measures across the dataset.

## M PROMPTS

We aim to elicit the LLM to give the final output as a Python dictionary of the following format:

```
formalization_dict template

{
  "parameters": {
    "values": {},
    "descriptions": {}
  },
  "state_space": {
    "variables": {},
    "constraints": {}
  },
  "objective_function": {
    "operational_cost_per_unit_time": null,
    "discount_factor": null,
    "description": null
  },
  "events": {},
  "events_probabilities": {
    "uniformization_factor": null,
    "probabilities": {}
  },
  "operators": {}
}
```

In the following, we describe the prompts that ask the LLM to generate nodes in the Monte Carlo tree. As we will see, the prompts are completely application-agnostic and can be directly applied to problems in different domains.

We give the LLM a general context prompt at the beginning.

---

**General context prompt**

**I have a sequential decision problem:** ______________________________________
<<<PROBLEM DESCRIPTION>>>

---

I want to analyze this problem using the **Event-based Optimization Framework** introduced by Koole (2007). This framework models systems where actions are taken in response to random, uncontrollable events that occur over time. The framework's core components are **event operators**, which serve as the building blocks for defining the system's value function.

**Framework Description**

1. **Event Operators**: Event operators represent the dynamic transformations of the value function in response to specific system events. Formally:

$$T_i : V \to V, \quad \text{for } i = 0, \dots, k - 1,$$

where $T_i$ maps a value function $V$ from the state space to a new value function over the same space.

2. **Recursive Value Function**: The system's value function, $V_n$, is defined recursively to capture the sequential nature of decision-making:

$$V_n = \sum_{i=0}^{k-1} p_i T_i(V_{n-1}),$$

where:

- $V_{n-1}$ is the value function from the previous step.
- $p_i$ is the probability of event $i$ occurring at each step, satisfying $\sum_{i=0}^{k-1} p_i = 1$.
- $T_i$ represents the impact of event $i$ on the system.
- $C$ is the operational cost.
- $\alpha$ is the discounting factor.

This formalization captures the stochastic nature of the problem, where random events dictate the evolution of the system, and the value function reflects the accumulated system performance over time.

**Objective** Given this theoretical foundation, we need to formalize the problem by defining the following components:

---

<<<FORMALIZATION DICT>>>

---

The following prompt generates the first-level nodes $m_1$ of the Monte Carlo tree.

---

**Parameters completion prompt**

**Task:** Complete `formalization_dict` based on the problem description, you should complete the `"parameters"` field which consists of assigning constants to descriptive variable names. Only complete `"parameters"` and nothing else.

**Guidelines:**

1. Your primary responsibility is to define all the parameters from the problem description that will later be used to define the state space, objective function, events and operators.

2. You may include additional parameters in a format suitable for facilitating the subsequent tasks of defining the state space, objective function, events and operators.

3. For parameters that involve multiple indices (e.g. `x[i]` or `x[i, j]`), use the most appropriate data structure, such as lists, dictionaries, or dictionaries with tuple keys, to represent them.

4. For each parameter, include a clear, descriptive comment explaining its meaning.

5. Ensure that the parameter names (keys) are descriptive and intuitive.

6. The dictionary should contain two keys: `"values"` and `"descriptions"`.

   - `"values"` should contain the actual parameter values.
   - `"descriptions"` should contain the descriptions of the parameters.

**Format:** Return only the Python dictionary update (i.e., `formalization_dict["parameters"] = ...`) following the described requirements.

---

The following prompt generates the second-level nodes $m_2$ of the Monte Carlo tree.

---

**State space completion prompt**

**Task:** Complete the `"state space"` field in the `formalization_dict` based on the problem description. Specifically, define:

**1. Variables:** Populate the `"variables"` field to represent the system's state. Each key-value pair must adhere to the following structure:

```
<key>: {
    "description": <description>,
    "type": <type>,
    "iteration_space": <space>,
    "default_value": <default_value>
}
```

**Guidelines:**

- **Essential Variables Only**: Include only the strictly necessary variables to describe the system's state. Exclude costs, events, or redundant variables.
- **Less is better**: Due to the curse of dimensionality, keep the number of state variables minimal. If a variable can be derived from others, do not include it.
- **Symbolic Name**: Use unique, descriptive names that reflect the variable's role.
- **Description**: Clearly explain each variable's role in the system.
- **Type**: Either `"int"` or `"float"`.
- **Parameter Variables**: Use parameter-defined values directly (without using `parameters[...]`)
- **Iteration Space**: Use Python-style list comprehension syntax (e.g., `range(n)`). Use `None` for scalar variables.
- **Default Value**: Must be a single `int` or `float` to initialize the variable across its iteration space.
- **Consolidation**: Merge similar variables under a single key with an appropriate iteration space.

**2. Constraints:** Populate the `"constraints"` field to define boundaries of the state space. Each key-value pair must adhere to the following structure:

```
<constraint_key>: {
    "equation": <mathematical_equation>,
    "description": <description>
}
```

**Guidelines:**

- **Descriptive Constraints**: Use meaningful names.
- **Mathematical Description**: Use Python-like math expressions. Use list comprehensions when appropriate.
- **Equality and Inequality**: Capture valid bounds and implicit problem constraints.
- **Parameter Variables**: Refer directly to them, no nested `parameters[...]` syntax.
- **Indexed Variables**: Use bracket notation (e.g., `x[i]`).
- **Comments**: Each constraint should be preceded by a comment explaining its purpose.

**Important Notes:**

- If the problem has no explicit constraints, consider implicit ones.
- If no constraints apply, return: `formalization_dict["state space"]["constraints"] = {None: None}`

---

**Return:** Only the Python dictionary update (i.e., `formalization_dict["state space"] = ...`) following the described requirements.

The following prompt generates the third-level nodes $m_3$ of the Monte Carlo tree.

---

**Objective function completion prompt**

**Task:** Complete `formalization_dict` based on the problem description, you should complete the `"objective function"` field. Follow these requirements:

Define the operational cost and discount factor such that it adheres to the following structure:

```
"objective function" = {
    "operational cost":  <cost>,
    "discount factor":  <factor>,
    "description":  <description>
}
```

**1. Operational Cost:**

- Replace `<cost>` with the **operational cost** of the system. This represents the cost incurred *between events*, such as maintaining the system or executing ongoing operations.
- **Important:** Do not include costs triggered by events or actions — those go in the `"events"` field.
- Use parameter-defined variables, not hard-coded values. Express the cost as a string formula.
- **Default:** If not provided, use `"0"`.

**2. Discount Factor:**

- Replace `<factor>` with the system's **discount factor**, which determines the relative importance of future rewards.
- Use parameter-defined variables if mentioned. Otherwise, use the default value `0.95`.

**3. Description:**

- Replace `<description>` with a short explanation justifying the chosen operational cost and discount factor.

**Return:** Only the Python dictionary update (i.e., `formalization_dict["objective function"] = ...`) following these requirements.

---

The next few prompts generate the fourth-level nodes $m_4$ of the Monte Carlo tree.

---

**Events completion prompt**

**Task:** Complete `formalization_dict` based on the problem description, you should complete the `"events"` field. Follow these requirements:

Each key-value pair in the dictionary must adhere to the following structure:

```
<key>:  {
   "description":  <description>,
   "actions":  {
     <action_key>:  {
       "description":  <action description>,
       "cost":  <cost>,
       "state_change":  <state_change>
     }
   }
}
```

**Guidelines:**

1. **Events:** Define the events that can occur in the system. An event is a random occurrence that changes the state, the cost, or triggers a need for action.

2. Each `<key>` must be a symbolic name representing a distinct event and will be used in the Python implementation of operators.

3. **Descriptive Events:** Use unique, symbolic names for each event.

4. **Event Description:** Replace `<description>` with a string describing the event's impact on the system.

5. **Actions:** Define the actions that can be taken in response to each event. If there is no decision involved, define only one action named `"default"`.

6. **Action Description:** Replace `<action description>` with a description of the action's role.

7. **Cost:** Replace `<cost>` with a string representing the formula for the cost of the event-action pair. Use parameter-defined variables.

8. **State Change:** Replace `<state_change>` with a list of equations (as strings) describing how state variables change due to the event and action.

9. **Constraints:** Do not repeat feasibility constraints — infeasible states automatically result in $V = +\infty$.

10. **Parameter Variables:** Use parameter-defined variables directly; do not reference them via `parameters[...]`.

11. **Indexed Variables:** Use bracket notation for indices (e.g., `x[0]`).

12. **One Event per Entry:** Do not merge events. Each event must have its own dictionary entry. Avoid undefined parameters (e.g., no generic `i` in event keys).

**Return:** Only the Python dictionary update (i.e., `formalization_dict["events"] = ...`) following these requirements.

---

---

**Events probabilities completion prompt**

**Task:** Complete `formalization_dict` based on the problem description, you should complete the `"events_probabilities"` field.

**Requirements:** Each key-value pair in the dictionary must adhere to the following structure:

`<key>:    <probability>`

1. **Events Probabilities:** Define the probabilities of each event that can occur in the system. If needed, consider a uniformization framework.

2. Each `<key>` links to the corresponding event defined in the `"events"` section.

3. Replace `<probability>` by the probability of the event. Use parameter-defined variables instead of hard-coded values. Put the formula in a string format.

4. **Parameter Variables:** Use parameter-defined variables directly (do not reference them via `parameters[...]`).

5. **Indexed Variables:** For indexed state variables or parameters, use standard bracket notation (e.g., `x[i]`).

**Return:** Only the Python dictionary update (i.e., `formalization_dict["events probabilities"] = ...`) following these requirements.

---

**Operators completion prompt**

**Task:** Complete `formalization_dict` based on the problem description, you should complete the `"operator"` field.

Each operator is a 'sub Bellman' equation linking the optimal value function after the occurrence of the corresponding event (and the optimal action) to the value function before that.

**List of Available Operators:**

- `T_{A}`
    - **Description:** Arrival operator
    - **Definition:** $T_A(\texttt{state\_variable})\,f(x) = f(x + e_{\texttt{state\_variable}})$
    - **Parameters:** `['state_variable']`
- `T_{CA}`
    - **Description:** Controlled arrival operator
    - **Definition:** $T_{CA}(\texttt{state\_variable}, c_1, c_2)\,f(x) = \min(f(x) + c_1, f(x + e_{\texttt{state\_variable}}) + c_2)$
    - **Parameters:** `['state_variable', 'c_1', 'c_2']`
- `T_{D}`
    - **Description:** Departure operator
    - **Definition:** $T_D(\texttt{state\_variable})\,f(x) = f((x - e_{\texttt{state\_variable}})^+)$
    - **Parameters:** `['state_variable']`
- `T_{CD}`
    - **Description:** Controlled departure operator
    - **Definition:**

$$T_{CD}(\texttt{state\_variable}, c_1, c_2)\,f(x) = \begin{cases} \min(f(x) + c_1, \\ \quad f(x - e_{\texttt{state\_variable}}) + c_2), & \text{if } x_{\texttt{state\_variable}} > 0 \\ c_1 + f(x), & \text{otherwise} \end{cases}$$

    - **Parameters:** `['state_variable', 'c_1', 'c_2']`
- `T_{TD}`

– **Description:** Tandem departure operator

– **Definition:**

$$T_{TD}(\texttt{state\_variable\_1},\texttt{state\_variable\_2})\,f(x) =$$

$$\begin{cases} f(x - e_{\texttt{state\_variable\_1}} \\ \quad + e_{\texttt{state\_variable\_2}}), & \text{if } x_{\texttt{state\_variable\_1}} > 0 \\ f(x), & \text{otherwise} \end{cases}$$

– **Parameters:** ['state\_variable\_1', 'state\_variable\_2']

• T\_{CTD}

– **Description:** Controlled tandem departure operator

– **Definition:**

$$T_{CTD}(\texttt{state\_variable\_1}, \texttt{state\_variable\_2}, c_1, c_2)\,f(x) =$$

$$\begin{cases} \min(f(x) + c_1,\ f(x + e_{\texttt{state\_variable\_1}} + e_{\texttt{state\_variable\_2}}) + c_2), & \text{if } x_{\texttt{state\_variable\_1}} > 0 \\ c_1 + f(x), & \text{otherwise} \end{cases}$$

– **Parameters:** ['state\_variable\_1', 'state\_variable\_2', 'c\_1', 'c\_2']

**Field Format:** Each key-value pair in the dictionary must follow this structure:

```
<key>: {
   "description": <description>,
   "operator": <operator>
}
```

**Guidelines:**

1. Replace `<key>` with the name of the event the operator corresponds to.

2. Replace `<description>` with a string that explains the operator's impact.

3. Replace `<operator>` with the selected operator and its parameters in string format (e.g., `"T_{CA}(i=x[1], c_1=1, c_2=2)"`).

4. If no operator fits, use `None`.

5. Use parameter-defined variables directly (not via `parameters[...]`).

6. Use bracket notation for indexed variables (e.g., `x[i]`).

7. For repeating patterns, use Python for-loops or list comprehensions where applicable.

8. Do **not** include event probabilities — they are handled elsewhere.

**Return:** Only the Python dictionary update (i.e., `formalization_dict["operators"] = ...`) following these requirements.

