# OpenReview forum: "Operator Theory-Driven Autoformulation of MDPs for Control of Queueing Systems"
_ICLR.cc/2026/Conference — ICLR 2026 Poster_

### Official Review · Reviewer_zt4g · 2025-10-26

**Soundness:** 4
**Presentation:** 3
**Contribution:** 4
**Rating:** 8
**Confidence:** 2

**Summary:**

A very interesting paper proposing an automated approach to translate and solve MDPs formulated as natural language descriptions within the domain of queueing theory. The approach uses LLMs to translate the natural language into an operator graph; the paper then provides theoretical and algorithmic results to operate on such graphs. The paper is complemented by a large set of tasks, itself a valuable resource.

**Strengths:**

While this is not my research field, it seems to me that the topic and approach are highly original. The paper has a nice combination of theoretical results (the introduction of the operator graph and Thm 4.1), algorithmic results (including complexity results on them, Thm 4.2), and implementation. The data set assembled consists of many tasks from realistic scenarios, and itself may constitute a valuable resource.

**Weaknesses:**

My understanding is that the theoretical/ algorithmic results are still predicated on the LLM being able to construct a correct graph, which is clearly not always the case. The paper is very dense, nevertheless the addition of "Take away" summaries enable also the reader with passing knowledge to get a grasp of the main innovation.

**Questions:**

I don't have particular questions at the moment, the paper was clear in its logical flow.

---

> ### Author Response · Authors · 2025-11-26
>
> *We thank the reviewer for the insightful comments.*
>
> ---
>
> ### [P1] Reliance on the Capability of LLMs
> > Comment: My understanding is that the theoretical/ algorithmic results are still predicated on the LLM being able to construct a correct graph, which is clearly not always the case. The paper is very dense, nevertheless the addition of "Take away" summaries enable also the reader with passing knowledge to get a grasp of the main innovation.
>
> You are correct that a key part of the autoformulation process relies on the capability of the LLM in contructing the correct graph. Nonetheless, our conceptual contribution and theoretical results remain essential.
>
> First, by viewing the composition of the Bellman equation as a graph of operators, we can decompose the search process into different stages, which is suitable for Monte Carlo tree search (MCTS). In MCTS, the LLM only needs to generate one component of the formulation, instead of the entire formulation all at once.
>
> Second, Theorem 4.1 allows us to significantly reduce the search space of the LLM, and hence improve the search accuracy.
>
> Our experiments also demonstrate the improved accuracy of our proposed method over benchmark uses of LLMs.

---

> > ### Comment · Reviewer_zt4g · 2025-11-26
> >
> > Thanks for your response, my comment was only made so that you may want to consider highlighting the limitation perhaps in the discussion section, I appreciate that you can't solve all problems all at once!

---

### Official Review · Reviewer_7bBF · 2025-10-29

**Soundness:** 3
**Presentation:** 3
**Contribution:** 4
**Rating:** 10
**Confidence:** 4

**Summary:**

The paper describes an approach to use LLMs to automatically formulate sequential decision problems as a class of MDPs. Of particular interest is the formulation of the solution of MDPs in terms of "operators" or transformations, and a dynamic programming approach to reason over the compositions of operators. When the MDP exhibits some structural properties such as monotonicity, the formulation is able to exploit it and solve the MDP efficiently. Besides the search space of different formulations is also made efficient by the operator theory approach. MCTS is used to search the space of formulations and is shown to be effective over 36 natural language descriptions of queuing problems of different complexities.

**Strengths:**

+ Autoformulation is an important understudied problem that bridges the gap between the LLMs and the classical AI such as optimization. The current work closes an important gap by bringing autoformualtion to the MDP literature.
+ Automatically deriving the structural properties of the policies through the analysis of the operator composition is an important contribution. It makes the application of MDPs to real world problems much easier by non-experts.
+ Empirical results show that Syntax Checks (SC) and Solver Feedback (SF) makes MCTS much more effective, out-competing state of the art methods such as Chain of Thought and targeted prompts.

**Weaknesses:**

- The appendix of the paper is too long (40+ pages) and unsuitable for conference reviewing.

- On the other hand some parts of the appendix appear too important to be in the appendix, eg, the algorithms in F and the theorem in E.
The authors should consider a better packaging of the paper, cut the appendix, and write a full-length journal paper that includes all details.

070, 074 ComputationAL challenges
085. BellMAN equation
311. "brutal force" -> "bruteforce"
354. SF and SC are referred here for the first time, but defined only in 364.
450. "untractable" -> "intractable"
479. "an universal" -> "a universal" ('u' is a vowel, but a/an distinction is based on how it sounds).

**Questions:**

The example in 313-317 sounds mysterious. You need an explanation of why A \bigcap B \bigcap C is the right answer right here and not in the appendix. In general it is not clear how \bigcap and \subset should be interpreted here and hence what B \bigcap C \subset E means.

There seems to be an error in the Equations in page 25. The last step of Equation (58) has one \gamma. However, the next 3 transformations contain two \gammas (the first and the third) and composing them seems to yield \gamma^2. Can you clarify?

---

> ### Author Response · Authors · 2025-11-26
> **Response to Reviewer 7bBF (Part 1/2)**
>
> *We thank the reviewer for the insightful comments.*
>
> ---
>
> ### [P1] Organization of The Paper
> > - The appendix of the paper is too long (40+ pages) and unsuitable for conference reviewing.
> > - On the other hand some parts of the appendix appear too important to be in the appendix, eg, the algorithms in F and the theorem in E. The authors should consider a better packaging of the paper, cut the appendix, and write a full-length journal paper that includes all details.
>
> We appreciate the reviewer’s feedback regarding the length and organization of the appendix.
>
> In structuring the main text, we prioritized clarity on the context, goals, and contributions of the paper. Unfortunately, due to conference space constraints, the most technical components could not be included in the main text. We also included some existing results from the literature (`Appendix C.1` and `Appendix D`; 6.5 pages in total), just to make the paper self-contained. We agree that some of these details are important, and we appreciate the suggestion to prepare a full-length journal version that integrates them more naturally. We will take this recommendation into careful consideration.
>
>
> ### [P2] Typos
> > Comment: 070, 074 ComputationAL challenges 085. BellMAN equation 311. "brutal force" -> "bruteforce" 354. SF and SC are referred here for the first time, but defined only in 364. 450. "untractable" -> "intractable" 479. "an universal" -> "a universal" ('u' is a vowel, but a/an distinction is based on how it sounds).
>
> Thank you for the careful reading. We have corrected all the noted typographical and grammatical issues in the revised manuscript.
>
> ### [P3] Clarity of the Example
> > Comment: The example in 313-317 sounds mysterious. You need an explanation of why A \bigcap B \bigcap C is the right answer right here and not in the appendix. In general it is not clear how \bigcap and \subset should be interpreted here and hence what B \bigcap C \subset E means.
>
> Thank you for raising this important point. We agree that the example needed a better explanation.
>
> **Meaning of $B \cap C \subset E$.** We realized that the confusion comes from our abuse of language. By "property B", we really meant the *set of functions* that satisfy that property. Hence, $\cap$ and $\subset$ are standard set operations. For example, $B$ can be the set of supermodular functions, $C$ can be the set of "superconvex" functions as defined in `Section 6.3` of (Koole, 2007), and then $B \cap C$ is the set of supermodular and superconvex functions. We know that the set $B \cap C$ is a subset of component-wise convex functions, denoted by $E$. So in general, the notation $B \cap C \subset E$ means that any function that satisfies both properties $B$ and $C$ also satisfies property $E$.
>
> **Why $A \cap B \cap C$ is the right answer.** Consider the example where operator $T_1$ propagates $A \cap B$, $E \cap C$, and $D$, and operator $T_2$ propagates $C \cap A$, $D \cap F$, and $B$. Note that $T_2$ does *not* propagate $A$ by itself. We aim to find an intersection of some sets among $A-F$ that can be propagated by both $T_1$ and $T_2$. For example, $A \cap B$ is propagated by $T_1$, but not by $T_2$, because $C$ is needed for $A$ to be propagated by $T_2$. Without $B \cap C \subset E$, we can verify that nothing is propagated by both $T_1$ and $T_2$. However, with $B \cap C \subset E$, we have $A \cap B \cap C = (A \cap B) \cap (E \cap C)$, which is propagated by $T_1$. It is not hard to see $A \cap B \cap C$ is also propagated by $T_2$.
>
> **Actions Taken:** We added the above explanations in `Section 4.3`, and added `Appendix F.1` for even more detailed discussions.

---

> ### Author Response · Authors · 2025-11-26
> **Response to Reviewer 7bBF (Part 2/2)**
>
> ### [P5] Proof in `Appendix E`
> > There seems to be an error in the Equations in page 25. The last step of Equation (58) has one \gamma. However, the next 3 transformations contain two \gammas (the first and the third) and composing them seems to yield \gamma^2. Can you clarify?
>
> Thank you for this very important remark. You correctly pointed out an error in the proof presented in the paper. We have corrected this issue and also adjusted the flow of the proof. The updated version of the proof is provided below.
>
> For any MDP $(\\mathcal{S}, \\mathcal{A}_s, c, P, \\gamma)$, the standard Bellman equation for the value function $V\_n(s)$ on the full state $s$ can be written as:
>
> \begin{align}
> V\_{n+1}(s) = \\min\_{a\\in \\mathcal{A}\_s}\\left\\{c(s,a) + \\gamma \\sum\_{s' \\in \\mathcal{S}} P(s'\\mid s,a) V\_n(s') \\right\\}.
> \end{align}
>
> Separating the controllable state and the event $s=(x,e)$ and using the definition of event-based MDPs, we have
>
> \begin{align}
> V_{n+1}(x, e) &= \\min\_{a\\in \\mathcal{A}\_{(x, e)}}\\left\\{
> c[(x,e),a] + \\gamma \\sum\_{(x', e')} P\\left[(x^\\prime,e^\\prime) \\,|\\, (x,e), \\, a\\right] \\cdot V\_n(x', e')\\right\\} \\\\
> &= \\min\_{a\\in \\mathcal{A}\_{(x, e)}}\\left\\{
> c[(x,e),a] + \\gamma \\sum\_{(x', e')} P\_{\\texttt{x}}\\left[x' \\,|\\, (x,e), \\, a\\right] \\cdot P\_{\\texttt{e}}(e' \\,|\\, x') \\cdot V\_n(x', e')\\right\\} \\\\
> &= \\min\_{a\\in \\mathcal{A}\_{(x, e)}}\\bigg\\{
> c[(x,e),a] + \\sum\_{x'} P\_{\\texttt{x}}\\left[x' \\,|\\, (x,e), \\, a\\right] \\cdot
> \\underbrace{{\\color{red}\\gamma} \\sum\_{e'} P\_{\\texttt{e}}(e' \\,|\\, x') \\cdot V\_n(x', e')}\_{\\triangleq V\_n^\*(x')}\\bigg\\} \\\\
> &=  \\min\_{a\\in \\mathcal{A}\_{(x, e)}}\\left\\{
> c[(x,e),a] + \\sum\_{x'} P\_{\\texttt{x}}\\left[x' \\,|\\, (x,e), \\, a\\right] \\cdot V\_n^\*(x')\\right\\},
> \end{align}
>
> where we define the value function $V_n^*(x)$ on the controllable state :
>
> \begin{align}
> V\_n^\*(x) = {\\color{red}\\gamma}\\sum\_{e} P\_{\\texttt{e}}(e \\,|\\, x) V\_n(x, e).
> \end{align}
>
> **Remark about this change :** We define $V\_n^{\\star}(x)$ as the value of the state immediately after an action is taken but *before* waiting for the next event. Because there is a temporal gap between ${V\_n^{\\star}}$ and $V_n$, the discount factor must be applied at this stage to account for that delay. In the former definition, $V\_n^{\\star}(x)$ represented the value of the state just before an event occurred (after the waiting time). This is a slight change in timing, but it caused the discount factor to appear in the wrong place in the final Bellman equation.
>
> We can always decompose the cost $c[(x,e),a]$ into two parts: (1) a cost $c(x)$ that depends only on the controllable state (e.g., the holding cost in the M/M/1 example), and (2) a cost $\\tilde{c}[(x,e),a]$ that depends on the full state-action pair, namely
>
> \begin{align}
> c[(x,e),a] = c(x) + \\tilde{c}[(x,e),a].
> \end{align}
>
> Note that if the component $c(x)$ does not exist, we can always set $c(x) = 0$ and $\\tilde{c}[(x,e),a] = c[(x,e),a]$.
> Then the Bellman equation above can be rewritten as
>
> \begin{align}
> V\_{n+1}^{\\star}(x)
> &= {\\color{red}\\gamma}\\sum\_{e} P\_{\\texttt{e}}(e \\,|\\, x) V\_{n+1}(x, e) \\\\
> &= {\\color{red}\\gamma}\\sum\_{e} P\_{\\texttt{e}}(e \\,|\\, x)\\bigg[
> c(x) + \\min\_{a\\in \\mathcal{A}\_{(x, e)}}\\left\\{
> \\tilde{c}[(x,e),a] + \\sum\_{x'} P\_{\\texttt{x}}[x' \\,|\\, (x,e), \\, a] \\cdot V\_n^{\\star}(x')\\right\\}
> \\bigg] \\\\
> &= \\underbrace{{\\color{red}\\gamma}c(x)}\_{\\color{red}{\\triangleq c'(x)}} + {\\color{red}\\gamma}\\sum\_{e} P\_{\\texttt{e}}(e \\,|\\, x)
> \\left[
> \\min\_{a\\in \\mathcal{A}\_{(x, e)}}\\left\\{
> \\tilde{c}[(x,e),a] + \\sum\_{x'} P\_{\\texttt{x}}[x' \\,|\\, (x,e), \\, a] \\cdot V\_n^{\\star}(x')
> \\right\\}
> \\right].
> \end{align}
>
> Therefore, we can go from $V\_n^{\star}(x)$ to $V\_{n+1}^{\star}(x)$ through the following three operators:
>
> \begin{cases}
> T\_{\texttt{e}\_j}[V\_n^{\\star}] = V\_{n+1}(x,\\texttt{e}\_j)
> = \\min\_{a\\in \\mathcal{A}\_{(x, e)}}\\left\\{
> \\tilde{c}[(x,e),a] + \\sum\_{x'} P\_{\\texttt{x}}[x' \\,|\\, (x,e), \\, a] \\cdot V\_n^{\\star}(x')
> \\right\\} \\\\
> T\_{\texttt{unif}}[U\_1,\\ldots,U\_\ell]
> = \\sum\_{j=1}^\ell P(\\texttt{e}\_j \\,|\\, x) U\_j \\\\
> T\_{\texttt{cost}}[U] = {\\color{red}c'(x)} + \\gamma U
> \end{cases}
>
> The operator-based Bellman equation on the value function $V_n^{\star}(x)$ can be written as
>
> \begin{align}
> V\_{n+1}^{\\star}(x) = T\_{\texttt{cost}}\\Big(
> T\_{\texttt{unif}}\\big(
> T\_{\texttt{e}\_1}[V\_n^{\\star}], \\ldots, T\_{\texttt{e}\_\ell}[V\_n^{\\star}]
> \\big)
> \\Big).
> \end{align}

---

### Official Review · Reviewer_4PC5 · 2025-10-31

**Soundness:** 4
**Presentation:** 4
**Contribution:** 3
**Rating:** 8
**Confidence:** 3

**Summary:**

This paper introduces a novel operator-theoretic framework for the autoformulation of Markov Decision Processes (MDPs) from natural-language descriptions, with a specific focus on queueing control problems (e.g., hospital ward management, call centers). Unlike existing works that use large language models (LLMs) to autoformulate one-shot optimization problems, this paper extends the idea to sequential decision-making, which involves greater complexity due to dynamic transitions, stochasticity, and implicit constraints.

**Strengths:**

1. Theoretical novelty: The paper makes a conceptually and mathematically novel contribution by connecting operator theory with automatic MDP formulation. It introduces an operator-graph representation of Bellman equations, and rigorously proves the existence of a universal three-level operator topology applicable to a broad class of event-based MDPs. This gives the autoformulation task a solid theoretical structure.

2. End to end framework: The proposed operator-graph framework is comprehensive and well-engineered. It integrates syntax checking, solver feedback, and self-rewarding search to achieve high formulation accuracy and interpretable policies. Empirical evaluation on queueing control problems demonstrates consistent and significant gains over baselines.

This paper presents a theoretically grounded and well-validated framework that advances the automation of sequential decision problem formulation, making it a strong candidate for acceptance.

**Weaknesses:**

This framework is evaluated only on queueing control problems, which, although well-chosen, represent a narrow subclass of event-driven MDPs. The paper does not demonstrate whether the proposed method generalizes to other sequential decision domains such as robotics, finance, or inventory control.

**Questions:**

Beyond queueing systems:  would be good to see the proposed operator-graph topology and autoformulation framework generalize to non-queueing domains.

---

> ### Author Response · Authors · 2025-11-26
>
> *We thank the reviewer for the insightful comments.*
>
> ---
>
> ### [P1] Beyond Queueing Systems
> > - This framework is evaluated only on queueing control problems, which, although well-chosen, represent a narrow subclass of event-driven MDPs. The paper does not demonstrate whether the proposed method generalizes to other sequential decision domains such as robotics, finance, or inventory control.
> > - Beyond queueing systems: would be good to see the proposed operator-graph topology and autoformulation framework generalize to non-queueing domains.
> ---
>
> We agree with the reviewer that expanding beyond queueing problems is an important research direction, and we acknowledge that this point is not discussed sufficiently in the current version of the paper.
>
> Our view is that the most natural next step is to extend the framework to more general MDPs with discrete state space (not only event-based queueing systems). Doing so would require working without Theorem 4.1, but we are confident that a more flexible algorithm can be developed to automatically formulate a broader class of models. Importantly, the component of our method that identifies structural properties operates on any operator graph and would not require modification.
> Inventory control is a good example of a more complex MDP that remains conceptually close to queueing problems. For instance, order decisions can be made at any time rather than only after random events, and orders typically involve delays before arrival. Such systems can still be represented as DAGs of operators, but their topologies differ significantly from those the current algorithm is designed to construct.
>
> We believe the approach can generalize to wider domains, subject to certain constraints. First, we must be able to write down sufficiently explicit models of how actions and the environment interact to enable proving propagation results on the operators. The model does not need to be fully specified (e.g., arrival rates could be learned through RL), as long as the underlying operators can be identified. In robotics, however, environmental dynamics are often too complex to express through analyzable operators. The relevant structure in such domains tends to be higher-level than the analytical, low-level structural properties treated in this paper. These higher-level structures, which motivate methods such as hierarchical RL, should indeed be incorporated into future autoformulation algorithms, but doing so would require an approach that differs significantly from ours, even if the overarching philosophy (linking autoformulation with computational methods) remains similar.
>
> A second important constraint concerns the state space. Some domains have unstructured or highly irregular state spaces, which can break propagation properties. For example, convexity-based arguments rely on convex state spaces.
>
> In summary, we expect the current framework to generalize well to discrete MDPs with convex state spaces, such as inventory control and other operations research problems (e.g., hospital operations, freight dispatching as in the paper) in more complex settings, for instance with actions available continuously in time, "deterministic events" occurring at fixed frequencies, or richer environmental dynamics. For domains that fall outside these specifications, the autoformulation method would need to be substantially different, as the low-level structural results used in this paper may no longer be available, and other types of structure would need to be identified.
>
> **Actions Taken:** We added the discussion in the conclusion of the paper (`Section 6`).

---

### Official Review · Reviewer_w778 · 2025-11-02

**Soundness:** 2
**Presentation:** 2
**Contribution:** 3
**Rating:** 2
**Confidence:** 4

**Summary:**

This paper proposes a method to autoformulate MDPs for the control of queueing systems using LLMs and grounded on operator theory.
The core of the proposal is to construct the MDP's Bellman equation as a graph of operators, each operator transforming the value function corresponding to certain event. The paper develops a three-level operator-graph topology that covers a broad class of MDPs. The paper proposes a Monte Carlo tree search algorithm to build such operator graphs, and an algorithm that identifies structures of the optimal policy to accelerate the solution.

**Strengths:**

S1. Core structural result (Theorem 4.1) on the existence of a universal three-level operator-graph topology that represents a broad class of MDPs for queueing/control.

S2. Application of the operator graph to queueing-control MDPs and other examples, showing interpretability.

S3. Positioning within operator-learning trends for the sake of structured representation learning.

**Weaknesses:**

W1. The blanket claim that this the first work to represent Bellman equations as directed acyclic graphs (DAGs) of operators is unsupportable. Prior work already treats Bellman-style equations/updates compositionally or on graphs. The paper asserts precedence for the general idea of viewing Bellman equations as DAGs (graphs) of operators. There is substantial prior literature that already represents Bellman-like computations compositionally, localizes Bellman updates to nodes in a graph/DAG, or studies Bellman operators and their compositions. Representative, explicit prior works:

Yu, Mahmood, Sutton: On Generalized Bellman Equations and Temporal-Difference Learning, ICML 2017. This paper develops generalized Bellman equations, explicitly treats different multi-step / trace-based operators and shows how value-function equations arise from composing/choosing among operators; the paper frames Bellman relations in operator terms.

Gopalan et al.: Planning with Abstract Markov Decision Processes, ICAPS 2017. This work decomposes planning into a hierarchy (a DAG-like structure) of abstract subtasks and perform Bellman-style planning localized to nodes/levels.

Jothimurugan et al.: Compositional Reinforcement Learning from Logical Specifications (DiRL), NeurIPS 2021. This work encodes specifications as abstract graphs and composes high-level planning with low-level RL; value/policy computations are decomposed across the graph structure.

W2. Lack of comparison against existing structured MDP representations (factored MDPs, hierarchical MDPs, AMDPs, object-oriented MDPs, modular RL). Experimental comparisons use ad hoc baselines rather than state-of-the-art methods with established theoretical or empirical guarantees.

W3. The scope of applicability (specific classes of queueing/control MDPs) is narrower than the broader claims sound. It is not demonstrated that the same reduction works well for more complex continuous-control MDPs or even on classical RL benchmarks (gridworlds, inventory control, navigation).

W4. The search procedure is described only at a high level and its computational complexity is not specified.

W5. It is unclear whether the chosen operator set (shift, clamp, increment, etc.) applies to other domains.

W6. The claim that the operator-graph topology reduces the MDP search space dramatically is not empirically validated.

**Questions:**

1. What is the complexity of the search procedure?
2. How do the chosen operators apply to other domains?
3. How much is the MDP search space reduced?

---

> ### Author Response · Authors · 2025-11-26
> **Response to Reviewer w778 (Part 1/4)**
>
> *We thank the reviewer for the insightful comments.*
>
> ---
>
> ### [P1] Our Contribution on Representing Bellman Equations as Graphs of Operators -- Response to [W1] in Weaknesses
>
> We thank the reviewer for bringing up these three papers, which gives us opportunity to distinguish our work from **existing literature that uses "graphs" and "Bellman operators" in different contexts and for different purposes**.
>
> **1. Different definitions of "graph" and "operator" in different contexts.**
> While "operators" and "graphs" appeared in many works on Markov decision processes (MDPs), they have very different meanings and purposes in our work.
>
> **Bellman operator vs operator-based Bellman equation.** The three works mentioned, as well as numerous other works, define the *Bellman operator*, denoted $B$, as the *end-to-end mapping* of optimal state values $V^\star$, namely $V^\star = B(V^\star)$. The Bellman operator is essentially a synonym for the Bellman equation for the standard MDP in (Yu et al. 2017) and for the subtask MDP in (Gopalan et al. 2017; Jothimurugan et al. 2021). Yu et al. 2017 also generalize the Bellman operator to analyze their proposed temporal-difference (TD) learning with time-varying $\lambda$ parameters; but the generalized Bellman operators are still Bellman equations of MDPs, adjusted for time-varying $\lambda$ parameters.
>
> In our work, the *operators* $T_1,\ldots,T_n$ are the atomic operations that together construct the Bellman equation $B$, e.g., $B(V^\star) = T_1 [ T_2 ( T_3(V^\star), \cdots, T_n(V^\star)) ]$. So the Bellman operator $B$ in (Yu et al. 2017; Gopalan et al. 2017; Jothimurugan et al. 2021) is a composite function of the operators $T_1,\ldots,T_n$ in our work.
>
> **Graphs of subtasks and graphs of operators.** Gopalan et al. 2017 and Jothimurugan et al. 2021 study hierarchical MDPs and represent the overall task as a graph of subtasks. In contrast, we represent the Bellman equation $B$ as a graph of operators $T_1,\ldots,T_n$.
>
> The transition graph is sparingly used in (Yu et al. 2017) to represent the state transitions of the MDP.
>
> We summarize the discussions in the table below.
>
> `Table 1`. Different definitions and uses of *operator* and *graph*.
>
> | Aspect | Gopalan et al. 2017; Jothimurugan et al. 2021 | Yu et al. 2017 | Our Work
> | :---: | :---: | :---: | :---: |
> | Type of MDPs | Hierarchical MDPs | Standard MDPs | Event-based MDPs
> | What the "operator" is | Standard Bellman equation | Generalized Bellman equation with time-varying $\lambda$ parameters | Atomic operations that are components of Bellman equation
> | Main purpose of operator | To compute the optimal policy | To analyze TD learning with time-varying $\lambda$ | To translate problem description into interpretable building blocks of Bellman equation
> | What the "graph" is | Graph of subtasks to achieve the high-level goal | Graph of state transitions of the Markov chain | Graph of operators to construct the Bellman equation
> | Main purpose of graph | To organize high-level planning over subtasks | To analyze TD learning using structure of state transitions | To construct the Bellman equation; Conceptually useful to structure the search of Bellman equation in autoformulation
> | Interaction between "operator" and "graph" | Graph of subtasks decides which Bellman operator to solve | Graph of states is used to analyze generalized Bellman operator | Graph determines how operators are connectd to construct the Bellman equation
>
> **2. Our contribution put in the context.** As stated in `Line 082` of our paper, our view of *Bellman equation as a composition function of operators* comes from (Koole 1998; 2007), which are two key references in our paper. Therefore, the operator in our work is defined in the same way as Koole's and others' works (e.g., Bekker et al. 2017), and we did *not* claim we invented the operator. However, within this line of works, we are *indeed the first* to view the connection of operators as a *graph*.
>
> By viewing the composition of operators as a graph, we reformulate the problem of searching for the correct Bellman equation (i.e., autoformulation) as a problem of searching for the operator graph. This allows us to discover the existence of a universal graph topology to reduce the search complexity (`Theorem 4.1`) and to propose an algorithm to identify structural properties of optimal policies (`Theorem 4.2`).
>
> **3. Concluding remarks.** In summary, the three works mentioned focus on *solving* the MDP, while our work focuses on *autoformulating* the MDP from natural-language problem description. Their Bellman operators are Bellman equations, while our operators are *components of* Bellman equations. Their graphs describe subtasks or state transitions, while our graph describes how to construct the Bellman equation.
>
> **Actions Taken:** We will add the above discussion in a new `Appendix L` of "Extended Related Works".

---

> ### Author Response · Authors · 2025-11-26
> **Response to Reviewer w778 (Part 2/4)**
>
> ### [P2] Comparison Against Existing Structured MDP Representations
> > Comment: Lack of comparison against existing structured MDP representations (factored MDPs, hierarchical MDPs, AMDPs, object-oriented MDPs, modular RL). Experimental comparisons use ad hoc baselines rather than state-of-the-art methods with established theoretical or empirical guarantees.
>
> We thank the reviewer for this comment, which allows us to clarify that our work focuses on *autoformulating* MDPs, instead of *solving* MDPs.
>
> Specifically, we consider the problem of translating natural-language problem description into mathematical formulation of MDPs (autoformulation), while existing works aim at solving MDPs *that are already formulated*, such as (Gopalan et al. 2017; Jothimurugan et al. 2021) for abstract/hierarchical MDPs. Hence, our work and these existing works **target different stages in the pipeline of solving sequential decision-making problems**. Therefore, it is unnecessary to compare against these solution methods.
>
> The possible confusion may come from the capability of our proposed autoformulation method in facilitating faster solving. Our method allows us to identify structures of the optimal policy straight from the MDP formulation *before* solving the MDP. The identified structures can be potentially incorporated into any existing solving method to further reduce the complexity. Hence, our work can be complementary to those methods for solving MDPs, but is not competing against them.
>
> **Actions Taken:** We will add the above discussion in a new `Appendix L` of "Extended Related Works".
>
> ---
>
> ### [P3] Scope of Applicability
> > Comment: The scope of applicability (specific classes of queueing/control MDPs) is narrower than the broader claims sound. It is not demonstrated that the same reduction works well for more complex continuous-control MDPs or even on classical RL benchmarks (gridworlds, inventory control, navigation).
>
> We thank the reviewer for this comment.
>
> As stated in the title and in the abstract (`Lines 018, 021, 031`), we focus on control of queueing systems. In `Definition 3.1`, we rigorously defined the class of MDPs that we study, namely *event-based MDPs* and provide concrete examples of the class of MDPs in our dataset. Our work is a first step in a new research direction, and we do not claim generality.
>
> In principle, our autoformulation framework could be extended to richer families of MDPs, for instance to models where events depend on both states and actions, where events have deterministic waiting times, or where actions are available at any time, provided that their Bellman equations can still be expressed as compositions of (possibly more general) operators. However, such extensions lie outside the scope of Theorem 4.1, since they would require searching over different graph topologies.
>
> **Actions Taken:** We added this discussion in the concluding `Section 6`.
>
> ---
>
> ### [P4] Application of Operators to Other Domains
> > Comment: It is unclear whether the chosen operator set (shift, clamp, increment, etc.) applies to other domains. How do the chosen operators apply to other domains?
>
> The operator theory has been applied to diverse application domains, such as hospital management, telecommunications, freight dispatching, assembly lines, and traffic control (see `Table 5` in `Appendix B` for a list of references). In the experiments, our dataset contains problems from all the domains mentioned above, which are adapted from the references in `Table 5`.

---

> ### Author Response · Authors · 2025-11-26
> **Response to Reviewer w778 (Part 3/4)**
>
> ### [P5] Computational Complexity of the Search Procedure
> > Comment:  The search procedure is described only at a high level and its computational complexity is not specified. What is the complexity of the search procedure?
>
> MCTS does not have a single, fixed "big-O" complexity in the same sense as classical algorithms like binary search. Its computational cost is largely determined by the number of rollouts $N$, but this quantity does not reflect the intrinsic difficulty of the underlying problem—prompting an LLM once, for example, has a trivially low “complexity” under this metric yet completely fails to solve the task. In classical settings, meaningful difficulty metrics exist (e.g., the input size $N$ in sorting, which directly leads to the $O(N \log N)$ complexity of the most efficient algorithms), but for auto-formulation tasks no such analogue exists. Potential descriptors such as the number of events or the dimensionality of the state space do not capture the true difficulty (which also depends on how the problem is formulated) and do not allow predicting how many MCTS iterations are needed for good performance. As a result, the only complexity measure we can provide is in terms of the number of rollouts $N$, even though this is only a partial indicator of actual efficiency. We therefore first discuss MCTS complexity as a function of $N$.
>
>
> ### **Per-rollout complexity**
>
> Let
>
> * $N$ = number of MCTS iterations (rollouts)
> * $b$ = branching factor (average number of choices per depth)
> * $d$ = search-tree depth
> * $L$ = maximum running duration of the solver (time-out)
>
> A single rollout consists of four steps:
>
> 1. Selection : The algorithm descends from the root to a leaf. In the worst case, it traverses the full tree, giving a complexity of $O(bd)$. This step does not involve LLM or solver calls and is not a bottleneck.
> 2. Expansion : In the worst case (e.g., during the very first rollout), the algorithm expands nodes across all depths, generating candidate options, calling the LLM for each, and comparing them. This yields a complexity of $O(bd)$.
>    Later rollouts typically expand far fewer nodes because much of the tree has already been built.
> 3. Simulation : In our setting, simulation corresponds to running the solver on the formulated problem and evaluating the resulting quality. This step has complexity $O(L)$.
> 4. Backpropagation : Values are propagated back up the path, with complexity $O(d)$.
>
> Overall, the per-rollout complexity is
> $$
> O(bd + L).
> $$
> On harder problems with large state spaces, the solver cost $L$ tends to dominate, whereas on simpler problems the tree-expansion term $bd$ is typically the main contributor.
>
>
> ### **Total complexity over $N$ rollouts**
>
> Across $N$ iterations, the total complexity is
> $$
> O\left(N(bd + L)\right).
> $$
>
> Importantly, this upper bound does not reflect the fact that MCTS reuses large portions of the tree across rollouts. Empirically, we observe this in the token-usage statistics: early in the process, MCTS consumes many tokens because it must generate entire paths, whereas later the token consumption per rollout decreases substantially, confirming that expansion becomes far less expensive.
>
>
> ### **Comparison with best-of-$N$**
>
> The best-of-$N$ strategy has a worst-case complexity of
> $$
> O(NL),
> $$
> which may appear smaller. However, it uses each iteration far less efficiently than MCTS. In practice, MCTS typically requires fewer simulations $N$ to reach comparable or better solution quality.
>
> In conclusion, while the theoretical complexity of MCTS is difficult to express in a way that captures problem difficulty, we can describe it in terms of rollout count as $O(N(bd + L))$. More importantly, our experimental results show that MCTS discovers high-quality solutions faster than the baselines, both in terms of the number of iterations and the number of tokens consumed.

---

> ### Author Response · Authors · 2025-11-26
> **Response to Reviewer w778 (Part 4/4)**
>
> ### [P6] Analysis on Reduction of Search Space
> > Comment: The claim that the operator-graph topology reduces the MDP search space dramatically is not empirically validated. How much is the MDP search space reduced?
>
>
> We thank the reviewer for this insightful question, which motivates the following theoretical analysis of the reduction of search space.
>
> A directed acyclic graph (DAG) of operators is defined by the underlying graph topology together with the operators assigned to its nodes. The entire search space is all the DAGs with different topologies and operators. Theorem 4.1 reduces the search space to *three-level trees with $T_{\text{cost}}$ as the root, $T_{\text{unif}}$ as the single child of the root, and event operators as leaves*. Therefore, the reduction comes from (1) the fixed tree topology and (2) the fixed *types of operators at each level*.
>
> **The reduction in the topology** grows in the order of $2^{N^2}$, where $N$ is the number of nodes. This is computed from the analytical expression for the number of DAGs with $N$ nodes and 1 out-point (i.e., one root node that outputs the state values).
> > Source: The Online Encyclopedia of Integer Sequences A003025:
> Number of n-node labeled acyclic digraphs with 1 out-point. Available at https://oeis.org/A003025.
>
> To have a sense, the numbers of DAGs for $N$ from 2 to 9 are:
> $$
> 1, 2, 15, 316, 16885, 2174586, 654313415, 450179768312.
> $$
>
> **The reduction in the choice of operators** may depend on the application: some applications may have more types of cost functions and events than others. But any reduction here is additional to the reduction in the topology.
>
> **Actions Taken:** We will add the above discussion after `Theorem 4.1`.

---

### Author Response · Authors · 2025-12-03
**Global response**

We would like to thank both ACs and all four reviewers for their time and efforts in providing insightful and constructive feedback.

We are glad that the reviewers acknowledged that **"the topic and approach are highly original"** (`Reviewer zt4g`) and our work **"closes an important gap by bringing autoformualtion to the MDP literature"** (`Reviewer 7bBF`). Our work **"has a nice combination of theoretical results (the introduction of the operator graph and Thm 4.1), algorithmic results (including complexity results on them, Thm 4.2), and implementation"** (`Reviwer zt4g`). Specifically, it contains:
- *"a theoretically grounded and well-validated framework that advances the automation of sequential decision problem formulation"* (`Reviewer 4PC5`);
- *"core structural result (Theorem 4.1) on the existence of a universal three-level operator-graph topology that represents a broad class of MDPs for queueing/control"* (`Reviewer w778`);
- *"application of the operator graph to queueing-control MDPs and other examples, showing interpretability"* (`Reviewer w778`);
- *"data set assembled consists of many tasks from realistic scenarios, and itself may constitute a valuable resource"* (`Reviewer zt4g`).

During the rebuttal, we have responded to the reviewers' comments and made corresponding changes in the revised manuscript (marked by red left border and red text). Our responses and revisions are summarized below.
- We have cleared confusions about the same terminology ("operator" and "graph") that are used in different contexts and for different purposes in the literature (Weakness [W1] by `Reviewer w778`). The discussion was added to the new `Appendix C` "Extended Related Works".
- We have cleared confusions about our focus of *autoformulation* (i.e., translating natural language problem description to mathematical problem formulation), which is orthogonal and complementary to the vast literature on efficient *computational* method (Weakness [W2] by `Reviewer w778`).
- We have clarified the applicability of our framework (Weaknesses [W3][W5] by `Reviewer w778`). A discussion was added to `Section 6` "Discussion".
- We have added the analysis on the reduction of the search space due to `Theorem 4.1` (Weaknesses [W6] by `Reviewer w778`).
- We have added a detailed walkthrough of the example that explains the difficulty and the procedure of identifying structures of the optimal policy (Questions [1] by `Reviewer 7bBF`). The discussion was added to `Section 4.3` and `Appendix G.1`.
- We have fixed the issue of the discounting factor in `Appendix F` (Questions [2] by `Reviewer 7bBF`).

---

### Meta-Review · Area_Chair_KkJ3 · 2026-01-07

**Summary:**

This paper introduces an operator-theoretic framework for autoformulating MDPs from natural-language descriptions, with a focus on queueing-control problems. Reviewers except Reviewer w778 praised the originality of the operator-graph formulation, the theoretical contributions (especially the universal three-level topology), and the well-engineered MCTS-based autoformulation pipeline enhanced by syntax checking and solver feedback. The empirical results across a diverse queueing-control dataset were viewed as convincingly supportive of the approach. While reviewers noted that the scope is currently limited to event-based queueing MDPs and that the appendix is extensive, these issues were not considered fundamental weaknesses.

**Reviewer Concerns:**

Following the rebuttal and author clarifications, I consider all reviewer concerns to have been satisfactorily addressed:

- **Prior work and positioning**: The authors clarified how their notion of operators and graphs differs from existing works, providing a clear contextualization in a new “Extended Related Works” section.
- **Scope of applicability**: The paper now explicitly defines its intended domain (event-based queueing MDPs) and discusses pathways for generalization without overstating claims.
- **Complexity, operator set, and search-space reduction**: The authors added detailed explanations of MCTS rollout complexity, clarified applicability of the operator set, and provided theoretical analysis of search-space reduction.
- **Clarity issues and example explanation**: The authors revised the main text to clarify the propagation example and corrected notational ambiguities.
- **Appendix organization, typos, and minor errors**: All noted issues were corrected, including the error in the Appendix E derivation.
- **LLM reliability**: The discussion now more clearly delineates the role of the LLM and how the operator-graph structure constrains the search.

Based on these revisions and the reviewers’ follow-up messages, there are no remaining outstanding concerns.

**Reviewer Scores:**

Given the rebuttal and subsequent discussion, I expect all reviewers except Reviewer w778 would maintain their original positive overall assessments. Although Reviewer w778 did not respond after the rebuttal, I think the authors provided convincing responses to his concerns.

---

### Decision · Program_Chairs · 2026-01-26

Accept (Poster)